# Physical and virtual nutrient flows in global telecoupled agricultural trade networks

Xiuzhi Chen [1,2,3,12], Yue Hou[2,4,12], Thomas Kastner [5], Liu Liu [1,2], Yuqian Zhang[3], Tuo Yin[2], Mo Li[6], Arunima Malik [7,8], Mengyu Li [8], Kelly R. Thorp [9], Siqi Han[2], Yaoze Liu[10], Tahir Muhammad[2,11], Jianguo Liu [3] ✉ & Yunkai Li [1,2] ✉

Global agricultural trade creates multiple telecoupled flows of nitrogen (N) and phosphorus (P). The flows of physical and virtual nutrients along with trade have discrepant effects on natural resources in different countries. However, existing literature has not quantified or analyzed such effects yet. Here we quantified the physical and virtual N and P flows embedded in the global agricultural trade networks from 1997 to 2016 and elaborated components of the telecoupling framework. The N and P flows both increased continuously and more than 25% of global consumption of nutrients in agricultural products were related to physical nutrient flows, while virtual nutrient flows were equivalent to one-third of the nutrients inputs into global agricultural system. These flows have positive telecoupling effects on saving N and P resources at the global scale. Reducing inefficient trade flows will enhance resource conservation, environmental sustainability in the hyperglobalized world.

Agricultural trade has played a critical role in implementing sustainable development by facilitating global food security and stimulating economic growth[1,2]. With rapid globalization, the economic value of global agricultural trade increased about threefold between 2000 and 2016[3], with consequential effects on the environment and natural resources[4,5]. Increasing cross-border trade exchanges around the world have interlinked the socio-economic and environmental sustainability of different countries and regions[6,7] and have established agricultural trade networks with telecoupled flows of nitrogen (N) and phosphorus (P) between multiple sending and receiving systems (Fig. 1a, b)[8,9]. Sending, receiving and spillover systems are fundamental

subsets of a telecoupled system, as for global trade, there are many such individual subsets (pairs of trading countries) in the trade networks although their roles are not unchangeable (a particular country can be both an exporter/sender and an importer/receiver), the simplified concept of a spillover system and its effects within the telecoupling framework is shown in Supplementary Information Fig. S1. With the aim to understand contemporary sustainability challenges, the conceptual framework of telecoupling[8] reveals the importance of socioeconomic and environmental interactions over distances and facilitates the evaluation of material flows through agricultural trade networks and their socioeconomic and environmental effects.

[1]National Key Laboratory of Efficient Utilization of Agricultural Water Resources, 100083 Beijing, China. [2]College of Water Resources and Civil Engineering, China Agricultural University, 100083 Beijing, China. [3]Department of Fisheries and Wildlife, Center for Systems Integration and Sustainability, Michigan State University, East Lansing, MI 48823, USA. [4]China International Engineer Consulting Cooperation Overseas Consulting Co., Ltd., 100048 Beijing, China. [5]Senckenberg Biodiversity and Climate Research Centre (SBiK-F), Senckenberganlage 25, 60325 Frankfurt-am-Main, Germany. [6]School of Humanities and Social Science, The Chinese University of Hong Kong, Shenzhen, 518172 Shenzhen, China. [7]ISA, School of Physics A28, The University of Sydney, Sydney, NSW, Australia. [8]Discipline of Accounting, The University of Sydney Business School, The University of Sydney, Sydney, NSW, Australia. [9]USDA Agricultural Research Service, 21881 N Cardon Ln., Maricopa, AZ, USA. [10]Department of Environmental and Sustainable Engineering, University at Albany, State University of New York, 1400 Washington Avenue, Albany, NY 12222, USA. [11]College of Hydrology and Water Resources, Hohai University, 210098 Nanjing, China. [12]These authors contributed equally: Xiuzhi Chen, Yue Hou. ✉e-mail: liuji@msu.edu; yunkai@cau.edu.cn

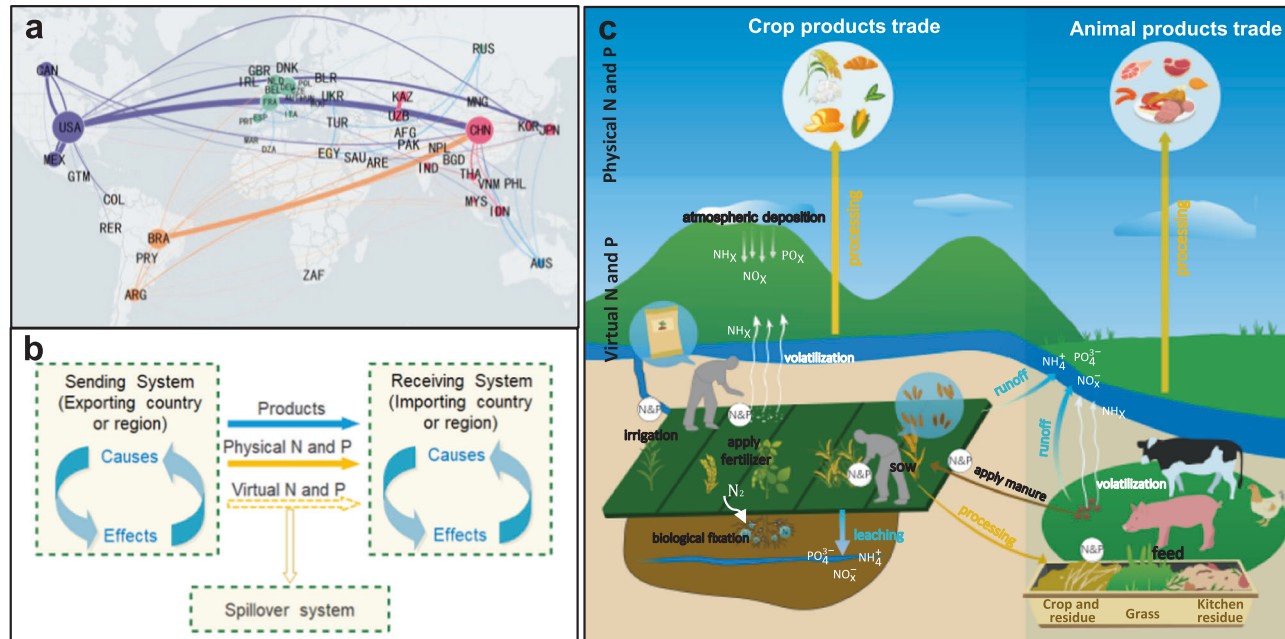

**Fig. 1 | Global nitrogen (N) and phosphorus (P) flows under the telecoupling framework. a** Global agricultural trade networks display the top 100 bilateral agricultural trade routes (the thickness of the trade vector represents the volume of the trade flow); **b** global agricultural trade in the telecoupling framework (subsystems and their effects in the telecoupling framework are illustrated in Supplementary Information Fig. S1); and **c** sources of physical and virtual N and P (**c** was generated by Y.H. using CorelDraw X8 under Microsoft Windows, https://www.coreldraw.com/en/pages/coreldraw-x8/). The base map is applied without endorsement from GADM data (https://gadm.org/).

The increasing flows of water, energy, carbon, and other materials in agricultural trade networks have attracted global attention, making environmental issues the key research points in fields such as ecology and economics[10–13]. Nitrogen (N) and phosphorus (P) are two indispensable elements for life on earth and to ensure high agricultural productivity[14]. In agricultural trade networks, intensive nutrient transfer and exchange through the flows of N and P[10,15] have dramatically changed the global nutrient cycle. Therefore, exploring the environmental and natural resource impacts of nutrient flows in telecoupled agricultural trade networks is important for global sustainable agricultural trade and nutrient management[16–19]. Recent studies have focused on flows of N or P in the global trade of agricultural products. Lassaletta et al.[20] estimated that the flows of N within the global food and animal feed trade increased by eightfold during 1961–2010, accounting for one-third of the total N produced in the world during that time. Nesme et al.[18] reported that the P flows in global agricultural trade increased by 750% during 1961–2011, reaching 17% of total global P-fertilizer input. In addition to the flows of physical N and P in agricultural products, the influence of trade on global N and P cycling is also connected with virtual nutrient flows[20]. Nutrients required for the production process of products are called virtual nutrients (Fig. 1c)[21,22]. Some surplus phosphorus can be stored in soils as legacy P and can be used for future crop uptake. However, a fraction of surplus nutrients are lost to the environment and likely affects air and water quality. Lassaletta et al.[20] and Lun et al.[23] estimated the N and P inputs required for global agricultural production and calculated virtual N and P flows through global agricultural trade, respectively. Malik et al.[24] conducted a global structural decomposition analysis of a change in global reactive nitrogen emissions from 1997 to 2017. James et al.[21] quantified the flows of N through the trade of animal feed products. Schipanski and Bennett[25] analyzed the flows of P caused by international agricultural trade among 12 countries. Xu et al.[26] assessed the evolution of global flows of virtual N and interactions between virtual water, energy, land, and other different flows through international trade networks during 1995–2008, and they distinguished the flows between adjacent versus non-adjacent countries. However, no studies have comprehensively measured both physical and virtual flows of N and P through global agricultural trade simultaneously, therefore making it difficult to compare the different effects of physical and virtual flows or to evaluate the telecoupling effects of N and P flows on a global scale. In addition, due to the different natural resources and environmental conditions of different countries or regions, the risks brought by agricultural trade flows with similar routes are not the same, making it difficult for macro-control and resource optimization.

To fill these important knowledge gaps, the goal of this study was to use the global agricultural trade matrix statistics and the conceptual framework of telecoupling[8] to calculate the physical and virtual flows of N and P for 320 agricultural products, in global agricultural trade among 221 countries or regions during 1997–2016. Specific objectives were to (1) clarify the spatial–temporal dynamic characteristics of physical and virtual flows of N and P in the global telecoupled agricultural trade network; (2) analyze the natural resource depletion and environmental effects of N and P flows on different sending and receiving systems and on global nutrient cycling; and (3) discuss the policy suggestions conducive to sustainable utilization of nutrient resources and environmental protection.

## Results

### Changes in physical and virtual nutrient flows in the global agricultural trade network

Figure 2 shows the trend of total physical and virtual nutrient flows in the global agricultural trade network. The total physical nutrient flows increased from 10.3 to 27.1 Tg N and from 1.4 to 3.5 Tg P, respectively, between 1997 and 2016 (Fig. 2a, b); the global virtual nutrient flows increased from 13.4 to 36.6 Tg N and from 8.9 to 24.5 Tg P, respectively (Fig. 2c, d). The virtual and physical nutrient flows of both N and P increased by about 2.8 times from 1997 to 2016. There are 48 countries with a more than tenfold increase in both physical N and P exports over the period. The country with the largest increase in imports was China, where physical N imports increased by 6.98 times and physical P

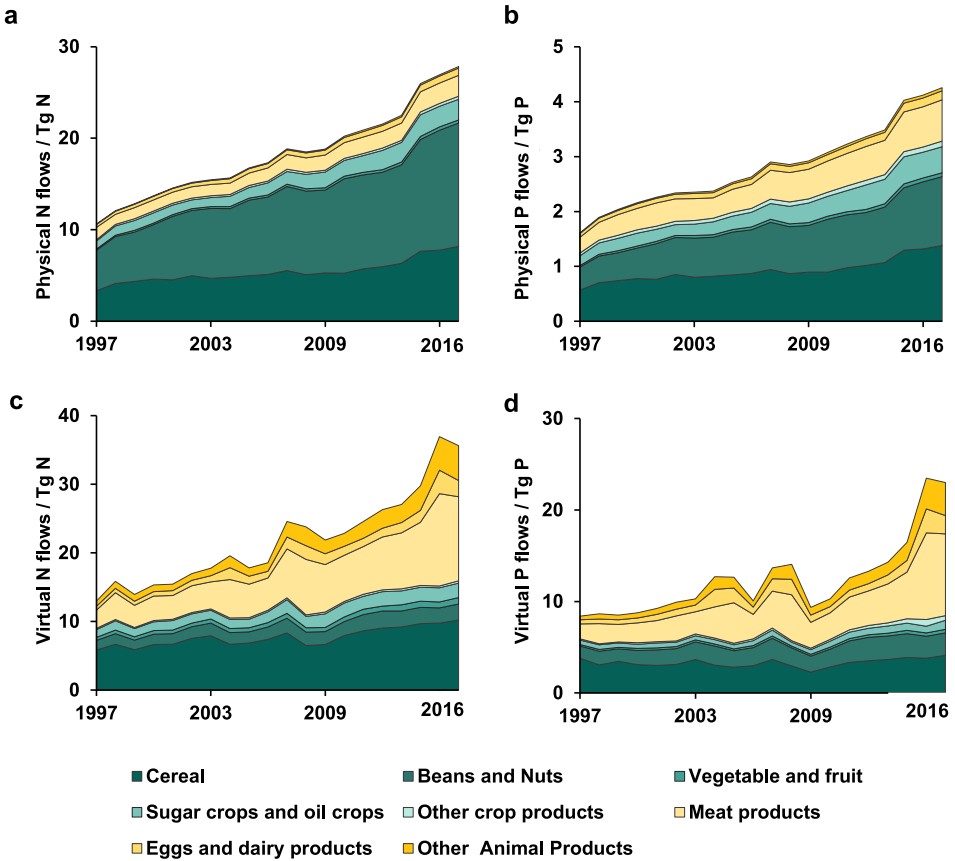

**Fig. 2 | Total physical and virtual nitrogen (N) and phosphorus (P) flows during 1997–2016. a** Physical N flows; **b** physical P flows; **c** virtual N flows; and **d** virtual P flows. All products were divided into eight categories. The specific products contained in each category are provided in Supplementary Data 1.

imports increased by 6.27 times. China was the largest nutrient importer between 2003 and 2016 (Fig. S2).

The physical nutrient flows mainly involved the trade of crop products and their processed products, accounting for 90% of physical flows in 2016 and far exceeding the flows in the trade of animal products. The trade of crops and their processed products contributed to half of the total virtual nutrient (both N and P) flows in 2016, while the trade of animal products accounted for the remaining half. Physical N embedded in soybean and its processed products contributed to the largest proportion of physical N (29.8%) in 2016, with an increasing trend between 1997 and 2016. The proportions of physical and virtual nutrients in beef and other processed products of cattle were substantially different, accounting for 12.4% and 16.8% of the total virtual N and P but only 5.0% and 4.3% of total physical N and P, respectively. The virtual nitrogen of beef is mainly attributed to animal feed, excrement, and other materials input during the production period. The virtual nitrogen transferred during the feeding process is less than the virtual nitrogen input for cultivating animal feed. However, the total virtual nitrogen content of beef and its processed products is significantly higher than that of other plant products. Adjusting the sources of animal feed and improving dietary consumption by replacing beef with other food sources can effectively reduce virtual nitrogen.

**Spatial distribution and variation of net receiving and sending systems**

The largest net receiving systems (e.g. countries, regions) of nutrients were mainly within Asia, while the net sending systems were mainly in North America and South America. The number of net-sending systems decreased from 99 to 62 during 1997 to 2016. By contrast, the net receiving systems increased from 122 in 1997 to 159 in 2016 (Fig. 3). The

top 10 net-sending systems contributed more than 70% of the total exported physical nutrients, and the top 20 net-sending systems contributed to more than 90% of the total exported physical nutrients.

Figure 4 shows the top 5 net receiving and sending systems of nutrients during 2016. China was the largest net receiving system of physical nutrients with 6.06 Tg N and 0.62 Tg P during 2016, accounting for 20.0% and 16.0% of the total global physical N and P flows. Thereafter, Japan (0.95 Tg N and 0.13 Tg P) and Mexico (0.78 Tg N and 0.23 Tg P) were the second and third-ranked receiving systems. The top 5 net receiving systems imported 33% of the total physical nutrient flows. On the other hand, the United States (5.20 Tg N and 0.55 Tg P) and Brazil (4.21 Tg N and 0.39 Tg P) generated the largest net exports during 2016. The total exported physical N and P by the United States and Brazil were 30.0% and 23.1% of the total global physical N and P flows, respectively.

**Spatial distribution of physical and virtual nutrient flows**

The bilateral spatial distribution of nutrient flows between the global sending and receiving systems was complex, involving 221 countries or regions. The number of trade flow routes was 7170 in 1997 and increased to 27,819 in 2016 (Table S3). A total of 41 flow routes were recorded for physical N flows >0.1 Tg, and 59 flow routes for physical P flows >0.01 Tg. Soybeans were calculated as the major agricultural product in the US–China trade flow. Due to the trade of soybean products, 40.1% of China's imported physical N was from the United States, and this accounted for 20.2% of physical N export from the United States. The soybean trade was responsible for a major part of nutrient flows, making US–China the largest export–import flow of physical N. The average flow volume between adjacent systems was higher than that of non-adjacent systems by more than three times (Fig. S3a). However, considering only non-adjacent systems at

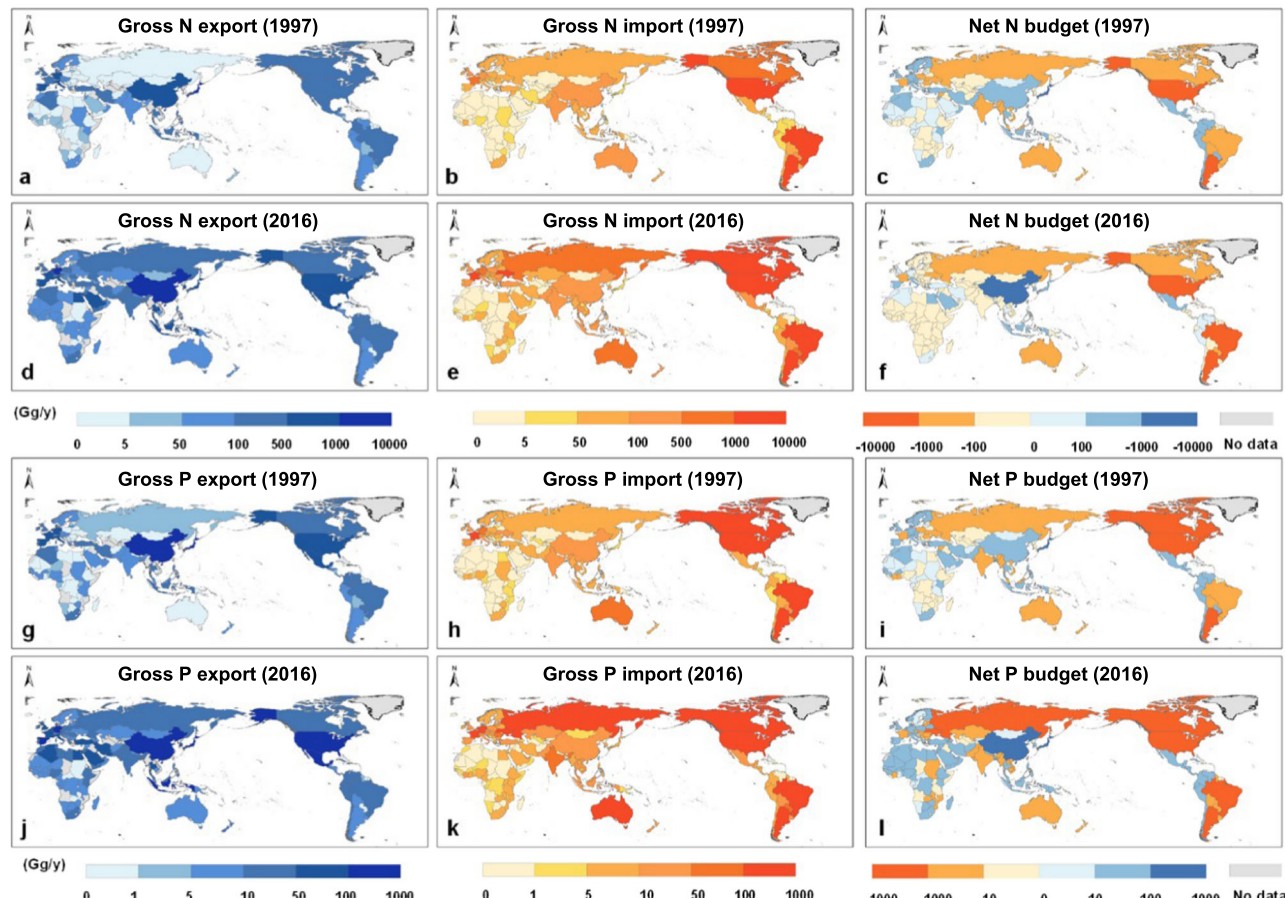

**Fig. 3 | Nitrogen (N) and phosphorus (P) receiving and sending systems in 1997 (a–c, g–i) and 2016 (d–f, j–l).** N flows through gross export (**a**, **d**), gross import (**b**, **e**), and net budget (**c**, **f**); P flows through gross export (**g**, **j**), gross import (**h**, **k**), and net budget (**i**, **l**). Net nutrient flows were calculated as the difference between gross physical N (or P) imported and gross physical N (or P) exported. The red and blue colors indicate exporting and importing, respectively; gray color indicates that no data were available. The base map is applied without endorsement from GADM data (https://gadm.org/).

different distances, average flow volume was positively correlated with distance (Fig. S3b). Among the flow routes in 2016, 256 virtual N flows were more than 1000 times larger than flows of physical N in terms of volume, and 1019 virtual N flows were more than 100 times larger than physical N flows. As for P flows, 409 virtual P flows were more than 1000 times larger than flows of physical P in terms of volume, and 1860 virtual P flows were more than 100 times larger than physical P flows. The virtual N and P flows from Qatar to the Netherlands were strongly driven by physical flows.

Figure 5 shows the spatial distribution of physical and virtual N and P flows associated with global agricultural trade among major country groups by geography in 2016 (see Fig. S4 for N and P flow patterns in 2000, 2005, 2010, and 2015). North American and South American countries were the top two nutrient-sending systems. The physical nutrient export volumes of these two groups during 2016 accounted for 76% and 59% of the total global flows. The export routes from North America to East Asia and from South America to East Asia had the largest nutrient flow, while Southeast Asia was also a major nutrient importer from North and South America. In addition, the volume of physical nutrient exports from Europe was estimated as 21% of the total global flows, but most of the flows occurred among European countries themselves.

**Sending–receiving effects, spillover effects, and telecoupling effects in agricultural trade network**

From a global perspective, the international nutrient flows generated in telecoupled systems of agricultural trade have become an increasingly important part of the global nutrient cycle, leading to the redistribution of nutrient resources. In 2016, the total physical N + P nutrient flows reached nearly 27% of the total volume of N + P resources in the consumption of agricultural products globally, and the total virtual N + P nutrient flows accounted for about 33.7% of the total N + P soil nutrient inputs of the global agricultural system[23,27]. The results demonstrated that the embedded nutrient flows presented significant positive sending–receiving effects (see definition in Methods) on saving N + P resources. The positive sending–receiving effects generally increased from 1997 to 2016 (Fig. 6a, b). Nutrients flow along agricultural products trade from the country or region with high efficiency of nutrient transformation to the country or region with low efficiency, which would save nutrient input compared with producing products in the country or region with low efficiency locally, which was manifested as nutrient saving effect; on the contrary, it was the waste of nutrients. Therefore, the trade flow was defined as an efficient flow when saving nutrients, while the opposite consequence was defined as inefficient. The global physical flows were up to 23.3 Tg N and 3.02 Tg P, respectively. Trade in agricultural products involved a large volume of nutrients and has important implications for global nutrient redistribution. The global virtual sending–receiving effect was up to 62.3 Tg N and 73.9 Tg P. Many products were transferred from high-efficient to low-efficient regions, if not optimal, still showing a highly positive saving effect. The virtual N and P sending–receiving effect of lentils, hazelnuts, coconuts, sunflower seeds, and seed cotton was 100 times higher than the physical N and P sending–receiving effect.

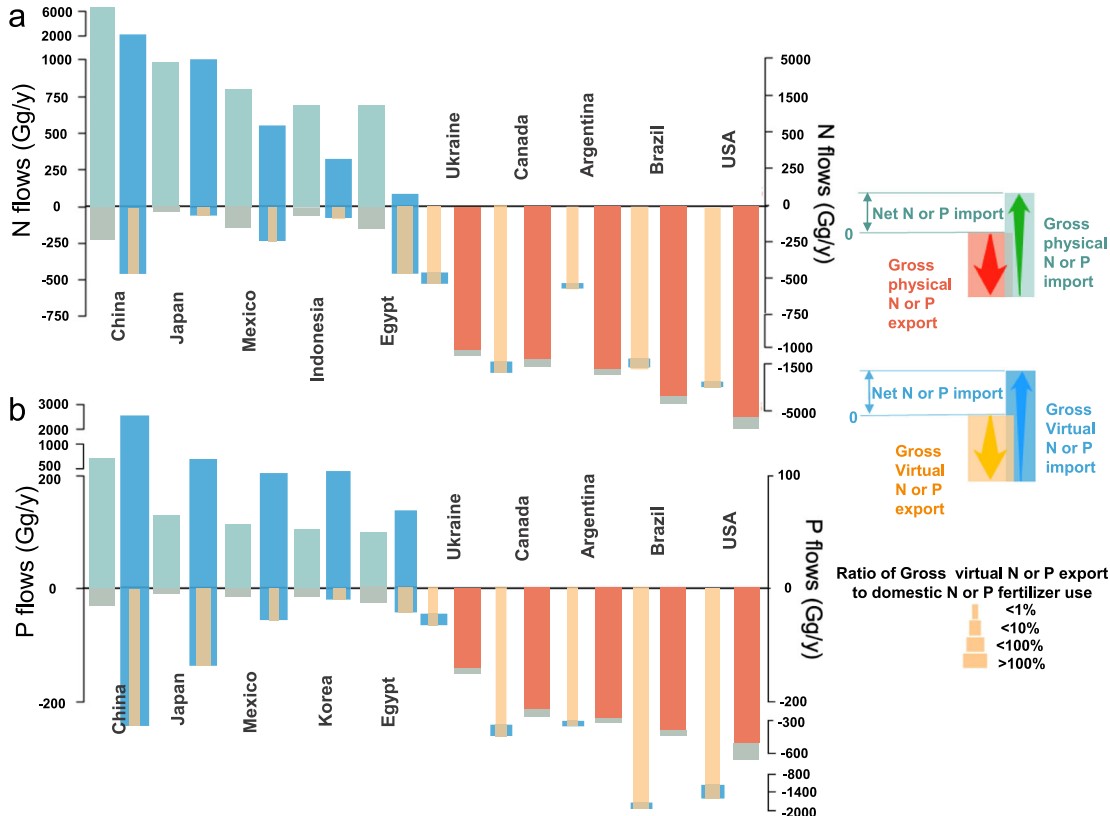

**Fig. 4 | Top 5 net nutrient receiving systems and sending systems in 2016. a** Top 5 net receiving system and sending system of nitrogen (N). **b** Top 5 net receiving systems and sending systems of phosphorus (P). Countries are ordered by the net volume of physical N or P they imported or exported. To compare the difference between import and export, we set the export volume starts at 0 and the import volume starts at the export volume, the part where the two colors do not coincide is the net import or export volume (for example, the red and yellow columns start at 0, while the green or blue columns start at the bottom of the red or yellow columns, the arrows indicate the direction of import or export volume). Results for all countries are shown in Supplementary Data 3.

The spillover effects (see definition in the "Methods" section) for different trade routes were different, due to the ratio of re-export volume to the total trade volume (Fig. 6c, d). The largest physical and virtual N spillover effects were 1.37 Tg for the German–Netherlands route and 7.56 Tg for the China–US route. In general, the spillover effect of physical flow is significantly greater than that of virtual flow, and this difference holds for the entire study period. However, N and P spillovers of both physical and virtual flows show an increasing trend over years. In 2016, the total spillover effect of N flows among global trade entities reached 3.81 Tg, while the total spillover effect of P flow entities reached 1.23 Tg. Wheat, corn, rapeseed, and soybean were the four products with the largest spillover effects, which reached more than 100 Gg. However, the spillover effect of wheat and corn decreased with time; soybean increased first and then decreased; and rapeseed increased across the years. The telecoupling effect of global agricultural trade fluctuated and increased over time, and the sending–receiving effect accounted for a large proportion, indicating that the spillover effects were less than the sending–receiving effects.

## Discussion

Nutrient flows in global agricultural trade networks exhibited a clear uneven structure. Overall, only 10% of the countries contributed about 90% of total exported nutrients in 2016. Trade liberalization made the homogenous products produced by multiple exporters compete in the international market[28]. The positive feedback and increasing demand from receiving systems also contributed to the expansion of production in sending systems[8]. However, to fulfill such intensive demand, sending systems must constantly maintain and develop new croplands, causing damage to forests, grasslands, and other natural ecosystems. For example, the shares of nutrients exported by Brazil in the international market through soybeans increased from 4% to 11% during 1997–2016. A total of 69% of the physical N exported by Brazil came from trading soybeans in 2016, and soybean exports brought considerable economic benefits to Brazil. To meet the massive production demand, Brazil converted a large area of tropical rain forests to soybean planting sites during the past few years, leading to the loss of forest ecosystems[29–31]. Another reason why Brazil's soybean exports increased was the change in the US–China soybean trade flow. China raised tariffs on soybeans imported from the US, which had a spillover effect on China–Brazil soybean trade and catalyzed the land-use changes in Brazil.

In telecoupled agricultural trade network, the larger nutrient exports exposed the net sending systems to the risks of soil nutrient resource deficiency and fertility decline[32,33]. It was because recycling P from water, soil, and other environmental systems after flowing out of the agricultural production system consumes extra energy and economic costs. The long-term continuous export of physical P changed the regional P recycling volume and pattern, resulting in constant loss risks of P nutrients from the original soil and the continuous decrease of regional P resource availability[25]. For instance, due to agricultural exports in 2010, the total volume of physical P flowing out of Argentina reached 0.21 Tg. As a net sending system, only 34% of the total nutrient uptake was recycled back to farmland, and the fertile soft land of the Pampas region faced P depletion risks[34]. Considering the risk of impact on agricultural land, the net outflows of N or P nutrients also affected the N or P cycle of regional croplands. The three most affected countries were Argentina, Ukraine, and the Russian Federation, with the area of $8.71 \times 10^4$, $7.79 \times 10^4$, and $6.15 \times 10^4$ km², respectively (the

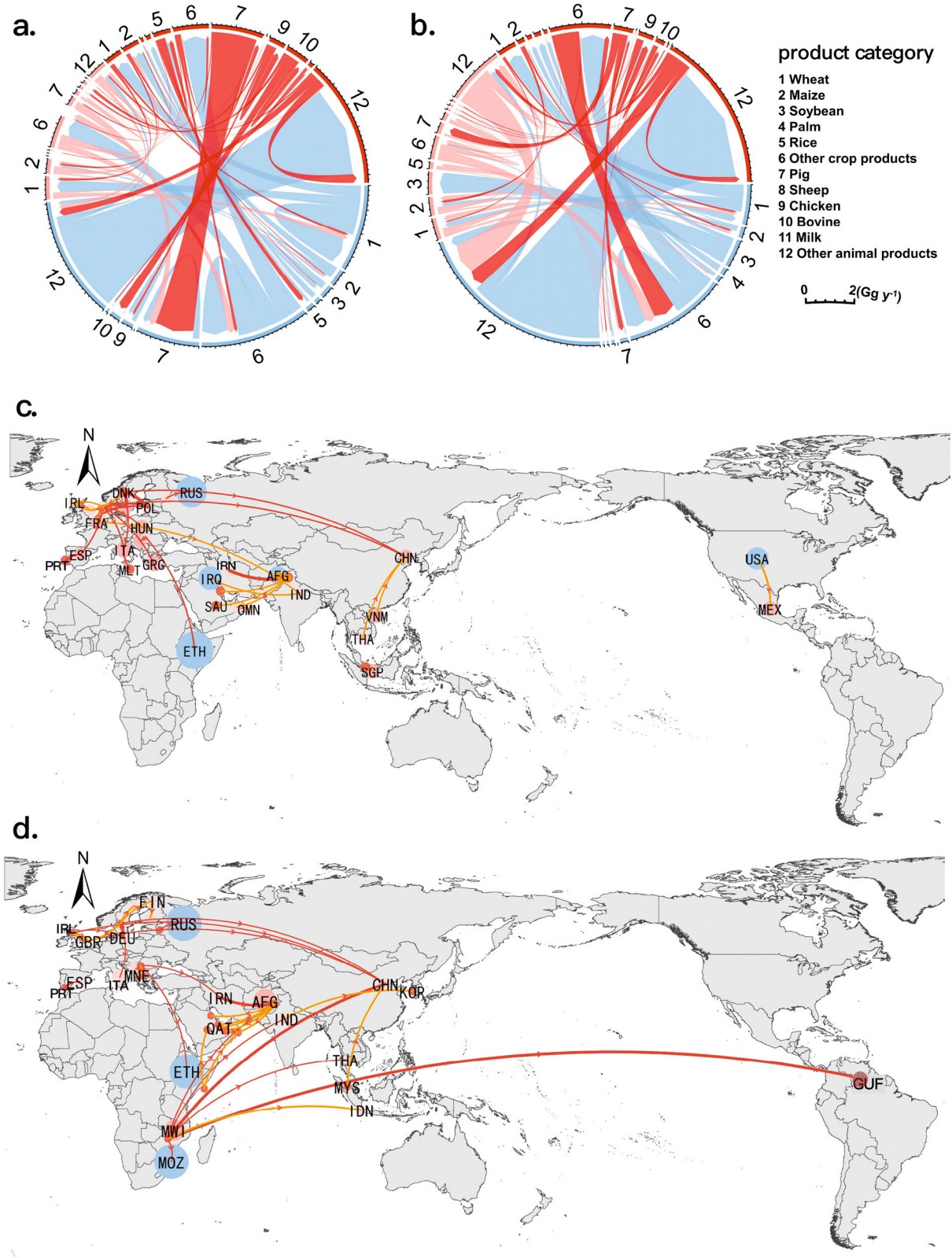

affected area was calculated by total net nutrient outflow and nutrient consumption per unit area). Ukraine, France, and other major nutrient-sending systems were also facing P deficits (Fig. S5b)[23]. Kissinger and Rees[35] explained that export-oriented agricultural production led to the gradual disappearance of Canadian grasslands and sharp declines in soil nutrients, which resulted in risks to regional ecological sustainability. In future research, it is urgent to identify the main sources of nutrient recovery barriers and explore the optimization policies to achieve the sustainability of regional nutrient resources. Also, a research priority should be given to quantify the impact of trade impeding the recycling of nutrients back to the croplands so as to better assess the risk of nutrient losses in net-sending systems.

As regards virtual nutrient flows, the environmental burden was transferred from the receiving systems to the sending systems, which

**Fig. 5 | The nitrogen (N) and phosphorus (P) flow patterns through the trade of agricultural products in 2016 (Gg yr⁻¹). a** Virtual N flows and **b** virtual P flows. The link colors correspond to the sending systems. The arrow points to the receiving systems. According to the global ranking, the agricultural production efficiency of each product in each country is divided into three grades: high, medium, and low. Bright red represents the outflow from the country or region with low efficiency, light red represents the outflow from the country or region with medium efficiency, and the green represents the outflow from the country or region with high efficiency. **c** and **d** Top 20 inefficient routes flow of N (**c**) and P (**d**) in 2016. Red represents the flow of inefficient areas to efficient areas, and yellow represents the flow of inefficient areas to inefficient areas. The bubble size of a country represents the overall nutrient conversion efficiency of the country. Countries are represented by abbreviations. The country names are abbreviated according to the ISO nomenclature. The country's full names and its abbreviations were shown in Supplementary Data 4. The base map is applied without endorsement from GADM data (https://gadm.org/).

was opposite to the flow direction of trade itself[36–38], leading to pollution in sending systems[17]. Li et al.[39] found that 27% of P use in the world was embedded in trade, which was all generated by sending systems. For example, the United States exported 3.67 Tg of virtual N and was the largest sending system of nutrients in 2016. Therefore, net-sending systems must formulate reasonable nutrient management strategies and account for the environmental and resources cost when pricing exported agricultural products.

Countries or regions with food shortages improved food and nutrition security through global agricultural trade[3], and imports became the main nutrient source for several countries in Africa and Asia. For example, the P inflow from agricultural products imported to Jordan in 2016 was 43 times greater than that of its own fertilizer input. For countries or regions with deficient soil nutrients, the inflow of physical nutrients through import trade could supplement nutrient reserves and finally alleviate soil fertility deficiencies through nutrient cycling, such as the return of manure and the application of organic fertilizer. Africa imported 0.38 Tg physical P through agricultural trade in 2016, which significantly alleviated P deficits in Africa[40]. The import of virtual nutrients meant the conservation of domestic nutrient resources, especially P (due to the low cost-effectiveness of recycling through natural cycling, Fig. S6). For instance, the imports to Africa in 2016 embedded 0.37 Tg of virtual P, meaning that the sending system instead of Africa consumed these resources.

Importing virtual nutrients also represented the transfer of the environmental burden from the receiving systems to the sending systems[40]. For example, the U.S. suffered pollution by leaking 0.11 million tons of N to the local environment from exporting pork and chicken to Japan[21]. Although the import of agricultural products could avoid environmental pollution caused by the production process, it increased food consumption, particularly for meat products. It also increased the N and P flowing into the soil through the food chain, leveraging the risk of water eutrophication for countries or regions with limited soil absorption capacity[25]. Many net receiving systems or regions such as South Korea and Japan had increasing environmental risks due to large imports of physical nutrients (Fig. S7)[41]. Changes in land use induced by imports could be another cause of environmental pollution. For instance, Sun et al[42] found that the conversion of soybean lands to corn fields and rice paddies due to soybean imports resulted in N pollution in importing countries such as China because more fertilizers were applied for other crops as compared to soybeans.

Although the global trade of agricultural products presented an overall positive effect on saving N and P resources, some inefficient flows resulted in an unnecessary loss. For example, rice exported from India to Croatia in 2016 resulted in an accumulative loss of 0.05 Tg N due to the higher nutrient use efficiency of local rice production in Croatia. We recommend that countries eliminate those inefficient flows with negative effects, by adjusting the production and trade structure and transferring the sending system to the regions with high nutrient utilization efficiency, which is of great significance for the further conservation of global nutrient resources[11]. In addition, reducing meat consumption can help reduce inefficient production at the consumer end. For example, the United Nations has recommended that EU member states reduce their consumption of meat and dairy

products. At the same time, it is also necessary to upgrade technology in countries or regions with low production efficiency.

Future studies need to consider how the agents of the telecoupling framework are represented in agricultural trade. Agents in agricultural trade include producers, agribusinesses, public and private investors, traders, financial investors, and consumers of products[8,43]. Governments are key agents in global trade, which are responsible for making and enforcing trade agreements and policies influencing domestic production and consumption[13]. Different from our research, some spillover effects considered upstream fertilizer producers who export nitrogen and phosphorus fertilizers as well as downstream importers of refined agricultural products[16]. The boundary of our research was bilateral agricultural trade flows, excluding the upstream or downstream agricultural industry chain.

The telecoupling framework enables investigations into how the countries or regions contribute to nutrient flows and how their contribution can influence the sending and receiving systems in the network. The international nutrient flows in telecoupled agricultural trade networks have become an increasingly important part of global N and P cycling, so policies to improve nutrient management should fully assess the impact of international trade on the N and P cycle to achieve win–win solutions. It is urgent to incorporate environmental risks and soil nutrient resources into national agricultural planning and evaluation frameworks to realize green and sustainable production and consumption[25]. Furthermore, efforts should be made to conserve nutrient resources and reduce pollution on the production side, including recycling organic manure and biosolids and use of phosphate solubilizing bacteria as bio-fertilizer instead of chemical fertilizers[44]. Globally, it is important that countries or regions importing products through inefficient trade flows enforce trade regulations and green procurement to adjust the inefficient flows by revising trade policy[39,45]. This will ensure the comparative advantage of nutrient utilization efficiency from sending systems to receiving systems and further expand the positive effects of agricultural product trade worldwide.

Restricted by the relevant data, this study did not consider the difference in nutritional element content between export products and non-export products. Future studies could further examine the difference between these two kinds of products to clarify the trade effect more precisely (see the uncertainty analysis in Supplementary Information Section S2). Future studies should focus on how the other aspects of the telecoupling framework are represented in global agricultural trade, including agents and spillover effects[46]. Further improvements in data integrity with a high level of detail in the upstream and downstream product supply chains are needed to provide a more sophisticated understanding of the influences of agents on trade flows and spillover effects[47]. A comprehensive understanding of the impacts of agricultural trade on global N and P use through the telecoupling framework can facilitate better management to achieve global food safety, environmental health, and sustainability.

## Methods
### Agricultural trade in the context of the telecoupling framework
Agricultural trade networks are subsets of the global trade networks, which have typical network characteristics (Fig. 1a)[48]. A total of 221

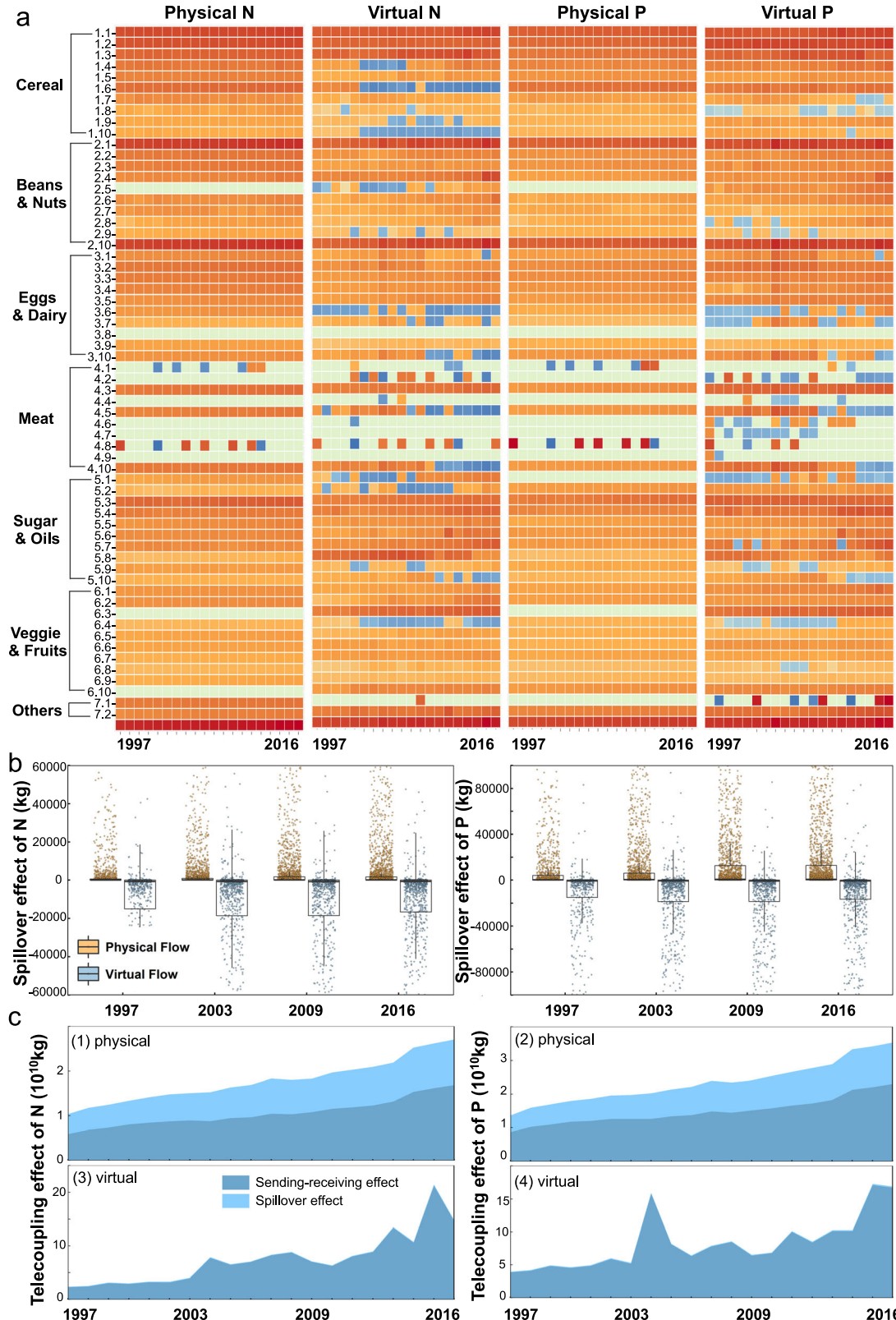

**Fig. 6 | Sending–receiving effects, spillover effects, and telecoupling effects of the agricultural trade network. a** Physical and virtual sending–receiving effects of nitrogen (N) and phosphorus (P) flow for each kind of product (the order number and agricultural products are shown in Supplementary Information Table S4, the red represents positive values and the blue represents negative values). **b** Spillover effects of nitrogen (N) and phosphorus (P) flow in 1997, 2003, 2009, and 2016. **c** (1), (2) Global agricultural trade physical telecoupling effect of nitrogen (N) and phosphorus (P) from 1997 to 2016. **c** (3), (4) Global agricultural trade virtual telecoupling effect of nitrogen (N) and phosphorus (P) from 1997 to 2016 (all data were provided in Supplementary Data 1).

countries or regions and 27,819 different export–import routes were involved in global agricultural trade networks in 2016[49]. In this study, we use the telecoupling framework[8] to analyze the international agricultural trade networks because it can facilitate a more comprehensive and systematic understanding than conventional trade research. In this study, the conceptual model of agricultural trade mainly focused on physical and virtual nutrient flows among different countries or regions. In telecoupled agricultural trade network, exporting and importing countries (or regions) are defined as sending and receiving systems, respectively; and the countries (or regions) that are affected by trade but not directly involved in the trade are defined as 'spillover systems'[8,13]. Global trade networks are aggregations of thousands of trade routes. For a specific trade route (an individual telecoupled system), importing countries (or regions) not only represent receiving systems but also play roles as 're-export transit stations' (e.g., Singapore imports rice from India and then exports rice to Australia, it is not directly consuming the rice imported from India, it serves only as a transit point for rice). The simplified diagram of telecoupling framework is shown in Supplementary Information Fig. S1. It is worth noting that the roles of sending, receiving, and spillover systems in the trade networks are not fixed, and these three are only clear in each specific trade route; in other words, a country or region may have multiple roles in the global trade network.

Along with the flow of agricultural products, physical and virtual nutrients move from sending to receiving systems. The socioeconomic and environmental effects are produced by the interactions between sending and receiving systems through trade and the accompanying nutrient flows. Each of the nutrient flows contributes to the changes in the global allocation of nutrient resources[18], which in turn has telecoupling effects on nutrient cycling and utilization on a global scale[32,50]. The systems flow, and effects are also interconnected with two other components of the telecoupling framework: causes (reasons behind the flows) and agents (decision-making entities such as traders that facilitate the flows)[51,52], but causes and agents are beyond the scope of this paper.

### Nutrient flows in telecoupled agricultural trade network

The nutrient flows embedded in the trade of agricultural products are quantified by multiplying the weight of agricultural products traded between sending and receiving systems by the N or P content per unit weight of products. The equation for calculating the N or P flows of a trade route is as follows:

$$F_{i,A,B} = C_{i,A,B} \times T_{i,A,B} \tag{1}$$

where $F_{i,A,B}$ are the physical or virtual N or P flows embedded in the trade of product $i$ exported from country $A$ to country $B$, t; $C_{i,A,B}$ are the physical or virtual of N or P contents of product $i$, %; and is the trade volume of product $i$ exported from country $A$ to country $B$, t. The flow of physical nutrients considers the direct trade data from the FAO trade matrix. Since the efficiency of nutrient use (virtual nutrient content) in agricultural production systems varies among different countries or regions, we used the data that excludes re-exports to match the nutrients flow to real producer/consumer, this method can realize the traceability of trade products and virtual nutrients flows[53,54]. Physical nutrients, which are the physical N or P elements contained in the products harvested by the agricultural production system, are transferred from the exporter to the importer along the international trade route. Compared with Lun et al.[23] and Barbieri et al.[55], this study expanded to 320 crops, including almost all conventionally traded products, the detailed description of these selected agricultural products and their physical N and P contents are listed in Supplementary Data 1.

In this paper, source–sink analysis was used to calculate virtual nutrients. In addition to fertilizer input, natural input is considered to make the results more complete[50]. The sources of virtual nutrients include the application of inorganic and organic fertilizers, seeds, irrigation water, atmospheric deposition, and biological N fixation (Fig. 1c)[38]. The nutrients, which are not absorbed by the products, e.g., discharged into the water or atmosphere and accumulated in the soil, will lead to negative impacts on the environment[51]. While physical nutrients are considered as nutrients contained in traded products, virtual nutrients are the total inputs in the production of agricultural products, whether being absorbed or not. Therefore, the flows of virtual nutrients involve the importer transferring inputs and losses of nutrients in the production process to the exporter[52,53]. The virtual nutrient contents of various agricultural products from each country or region and their specific calculation methods are explained in Table S3 and Supplementary Information Sections 1.2 and 1.3.

The total N and P flows through the international trade of agricultural products were calculated annually by adding the embedded nutrient flows of all trade routes. The final net import or export of each country or region was obtained by subtracting the total nutrient inflow associated with the import trade from the total nutrient outflow of the export trade. The total N and P flows among pairs of systems were obtained by adding all the nutrient flows between the two systems.

### Sending–receiving effects

Trade leads to the sending–receiving effects of redistributing global nutrient resource utilizations. As for physical nutrients, the sending–receiving effect is actual physical nutrients transferred from $A$ to $B$, and the specific calculation formula is

$$\mathrm{NSRE}_{\mathrm{p},i} = \sum_{l=1}^{m} N_{\mathrm{p},i} \times \left( T_{i,A,B} - T_{i,B,C_k} \right) \tag{2}$$

$$\mathrm{PSRE}_{\mathrm{p},i} = \sum_{l=1}^{m} P_{\mathrm{p},i} \times \left( T_{i,A,B} - T_{i,B,C_k} \right) \tag{3}$$

where $A$ and $B$ are the sending and receiving systems, respectively, in the $l$th trade route or nutrient flow of the product $i$, $l$th is the trade route number (from 1 to $m$); $T_{i,A,B}$ is the trade volume of product $i$ exported from A to B, t; $T_{i,B,C_k}$ is the trade volume of product $i$ re-exported from B to $C_k$ (the endpoints of re-export trade), t; $N_{\mathrm{p},i}$ and $P_{\mathrm{p},i}$ is the physical N and P content of product $i$; $\mathrm{NSRE}_{\mathrm{p},i}$ and $\mathrm{PSRE}_{\mathrm{p},i}$ are the physical sending–receiving effects of the product $i$.

This study explored whether the sending–receiving effects on global N and P utilizations were positive (nutrient saving) or negative (nutrient wasting). The specific calculation method is

$$\mathrm{NSRE}_{\mathrm{v},i} = \sum_{l=1}^{n} \left( N_{\mathrm{v},i,B} - N_{\mathrm{v},i,A} \right) \times \left( T_{i,A,B} - T_{i,B,C_k} \right) \tag{4}$$

$$\mathrm{PSRE}_{\mathrm{v},i} = \sum_{l=1}^{n} \left( P_{\mathrm{v},i,B} - P_{\mathrm{v},i,A} \right) \times \left( T_{i,A,B} - T_{i,B,C_k} \right) \tag{5}$$

where, $N_{\mathrm{v},i,A}$, $N_{\mathrm{v},i,B}$, $P_{\mathrm{v},i,A}$, $P_{\mathrm{v},i,B}$ are the virtual N and P contents of local product $i$ in country or region A and B, kg t$^{-1}$; $\mathrm{NSRE}_{\mathrm{v},i}$ and $\mathrm{PSRE}_{\mathrm{v},i}$ are the virtual sending–receiving effects of the product $i$. The sending–receiving effects of nutrient flows were obtained by the flows of all agricultural trade products considered in this study. If $\mathrm{NSRE}_{\mathrm{v},i}$ or $\mathrm{PSRE}_{\mathrm{v},i} > 0$, it indicates that the redistribution of agricultural product trades on the utilization of nutrient resources presents a positive sending–receiving the effect of saving N or P, which are defined as efficient flows; if $\mathrm{NSRE}_{\mathrm{v},i}$ or $\mathrm{PSRE}_{\mathrm{v},i} < 0$, it indicates a negative sending–receiving effect on the utilization of nutrient resources of wasting N or P, which are defined as inefficient flows; $n$ is the total number of nutrients flows.

## Spillover and telecoupling effects

Due to the existence of re-export trade, there is no complete overlap between trade importers, exporting countries, agricultural producers, and consumers. Moreover, due to the differences in production efficiency among countries, spillover effects are considered in this paper to quantify the nutrient impacts brought by the re-export trade process. The calculation method of spillover effect for physical and virtual flow is:

$$NSE_{p,l} = \sum_{i=1}^{m} N_{p,i} \times T_{i,B,C_k} \tag{6}$$

$$PSE_{p,l} = \sum_{i=1}^{m} P_{p,i} \times T_{i,B,C_k} \tag{7}$$

$$NSE_{v,l} = \sum_{i=1}^{n} \left( N_{v,i,C_k} - N_{v,i,A} \right) \times T_{i,B,C_k} \tag{8}$$

$$PSE_{v,l} = \sum_{i=1}^{n} \left( P_{v,i,C_k} - P_{v,i,A} \right) \times T_{i,B,C_k} \tag{9}$$

where $NSE_{p,l}$, $PSE_{p,l}$ are physical spillover effect of the $l$th trade route, kg; $NSE_{v,l}$, $PSE_{v,l}$ are virtual spillover effect of the $l$th trade route, kg; $N_{v,i,C_k}$ and $P_{v,i,C_k}$ are the virtual N, P contents of local product $i$ in systems $C_k$, kg t$^{-1}$; it is calculated according to the re-export volume from B to $C_k$ and the virtual N content of $C_k$; the telecoupling effect is the summation of sending–receiving effect and spillover effect of a specific trade route.

The physical telecoupling effect is the total amount of physical sending–receiving effects and physical spillover effects. Similarly, the virtual telecoupling effect is the summation of the virtual sending–receiving effects and virtual spillover effects:

$$NTE_p = \sum_{l=1}^{m} NSRE_{p,l} + \sum_{l=1}^{m} NSE_{p,l} \tag{10}$$

$$PTE_p = \sum_{l=1}^{m} PSRE_{p,l} + \sum_{l=1}^{m} PSE_{p,l} \tag{11}$$

$$NTE_v = \sum_{l=1}^{n} NSRE_{v,l} + \sum_{l=1}^{n} NSE_{v,l} \tag{12}$$

$$PTE_v = \sum_{l=1}^{n} PSRE_{v,l} + \sum_{l=1}^{m} PSE_{p,l} \tag{13}$$

where $NTE_p$, $PTE_p$ is the telecoupling effects of N and P flows, kg; $NTE_v$, $PTE_v$ is the virtual telecoupling effects of N and P flows, kg; and NTE, PTE is the telecoupling effects of N and P flows, kg.

## Data source

A total of 320 major agricultural products were considered in this study, including 17 major grain crops (e.g., barley, corn, rice, soybeans, and wheat), processed products obtained from the 17 crops (e.g., flour, bran, soybean oil), five types of livestock products (pork, beef, poultry, sheep, and goat), and other processed products. The detailed list of agricultural products is in Supplementary Data 1. These selected products together accounted for about 95% of global caloric consumption. The total volume of trade (320 agricultural products) was obtained from the FAOSTAT trade matrix[49]. Fish products were not included in this analysis. Although the volume of the global fish trade is growing, a large part of the product in the global fish trade still comes from direct capture, accounting for about 75% in 2016[54,55]. Therefore, the potential loss caused by this is small because there is less input of virtual nutrients (such as feed) in the operation of capture fisheries[52,56].

## Data availability

The authors declare that all data necessary to support the findings of this study are within the manuscript, supplementary information and supplementary datasets 1–4[57–61].

## Code availability

The drawing plots and computer codes are made using the open-source software R 4.0.2 and Python 3.8. The codes used in this work can be accessed by contacting the corresponding authors.

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

## Acknowledgements

The authors are grateful for financial support from the National Natural Science Foundation of China (51621061 and 51321001, Yunkai Li), National Science Foundation (1924111, J.L.), International Postdoctoral Exchange Fellowship Program (PC2021090, X.C.), World Sustainability Award (J.L.), and Gunnerus Award in Sustainability Science (J.L.), Deutsche Forschungs Gemeinschaft (German Research Foundation, KA 4815/1-1, T.K.), German Federal Ministry for Economic Cooperation and Development (GS22 E1070-0060/029, T.K.), Natural Science Foundation of Guangdong Province (2023A1515011815, Mo Li), Australian Research Council Grants (DE230101652, DP200103005 and LP200100311, A.M.). We also acknowledge data and help from Graham Macdonald.

## Author contributions

Yunkai Li designed the research. X.C. and Y.H. wrote the manuscript. X.C., Y.H., and T.Y. analyzed the data. Yunkai Li, J.L., K.T., L.L., Mo Li, Y.Z., A.M., Mengyu Li, S.H., Yaoze Liu, T.M. and T.K. provided comments on the manuscript. All authors reviewed the manuscript.

## Competing interests

The authors declare no competing interests.
