## [Peer Review File · Nature Communications]

Physical and virtual nutrient flows in global telecoupled agricultural trade networkReviewer #1 (Remarks to the Author):

Comments on manuscript NCOMMS-21-34571

This manuscript provides an overview of global physical and virtual N and P flows through trade of major agricultural products. It also explores to what extent trade of agricultural products helps to save N or P resources. The analysis provided here is of great importance to understand the global N and P cycles. As far as I am aware of the literature, this global study on both physical and virtual N and P flows is original. More precisely, putting together both physical and virtual nutrient flows is probably the greatest asset of this manuscript. Therefore, I think this manuscript is solid and has some potential for Nature Communications. However, I think it needs to undergo really serious revision before it can be considered for publication.

Major comments

- **The manuscript suffers from poor redaction in far too many places. This makes the text lacking of the needed rigorous expression. See my many detailed comments below. A serious and thorough check of the manuscript is needed. To be honest, I am on the edge of considering that such a lack of rigour of the text and figures is actually a reason for rejecting the manuscript...**
- **The authors have calculated virtual nutrient flows by considering any N or P source that is provided to soils. This includes chemical fertilisers but also animal manure, biological N fixation, nutrient in irrigation water, etc. This makes quite a difference with other studies that estimated virtual nutrient flows by focussing on chemical fertilisers only; see for instance 1,2. Such a difference in methods has to be detailed and seriously discussed by the authors. In particular, I recommend to provide (i) the reasons why these authors have decided to consider all the sources of nutrient added to soils and (ii) some comparison with previous estimates provided by the literature, e.g. about P for the USA or Europe.**
- **The authors proposed a method to estimate to what extent trade helps to save (or loose) nutrient resources. Although I understand the method that was used, I think it lacks two aspects. First, the self-reliant scenario (that consists of relocating any imported product within the imported country) is by itself unrealistic in many aspects: importing countries may lack of croplands, water resources, labour or even nutrient resources. This unrealistic aspect of this self-reliant scenario has to be discussed. Second, by concluding that traded products that flow from high nutrient-use efficient countries to low nutrient-use efficient countries helps to save nutrient resources, the authors neglect that trade is also a barrier to recycle nutrients back to the croplands where traded products have been produced. This should also be discussed and eventually quantified.**
- **One great asset of this manuscript is the quantification of both physical and virtual nutrient flow in the same single paper. However, the authors have not really compared those two kinds of estimates. It may be interesting, though, to explore to what extent virtual nutrient flows are driven by physical flows. In particular, I am curious to see if traded products are associated with higher or lower virtual nutrient requirements than non-traded products. Additional analysis may be expected here.**
- **Methods for estimating virtual flows are really vague and poorly presented (suppl section S1). Clear and detailed methods (with complete presentation, appropriate units, and actual data sources) are lacking for explaining how nutrient inputs to soils have been estimated by crop and by country. I also recommend that fertiliser and manure application rates to soils by crop and by country are given in a supplementary table.**

Minor comments

- **L37 and throughout the text: what does 'embodied' mean for the authors? It looks like it means 'physical' for them, whereas it is generally understood as 'virtual' in the many body of the literature (see for instance reference 1). This has to be clarified.**
- **L38: what does 'the global consumption of N/P in agricultural products' means? This is a typical example of very vague sentence that has to be avoided. Also, I strongly recommend to avoid using 'N/P' throughout the manuscript (including in the title): this**

is too confusing and does help to understand if you refer to N, P, N or P, or N+P.

- L39: please define 'virtual' here.
- L41: I can't understand that sentence 'more distant countries or regions tend to produce more N/P flows'.
- Figure 1: where does the panel a about agricultural trade network represent? Where do the data come from? Panel b: are you sure there is not a confusion between importing and exporting countries? Panel c: I can't understand the difference between 'actual' and 'virtual' flows here. And does 'actual' mean 'physical'?
- L82-83 (and line 84): I disagree. Not all nutrients that are supplied to soils in excess compared to crop uptake are lost to the environment, in particular for P. Some phosphorus can be stored in soils as legacy P and can be used for future crop uptake.
- L83: please define what 'virtual nutrient flows' are in your study.
- L88-89: what do you mean by 'interactions of the global flows of virtual N'?
- L98: what does 'differentiate' mean here?
- L107 and throughout the manuscript: please use SI units, not 109 kg P or similar inappropriate units.
- L108: the physical P trade flow is estimated to be 2.2 Tg P/yr in 2016 this study. This is quite lower than the estimated 3.0 Tg P/yr in 2011 by 3. Please provide some explanation for this lower value.
- L111-112: I can't understand that sentence.
- L112-113: why focussing on those countries? Those have a rather second role in agricultural trade, making their import or export data sensitive to single changes in trade in some specific years. Those countries also suffer from poor reporting of their trade data in international databases such as FAOStat. I recommend selecting countries with more solid statistical data and/or greater contribution to trade.
- L117-118: do this refer to crop commodities (instead of 'agricultural commodities')? Moreover, I don't understand how the 97.9% has been calculated.
- L122: does 'nutrient' refer to N or P??
- L122-125 and Figure 2: I am really surprised by the massive contribution of cattle to virtual nutrient flows. Can you provide more explanation about this?
- L139-142: those lines can be compressed.
- L154: does 'sending' mean 'exporting'? If yes, please select one term and use it consistently throughout the manuscript.
- L156: what does 'total nutrient flows' mean? Is this the sum of imported N (or P) flows?
- L155-161: the date provided in the text do not match the number provided in Fig4. Please clarify!
- Fig4: this is a nice and important figure but many aspects have to be improved. First, clarify if 'inflow' and 'outflow' at the extreme right of the figure are synonyms of 'import' and 'export'. Second, please clarify why green bars (representing imports) may exist below the X axis for some specific exporting countries (eg, France) and not above the X axis. Third, please clarify what the 'ratio of domestic N fertiliser use' means and how it has been calculated. Fourth, please explain and justify why the virtual flows have been plotted for export only (and not for import of virtual flows).
- L170: do you refer to physical or virtual flows?
- L171: please replace complicated by complex.
- L173-174: please replace 'more than' by 'greater than'.
- L179: I can't understand that sentence 'a total of 45.5%...'
- Figure S5 is interesting but it is quite contradictory with Table 1. Fig S5 shows that flows from adjacent countries contribute more to global N or P physical flows than flows from non-adjacent countries. This is in contrast with Table 1 that shows that major physical flows occur overseas, between countries from different continents. Please clarify.
- L190-191: which nutrient do you refer?
- L194 'in Europe': do you mean 'to' or 'from' Europe? And later in the same sentence: what does 'total global flows' mean?
- Figure 5: please provide numbers and units! Also, please provide differentiation between inflows and outflows to and from each world region.
- L212: please replace 'exported' by 'contributed'.

- L212-213: this sentence is quite speculative. Please provide solid justification and references.
- L218: I am not sure that the environmental damages that are detailed here are due to the nutrient-rich nature of croplands. Please reformulate more specifically.
- L224: what does 'demand in Brazil's soybean exports' mean?
- L225: can you detail the years or period for the China-US trade war? In my understanding, this trade war is posterior to the study period ending in 2016.
- L231: what does 'total volume of participation in the regional P recycling system' mean?
- L234: please provide the data source.
- L258-259: I think I agree with this idea but it has to be developed. In fact, the nutrients imported through trade do not reach agricultural soils directly. They may ultimately reach agricultural soils if those traded products are used for feed or food use and if animal manure and human excreta are recycled back to agricultural soils.
- L261-263: unclear sentences.
- L263-265: this is a repetition of lines 241-243.
- L268-269: I can't understand that sentence.
- L283-295: these new results should not be within the Discussion but should instead be moved to the Result section. Also, I'm not sure using 'waste' of nutrient is appropriate. Please consider using 'loss' or something similar instead.
- Fig6: please explain how products were ranked.
- L329-335: this is quite vague. Please improve.
- L338-359: those lines could be compressed.
- L381: although I'm wrong, I could not find this Table S3.
- L387: are you sure this is 'inflow' and not 'outflow'?
- L398: I can't understand what you mean about 'the lth trade route of the product i' Moreover, I can't see any l index in the equation at the right of the sigma.
- L403: please clarify 'the redistribution of ag product trades on the utilisation....'.
- L416: 60% of global caloric consumption is quite low. This means that the list of products you considered is in fact quite limited. I recommend that you add more products in your list. Also, please provide the corresponding estimate of crop production represented by this list of crop species you considered.
- Supp Section S1.2: please express your data in kg/ha not kg/hm². Also, please provide precise data sources, in particular for fertiliser and manure application rate by crop and by country.
- Supp L65: please double check this is 'organic' fertiliser.
- Supp L98: is this footprint or input?
- Suppl L71-75: this is confused. Please make this more explicit.
- Suppl L106-111: again, this is really confused. Please make this more explicit.
- Suppl L116: what does 'or obtains a variety of products' mean??
- Suppl L121-126: I can't understand why market values were used instead of more classical weighting by product biomass or caloric content. Also, I could not understand the difference between r and c. Finally, how do you account for raw products that lead to two different products that are considered in your study (e.g., soybean grains lead to soy oil and soy cakes)?
- Suppl L130: how did you account for supplementary P feed?
- Suppl L156-164: this is quite a vague discussion. Either improve or omit.

References

1. MacDonald, G. K., Bennett, E. M. & Carpenter, S. R. Embodied phosphorus and the global connections of United States agriculture. *Environ. Res. Lett.* 7, 44024 (2012).
2. Nesme, T., Roques, S., Metson, G. S. & Bennett, E. M. The surprisingly small but increasing role of international agricultural trade on the European Union's dependence on mineral phosphorus fertiliser. *Environ. Res. Lett.* 11, 025003 (2016).
3. Nesme, T., Metson, G. S. & Bennett, E. M. Global phosphorus flows through agricultural trade. *Glob. Environ. Chang.* 50, (2018).

Reviewer #2 (Remarks to the Author):

In the present manuscript, the authors present a global quantification and analysis of nitrogen (N) and phosphorus (P) physical (embedded) and virtual flows through commercial exchanges between 188 countries for the year 1997 to 2016 and 42 agricultural commodities. Furthermore, the paper aims at analysing the natural resource use and environmental impacts of N and P flows on the sending (exporting) and receiving (importing) systems and on global nutrient cycles, as well as to present some policy suggestions for a more sustainable use of nutrient resources. Despite some studies about physical and virtual nutrients resources in trade have been recently published, no study have presented a simultaneous estimation of such different flows. Therefore, the authors present here an original analysis that well complements the existing literature on this topic. A priori, The topic and questions raised fit for a publication in Nature Communication.

Despite the overall interest of the manuscript, I have a few major concerns that I would like to raise.

My first major comment is referred to the methodology to calculate virtual N and P flows. From the detailed methods section in your Supplementary Information, I cannot understand how the virtual flows for each of the 42 agricultural commodities was calculated, in particular for the virtual inputs referred to chemical fertilizers. According to lines 63-64, the authors calculated the "total nitrogen input for crops". To do so, they considered, for each country, the total amount of different N-based synthetic fertilizers divided by the total yield of all grain crops – all data derived mainly from FAOSTAT – . This is understandable, since FAOSTAT does not report any specific fertilization rate for each crop species. Nevertheless, most results are then presented for a set of crop commodities (e.g. Figure 2). It is not clear how the authors calculated virtual N and P resources for each crop commodity based on the methodological description. This is a major problem that requires a detailed explanation, since it could impact most of the manuscript results. Note also that in most papers dealing with virtual nutrient resources, only synthetic fertilizers are accounted for. Here the authors consider all nutrients inputs to soils; I think that this choice should be justified. Furthermore, I am not totally convinced by the accuracy of the calculation process used to transform processed commodities to primary commodities (S.1.2.2 and equation 8 in the SI). The authors use a price ratio (i.e. the ratio of the market value of the product to the aggregate market value of all products) to transform processed to primary products. Whether I think to understand the reasoning behind this choice, I am skeptical whether the choice of this coefficient is actually compelling. I guess that this has been done to avoid any double counting. That is that, for example, to produce 1 kg of soybean oils has required a certain amount of soybeans. But such soybeans has also produced some oilseed-cake. Thus, the transformation of these two sub-products in the original primary product should avoid any double counting. This is totally correct, but most studies use a coefficient based on the kcal ratio between the sub-product and the primary product to transform processed commodities into primary commodities (for example, McDonald et al. (2015) Rethinking agricultural trade relationships in an era of globalization). I am thus not sure that using a price index leads to similar results. In addition, the source of such ratios – reported in Table S2 – (and of commodities' prices) is not mentioned.

My second major comment concerns a general confusion between the results reporting over physical and virtual N/P flows. I would invite the authors to clearly distinguish whether the results and discussions are referred either to physical or virtual resources, since this is often not so clear. Especially, all results about net flows seem to refer to physical nutrients flows; if it is the case this should be clearly defined. A great plus of the present work, as mentioned in the introduction, is to simultaneously report over both physical and virtual flows. To better stress this innovative point, I think that the authors could expand their discussion bringing a few information about the relation between virtual and physical flows. For example, does countries characterized by high

physical exports shows above or below average virtual nutrients flows?

Finally, concerning the three objectives mentioned at line 101-103, I hardly found any discussion regarding policy suggestions. If this objective is maintained, I think that the authors has a bit of room for improving their discussion to these terms.

In addition to such major comments, here a series of minor comments that may help the author's in revising the paper.

L37: I think that the correct English term should be "N/P flows embedded" and not embodied. Furthermore, at the same line, to what the "other components" refers? Figure 1: panel a: flows are difficult to distinguish. Maybe filter on the most important flows? Panel b: I find this representation not totally clear: what are the causes and effects shown here? Does it refers to environmental effects? How is a spillover system defined?

L98: please briefly defined if the 42 products includes crops and/or livestock products, and their share in comparison to total trade flows.

L101: natural resources depletion ?

L119-120: Are you referring here to crop based nutrients flows, or animal products are also included here?

L125-127: this results is important, since it shows that the trade of livestock products has a hige impact on virtual resources (and so potential environmental impact). Maybe this result could be better developed in the discussion?

Figure 2: As referred to my previous comment, how the different virtual flows for each traded category have been estimated?

L138: net receiving/exporting systems are defined only in Figure 4. I think this definition should appear here. Furthermore, it is unclear whether the authors refers to physical or virtual resources.

L146: total physical or virtual nutrients?

Figure 4: It may be interesting to add the total virtual imports (if the figure does not become too charged).

L176: There is no Table S3 in the supplementary information.

L177 (and throughout the all paper): please, be consistent with the units used (sometimes kg, sometimes Tg, etc). Can you report all the results to the main unit – i.e. Tg ?

L173-188: I think that this paragraph could shot down to gain some space for expanding some of the results/discussions following my previous comments.

L188: you mention here a significance analysis, but I couldn't find any information on such analysis in the methods. This should be clarified.

Table 1: if I am not wrong, some of the numbers reported in coloumn "Virtual P" do not correspond to flows showed in Figure 5. For example, the flow from BRA+ARG (SA) to CHN (E.AS) of 0.67Tg does not match with the flow reported in figure 5b (0,28Tg)

Figure 5. This figure is interesting but I would revise it to (i) delete all minors flows, (ii) indicated the values in Tg on the round axis. As mentioned, also check also the match with data reported in table 1.

L216: why agricultural production efficiency was the main economic cause for high export? Either (i) a result or (ii) a reference should be brought in support.

L223: 72% of physical or virtual nutrients?

L232-233: this is true for P, but much less the case for N (which do not accumulate in soils).

L233-234: this sentence is unclear. Please clarify. Why would it possible to recycle N and not P.

L234-236: This would not be the case if exports are compensated by fertilizers inputs.

L239: P depletion → how was the P budget calculated? Is this a calculation done by the authors?

L239-241: I agree that P depletion would happen in the long term. But negative P budgets can be partially buffered by high soil P stocks. I think that the authors should discuss this point here. Furthermore, looking at figure S7b, it seems that Net exporter countries are not necessarily linked with a negative P budget. Maybe there is a cleared

link with total P virtual exports?

L248-249: I think this example is out of place here.

L259 "imports became the main nutrient source": this is just theoretical as a number of organic waste resources after the consumption of agricultural commodities are not exploited in many countries (especially waste-water resources).

L266-263: "can supplement nutrients reserves": this is true for P, but not for N

L264: maybe you could report an indicator, such as a ration between the P imported over P deficit?

L266: how does P regenerates? This formulation is unclear to me.

L268: I would start a new paragraph here.

L272-273: Why this promotes "the increase of nutrient consumption"? Unclear to me.

L276: please revise this sentence (English formulation)

Figure 6: this is a nice figure. I would just invert the colors if possible (red for wasting and blue for savings).

L313-315: this is especially true for P, which is a resource that is highly concentrated in a few countries (morocco, US, china).

L345: here are "well defined network characteristics?"

L356: How where "spillover systems" considered in this analysis?

L386: referred to my main comment #1, are virtual flows calculated for each commodity or aggregated? Please clarify.

L422-425: does the authors accounted for re-exports – i.e. commodities produced by country A, could transit through country B, to be finally consumed in country C → the FAOSTAT trade matrix does not accounts for re-exports. Some previous works on flows has developed procedures for this (Kastner et al. 2011 Tracing distant environmental impacts of agricultural products from a consumer perspective). I think that the authors should be clarified if re-exports were accounted for or not.

Supplementary information

L 112-113: for manure, FAOSTAT does not provided P inputs. How was this estimated?

L140: why primary livestock products are also transformed here?

Figure S4, L208-212: I suggest to report this lines also to the caption of Figure 3.

Figure S7: As mentioned before, please clarify how P budgets were calculated

figure S8: Please, verify the quality of the figure. What does black and white colored countries stand for? Maybe you could also scale down your legend (no dark blue/red countries in the maps).

Point by point responses(bold) to reviewers' comments(regular)

REVIEWER COMMENTS

Reviewer #1 (Remarks to the Author):

Comments on manuscript NCOMMS-21-34571

• This manuscript provides an overview of global physical and virtual N and P flows through trade of major agricultural products. It also explores to what extent trade of agricultural products helps to save N or P resources. The analysis provided here is of great importance to understand the global N and P cycles. As far as I am aware of the literature, this global study on both physical and virtual N and P flows is original. More precisely, putting together both physical and virtual nutrient flows is probably the greatest asset of this manuscript. Therefore, I think this manuscript is solid and has some potential for Nature Communications. However, I think it needs to undergo serious revision before it can be considered for publication.

Answer: Thanks a lot for your comments and support for the innovation of this article! We agree with you and have revised the manuscript according to your valuable suggestions. The point-by-point responses and revised details are as follows.

Major comments

• The manuscript suffers from poor redaction in far too many places. This makes the text lacking of the needed rigorous expression. See my many detailed comments below. A serious and thorough check of the manuscript is needed. To be honest, I am on the edge of considering that such a lack of rigour of the text and figures is actually a reason for rejecting the manuscript...

Answer: Thank you for your comments. We have revised our writing with clear and concise words to ensure rigor and accurate expression. Furthermore, we have revised the text and figures.

• The authors have calculated virtual nutrient flows by considering any N or P source that is provided to soils. This includes chemical fertilisers but also animal manure, biological N fixation, nutrient in irrigation water, etc. This makes quite a difference with other studies that estimated virtual nutrient flows by focussing on chemical fertilisers only; see for instance 1,2. Such a difference in methods has to be detailed and seriously discussed by the authors. I recommend to provide (i) the reasons why these authors have decided to consider all the sources of nutrient added to soils and (ii) some comparison with previous estimates provided by the literature, e.g., about P for the USA or Europe.

Answer: Thank you very much for your comments! In order to compare with physical nutrients, we defined the boundary of virtual nutrients as all nutrient inputs, referring to the

concept of virtual water¹. The source of physical nutrients is not only chemical fertilizer, but also other nutrient sources (including manure, seed input, atmospheric deposition, etc.) which also have a significant impact on the local soil and water environment. In the revised manuscript, we have added comparisons and explanations for the differences with previous studies.

We have also revised the Introduction and Method sections to explain the method and scope of the virtual nutrient calculation.

- The authors proposed a method to estimate to what extent trade helps to save (or loose) nutrient resources. Although I understand the method that was used, I think it lacks two aspects. First, the self-reliant scenario (that consists of relocating any imported product within the imported country) is by itself unrealistic in many aspects: importing countries may lack of croplands, water resources, labour or even nutrient resources. This unrealistic aspect of this self-reliant scenario has to be discussed. Second, by concluding that traded products that flow from high nutrient-use efficient countries to low nutrient-use efficient countries helps to save nutrient resources, the authors neglect that trade is also a barrier to recycle nutrients back to the croplands where traded products have been produced. This should also be discussed and eventually quantified.

Answer: Thank you for your comments! We have deleted the self-reliant scenario. Based on the analysis of the FAO production database, we found that about 7% of trade is not produced by consumer countries themselves, which is a small proportion would not influence the main trend, thus we have used the global average value to assume the virtual nutrient inputs of these countries. We have re-considered the telecoupling effects under the telecoupling framework. We have clarified the quantification method of sending-receiving effects, spillover effects and telecoupling effects and did quantity analysis. The revised section is as follows:

“Agricultural trade in the context of the telecoupling framework. Agricultural trade networks are subsets of global trade networks, which have typical network characteristics (Fig. 1a). A total of 221 countries or regions and 27819 different export-import routes were involved in global agricultural trade networks in 2016⁴⁵. The international agricultural trade networks were viewed through the telecoupling framework⁸ that facilitates more comprehensive and systematic understanding than conventional trade research. In this study, the conceptual model of agricultural trade mainly focused on physical and virtual nutrient flows among different countries or regions. In telecoupled agricultural trade networks, exporting and importing countries (or regions) are defined as sending and receiving systems, respectively; and the countries (or regions) that are affected by trade but not directly involved in trade are defined as "spillover systems" ^{8,13}. Global trade network is an aggregation of thousands of trade routes. For a specific trade route (telecoupling subsystem), importing or exporting countries (or regions) not only represent sending and receiving systems but also play roles as “re-export transit stations”. For example, Singapore, which imports rice from India and re-

exports it to Australia, is not directly involved in the production and consumption of the rice imported from India. In this case, Australia is a spillover system. The diagram of spillover systems in the telecoupling framework is shown in Supplementary Materials Fig. S1. It is worth noting that the roles of sending, receiving and spillover systems in the trade network are not fixed, these three are only clear in each specific trade route; in other words, a country or region may have multiple roles in global trade network.”

Sending-receiving effects

Trade leads to the sending-receiving effects of redistributing global nutrient resource utilizations. As for physical nutrients, the sending-receiving effect is actual physical nutrients transferred from A to B, the specific calculation formula is:

$$NSRE_{p,i} = \sum_{l=1}^m N_{p,i} \times (T_{i,A,B} - T_{i,B,C_k}) \quad (2)$$

$$PSRE_{p,i} = \sum_{l=1}^m P_{p,i} \times (T_{i,A,B} - T_{i,B,C_k}) \quad (3)$$

where, A and B are the sending and receiving systems, respectively, in the l th trade route or nutrients flow of the product i ; $T_{i,A,B}$ is the trade volume of product i exported from A to B, t ; T_{i,B,C_k} is the trade volume of product i re-exported from B to C_k , t ; $N_{p,i}$ and $P_{p,i}$ is the N, P content of product i ; $NSRE_{p,i}$ and $PSRE_{p,i}$ are the physical sending-receiving effects of the product i ; m is the total number of trade routes.

This study explored whether the sending-receiving effects on global N, P utilizations were positive (nutrient saving) or negative (nutrient wasting). The specific calculation method is:

$$NSRE_{v,i} = \sum_{l=1}^n (N_{v,i,B} - N_{v,i,A}) \times (T_{i,A,B} - T_{i,B,C_k}) \quad (4)$$

$$PSRE_{v,i} = \sum_{l=1}^n (P_{v,i,B} - P_{v,i,A}) \times (T_{i,A,B} - T_{i,B,C_k}) \quad (5)$$

where, $N_{v,i,A}$, $N_{v,i,B}$, $P_{v,i,A}$, $P_{v,i,B}$ are the virtual N, P contents of local product i in systems A and B, kg/t; $NSRE_{v,i}$ and $PSRE_{v,i}$ are the virtual sending-receiving effects of the product i . The sending-receiving effects of nutrient flows were obtained by the flows of all agricultural trade products considered in this study. If $NSRE_{v,i}$ or $PSRE_{v,i} > 0$, it indicates that the redistribution of agricultural product trades on the utilization of nutrient resources present a positive sending-receiving effect of saving N or P; if $NSRE_{v,i}$ or $PSRE_{v,i} < 0$, it indicates a negative sending-receiving effect on the utilization of nutrient resources of wasting N or P; n is the total number of nutrients flows.

Spillover effect and telecoupling effect.

The spillover effect of physical trade flow and virtual nutrients flow were calculated by ratio of re-export to total export for receiving system. The calculation method of spillover effect for physical and virtual flow is:

$$NSE_{p,l} = \sum_{i=1}^m N_{p,i} \times T_{i,B,C_k} \quad (6)$$

$$PSE_{p,l} = \sum_{i=1}^m P_{p,i} \times T_{i,B,C_k} \quad (7)$$

$$NSE_{v,l} = \sum_{i=1}^n (N_{v,i,C_k} - N_{v,i,A}) \times T_{i,B,C_k} \quad (8)$$

$$PSE_{v,l} = \sum_{i=1}^n (P_{v,i,C_k} - P_{v,i,A}) \times T_{i,B,C_k} \quad (9)$$

where $NSE_{p,l}$, $PSE_{p,l}$ are physical spillover effect of the l th trade route, kg; $NSE_{v,l}$, $PSE_{v,l}$ are virtual spillover effect of the l th trade route, kg; N_{v,i,C_k} and P_{v,i,C_k} are the virtual N, P contents of local product i in systems C_k , kg/t, it is calculated according to the re-export volume from B to C_k and the virtual N content of C_k ; the telecoupling effect is the summation of sending-receiving effect and spillover effect of a specific trade route. The calculation method is:

Physical telecoupling effect is the total amount of physical sending-receiving effect and physical spillover effect. Similarly, the virtual telecoupling effect is the summation of virtual sending-receiving effect and virtual spillover effect:

$$NTE_p = \sum_{l=1}^m NSRE_{p,l} + \sum_{l=1}^m NSE_{p,l} \quad (10)$$

$$PTE_p = \sum_{l=1}^m PSRE_{p,l} + \sum_{l=1}^m PSE_{p,l} \quad (11)$$

$$NTE_v = \sum_{l=1}^n NSRE_{v,l} + \sum_{l=1}^n NSE_{v,l} \quad (12)$$

$$PTE_v = \sum_{l=1}^n PSRE_{v,l} + \sum_{l=1}^n PSE_{v,l} \quad (13)$$

where NTE_p , PTE_p is the telecoupling effects for N, P flows, kg; NTE_v , PTE_v is the virtual telecoupling effects for N, P flows, kg; NTE , PTE is the telecoupling effects for N, P flows, kg.

And we added the results for sending-receiving effect, spillover effect, and telecoupling effect, see details as follows:

“Sending-receiving effects, spillover effects, and telecoupling effects in agricultural trade network. From a global perspective, the international nutrient flows generated in telecoupled systems of agricultural trade have become an increasingly important part of the global nutrient cycle, leading to the redistribution of nutrient resources. In 2016, the total physical N + P flows reached nearly 27% of the total volume of N + P in global agricultural products consumption and the total virtual N + P flows accounted for about 33.7% of the total N + P inputs into the global agricultural system^{23,27}. The results demonstrate that the embedded nutrient flows presented significant positive sending-receiving effects on saving N + P resources. The positive sending-receiving effects generally increased from 1997-2016 (Fig.6a, 6b). The trade flow is defined as an efficient flow when saving nutrients, while the opposite consequence is defined as inefficient. For example, the trade flow of soybeans from Brazil to China is efficient. The global physical sending-receiving effect of N and P was up to 23.3 Tg and 3.02 Tg, respectively, and the global virtual sending-receiving effect of N and P was up to 62.3 Tg and 73.9 Tg, respectively, showing a high positive saving effect. The virtual N and P sending-receiving effects of lentils, hazelnuts, coconuts, sunflower seed, and seed cotton is 100 times higher than the physical N and P sending-receiving effect.

The dispersion degree of spillover effect of different trade routes is different, which is related to the total import and export volume and re-export rate of trading countries (Fig. 6c, 6d). The largest physical N and virtual N spillover effects were 1.37 Tg for the German-Netherlands route and 7.56 Tg for the China-USA route, respectively. In general, the spillover effect of physical flow is significantly greater than that of virtual flow, and this difference holds for the entire study period. However, as time goes by, N and P spillovers of both physical and virtual flows show an increasing trend. In 2016, the total spillover effect of N flows among global trade entities reached 3.81 Tg, while the total spillover effect of P flow entities reached 1.23 Tg. Wheat, corn, rapeseed, and soybean were the four products with the largest spillover effects, which reached more than 100 Gg. However, the spillover effect of wheat and corn decreased with time, soybean increased first and then decreased, and rapeseed increased by years. The telecoupling effect of global agricultural trade fluctuates and increases over time, and the sending-receiving effect accounts for a large proportion, which indicates that the spillover effects are less than the sending-receiving effects.

Fig. 6. Sending-receiving effects, spillover effects, and telecoupling effects of agricultural trade networks. **a.** Physical and virtual sending-receiving effects of nitrogen (N) and phosphorus (P) flow for each kind of product (red represents positive values and blue represents negative values); **b, c.** spillover effects of nitrogen (N) and phosphorus (P) flow in 1997, 2003, 2009, and 2016; **d, e.** global agricultural trade physical telecoupling effect of nitrogen (N) and phosphorus (P) from 1997 to 2016; **f, g.** global agricultural trade virtual telecoupling effect of nitrogen (N) and phosphorus (P) from 1997 to 2016.

For the second comment, we agreed with your opinion, the trade itself has changed the nutrient cycling and made it more difficult for nutrient recycling to cropland systems. However, there is still a lack of data on nutrient flow process. When sufficient data are available, we can conduct quantitative calculation on this issue in the future. The discussion and simple estimation are made here. According to the virtual nutrient budget, we calculated the land area affected by the net nutrient outflow caused by trade through the output per unit area and N, P input of each country, and carried out quantitative analysis. “For instance, due to agricultural exports in 2010, the total volume of physical P flowing out of Argentina reached 0.21 Tg. As a net sending system, only 34% of the total nutrient uptake was recycled back to farmland, and the fertile soft land of the Pampas region has faced P depletion risks³⁵. Net outflow of N or P nutrients equivalent to blocking the N or P cycle of cropland, especially for P. The three most affected countries are Argentina, Ukraine, and the Russian Federation, with the area of 8.71, 7.79, and 6.15 million hm², respectively. Ukraine, France, and other major nutrient sending systems are also facing P deficits (Fig.S7b)²⁴. In future studies, based on detailed recycling and processing data of kitchen residue and excrement in cities and villages of different countries, it is necessary to quantify the impact of trade impeding recycle of nutrients back to the croplands to assess the risk of nutrient loss in net sending systems”

- One great asset of this manuscript is the quantification of both physical and virtual nutrient flow in the same single paper. However, the authors have not really compared those two kinds of estimates. It may be interesting, though, to explore to what extent virtual nutrient flows are driven by physical flows. I am curious to see if traded products are associated with higher or lower virtual nutrient requirements than non-traded products. Additional analysis may be expected here.

Answer: Thank you for your comments! We added a comparison of the physical versus virtual flow relationship in the Results section. At the same time, we have re-considered the telecoupling effects under the telecoupling framework. We have clarified the quantification method of sending-receiving effects, and analyzed how physical sending-receiving effects drives virtual sending-receiving effects. The revised section is as follows:

“Among the flow routes in 2016, 256 virtual N flows were more than 1000 times larger than flows of physical N in terms of volume, and 1019 virtual N flows were more than 100 times larger than physical N flows. As for P flows, 409 virtual P flows were more than 1000 times larger than flows of physical P in terms of volume, and 1860 virtual P flows were more than 100 times larger than physical P flows. The virtual N and P flows from Qatar to the Netherlands were strongly driven by physical flows.

Sending-receiving effects

Trade leads to the sending-receiving effects of redistributing global nutrient resource utilizations. As for physical nutrients, the sending-receiving effect is actual physical nutrients

transferred from A to B, the specific calculation formula is:

$$NSRE_{p,i} = \sum_{l=1}^m N_{p,i} \times (T_{i,A,B} - T_{i,B,C_k}) \quad (2)$$

$$PSRE_{p,i} = \sum_{l=1}^m P_{p,i} \times (T_{i,A,B} - T_{i,B,C_k}) \quad (3)$$

where, A and B are the sending and receiving systems, respectively, in the l th trade route or nutrients flow of the product i ; $T_{i,A,B}$ is the trade volume of product i exported from A to B, t; T_{i,B,C_k} is the trade volume of product i re-exported from B to C_k , t; $N_{p,i}$ and $P_{p,i}$ is the N, P content of product i ; $NSRE_{p,i}$ and $PSRE_{p,i}$ are the physical sending-receiving effects of the product i ; m is the total number of trade routes.

This study explored whether the sending-receiving effects on global N, P utilizations were positive (nutrient saving) or negative (nutrient wasting). The specific calculation method is:

$$NSRE_{v,i} = \sum_{l=1}^n (N_{v,i,B} - N_{v,i,A}) \times (T_{i,A,B} - T_{i,B,C_k}) \quad (4)$$

$$PSRE_{v,i} = \sum_{l=1}^n (P_{v,i,B} - P_{v,i,A}) \times (T_{i,A,B} - T_{i,B,C_k}) \quad (5)$$

where, $N_{v,i,A}$, $N_{v,i,B}$, $P_{v,i,A}$, $P_{v,i,B}$ are the virtual N, P contents of local product i in systems A and B, kg/t; $NSRE_{v,i}$ and $PSRE_{v,i}$ are the virtual sending-receiving effects of the product i . The sending-receiving effects of nutrient flows were obtained by the flows of all agricultural trade products considered in this study. If $NSRE_{v,i}$ or $PSRE_{v,i} > 0$, it indicates that the redistribution of agricultural product trades on the utilization of nutrient resources present a positive sending-receiving effect of saving N or P; if $NSRE_{v,i}$ or $PSRE_{v,i} < 0$, it indicates a negative sending-receiving effect on the utilization of nutrient resources of wasting N or P; n is the total number of nutrients flows.

The global physical sending-receiving effect of N and P was up to 23.3 Tg and 3.02 Tg, respectively, and the global virtual sending-receiving effect of N and P was up to 62.3 Tg and 73.9 Tg, respectively, showing a high positive saving effect. The virtual N and P sending-receiving effects of lentils, hazelnuts, coconuts, sunflower seed, and seed cotton is 100 times higher than the physical N and P sending-receiving effect.”

Fig. S1 Systems and effects within an agricultural trade network under the telecoupling framework. a. Schematic diagram of global agricultural trade network; b. A specific trade route shows sending system, receiving system, and spillover system as well as sending-receiving effect, spillover effect, and telecoupling effect.

But now we are not able to distinguish the content of virtual nutrients of traded and non-traded products in the calculation because of the lack of data to distinguish the differences of nutrients consumption in planting period between export production and local consumed production, and it is difficult to compare virtual nutrient requirements of these two. We think it would be an interesting research question that requires a separate quantification in the future. We have revised and discussed this question in the Discussion section as follows: “Restricted by the relevant data, this study did not consider the difference of nutritional element content between export products and non-export products. Future studies could further consider the difference between these two kinds of products to clarify the trade effect more precisely.”

- Methods for estimating virtual flows are vague and poorly presented (suppl section S1). Clear and detailed methods (with complete presentation, appropriate units, and actual data sources) are lacking for explaining how nutrient inputs to soils have been estimated by crop and by country. I also recommend that fertiliser and manure application rates to soils by crop and by country are given in a supplementary table.

Answer: Thank you for your comment! We have added the explanation of the calculation method of virtual nutrients in detail in the Supplementary Materials Section S1 (including data source, calculation formula etc.). We have added an Excel file named Supplementary Data2 that contains information on fertiliser and manure application rates to soils by crop and by country.

We updated the N or P application rate method by N or P balance, using the data of crop-

specific yield, N and P content in crops, crop harvest area fraction, and N and P balance (which responded to excess or deficiency of nitrogen and phosphorus compared to the current level). The N or P calculated by this method is the actual nutrient input in farmland, including fertilizer, manure, etc. The detailed method and revised Section S1 are as follows:

“Section S1. Methods for calculating the unit nutrient contents of agricultural products

S1.2 Calculation of virtual nutrient contents of agricultural products

S1.2.1 Calculation of virtual nutrient contents of crops

In this study, gridded crop-specific fertilizer, and manure application data at 0.083 degrees, downloaded from EarthStat (referred to the global baseline level were used for the calculation, as shown in the following equation:

$$N_{app} = crop_{yield} \times N_{content} + crop_{HAF} \times crop_{balance} \quad (1)$$

where N_{app} is N application rate, kg; $crop_{yield}$ is crop-specific yield, kg; $N_{content}$ is N content; $crop_{HAF}$ is harvested area fraction of the specific crop; $crop_{balance}$ is the N balance of crop, kg.

The calculation process for phosphorus was similar:

$$P_{app} = crop_{yield} \times P_{content} + crop_{HAF} \times crop_{balance} \quad (2)$$

where P_{app} is P application rate, kg; $crop_{yield}$ is crop-specific yield, kg; $P_{content}$ is P content; $crop_{HAF}$ is harvested area fraction of the specific crop; $crop_{balance}$ is the P balance of crop, kg.

N and P content data were sourced from <https://fao.org/economic/the-statistics-division-ess/publications-studies/publications/nutritive-factors/en/> and <https://www.gov.uk/government/publications/composition-of-foods-integrated-dataset-cofid>. Crop-specific without N or P content were replaced by taking the average of the same item group. Crop-specific yields and other related data were sourced from <http://www.earthstat.org/>.

The N and P application rate of 102 different crops in different countries in 2000 were finally obtained. The results obtained by this calculation method matched with the actual N and P application rate. Then, the actual N and P application rate and the calculated N and P application rate were linearly scaled from 2000 to 2016 over all countries to ensure that the total N and P application rate of 102 crops was consistent with the actual N and P application rate.

It should be noted that N and P contents used in this study were based on the weight of the fruiting part rather than the whole crop-specific yield, which could be uncertain for global-scale calculation. In addition, the N balance deficit of some countries was too large, resulting in negative N and P application rates. In this study, this paper treated it as an outlier and normalized to 0.”

S1.2.2 Virtual nutrient calculation of primary livestock products

The virtual nutrients of primary animal products mainly come from feed inputs. We used the feed conversion coefficient to convert virtual nutrients of animals into virtual nutrients of feed

crops. The specific formula is as follows:

$$N_i = C_i \times F_i \times N_{feed} \quad (3)$$

$$P_i = C_i \times F_i \times P_{feed} \quad (4)$$

where N_i, P_i is virtual N or P content of the animal product, kg/t; C_i is feed conversion coefficient of the animal, kg/kg, retrieved from Mekonnen & Hoekstra¹; F_i is fraction of crop-based feed in total feed, retrieved from Mekonnen & Hoekstra¹; N_{feed}, P_{feed} is virtual N or P content of crop-based feed of the animal product, kg/t, crop-based feed were partitioned among seven main grain feed commodities - barley, maize, peas, rapeseed, sorghum, soybean, and wheat—according to data retrieved from Herrero et al.² All the data for feed conversion coefficients, fractions, and compositions of grains in crop-based feed by animal species and by region are provided in Table S2-S4.

S1.3 Calculation of virtual nutrient contents of processed products

We use factors based on caloric equivalents, according to the method by Kastner³ to transform the virtual nutrient contents of primary crop or animal products into that of processed products. The calculation formula is as follows:

$$N_{i,p} = K_i \times N_i \quad (5)$$

$$P_{i,p} = K_i \times P_i \quad (6)$$

where $N_{i,p}, P_{i,p}$ is virtual N or P content of the processed product, kg/t; N_i, P_i is virtual N or P content of the primary product, kg/t; K_i is kcal ration between the processed product and the primary product.

The kcal contents of all products are provided in Table S1, according to FAO standard factors on nutritive values (<https://www.fao.org/economic/the-statistics-division-ess/publications-studies/publications/nutritive-factors/en/>)."

The detailed description of 320 selected agricultural products and their physical N, P contents are listed in Table S1 in Supplementary Materials.

Minor comments

- L37 and throughout the text: what does ‘embodied’ mean for the authors? It looks like it means ‘physical’ for them, whereas it is generally understood as ‘virtual’ in the many bodies of the literature (see for instance reference 1). This has to be clarified.

Answer: Thank you for your comments! We have taken the suggestion based on reviewer#2 and replaced it with “N and P flows embedded” which is more accurate in this article. We have double checked the whole manuscript and revised.

- L38: what does ‘the global consumption of N/P in agricultural products’ means? This is a typical example of very vague sentence that has to be avoided. Also, I strongly recommend to avoid using ‘N/P’ throughout the manuscript (including in the title): this is too confusing and does help to

understand if you refer to N, P, N or P, or N+P.

Answer: Thanks for your valuable comments! We have double checked and made revision in our manuscript, here we refer to N+P.

• L39: please define ‘virtual’ here.

Answer: We have revised the writing and added the explanation of “virtual” in the Abstract. The details are as follows: “The flow of real nutrients (physical nutrients) and total nutrients required for production processes (virtual nutrients) has different impacts on resources and environment, but the effects have not been simultaneously quantified and analyzed.”

This study quantified the physical and virtual N, P flows embedded in the global telecoupled agricultural trade networks among 221 countries or regions from 1997 to 2016 and elaborated other components of the telecoupling framework.”.

• L41: I can’t understand that sentence ‘more distant countries or regions tend to produce more N/P flows’.

Answer: Apologies for the unclear sentence. We have revised the sentence as follows: “Countries or regions with longer trade distance tend to produce more N and P flows.”. We also double checked this problem in other section and revised the relevant parts.

• Figure 1: where does the panel a about agricultural trade network represent? Where do the data come from? Panel b: are you sure there is not a confusion between importing and exporting countries? Panel c: I can’t understand the difference between ‘actual’ and ‘virtual’ flows here. And does ‘actual’ mean ‘physical’?

Answer: Thanks for your suggestion. Data used in Figure 1 are from the FAO global trade matrix. We have simplified the flow in Panel a and have selected the top 100 trade routes.

For Panel b and Panel c: Thank you for spotting the error. Sending systems refers to export countries or regions and the receiving systems refers to importing countries or regions, so we have revised and redrawn the Figure. The ‘actual’ in Panel C should be ‘physical’. Please see the new Figure below:

Fig. 1. Global nitrogen (N) and phosphorus (P) flows under the telecoupling framework. a. global agricultural trade networks display the top 100 bilateral agricultural trade routes (the thickness of the trade vector represents the volume of the trade flow); **b.** global agricultural trade in the telecoupling framework (systems and effects in the telecoupling framework are illustrated in Supplementary Material Fig. S1); and **c.** sources of physical and virtual N and P.

• L82-83 (and line 84): I disagree. Not all nutrients that are supplied to soils in excess compared to crop uptake are lost to the environment, in particular for P. Some phosphorus can be stored in soils as legacy P and can be used for future crop uptake.

Answer: Thanks for your thoughts! We agree with your opinion. In agricultural systems, some surplus nutrients may stay in the soil and may be used by further crops. We think that these residual nutrients may also cause environmental pollution risks through runoff processes or other pathways. Here we only aimed to determine the risks caused by surplus nutrients. The relevant section has been revised as follows: “Some surplus phosphorus can be stored in soils as legacy P and can be used for future crop uptake. However, a fraction of surplus nutrients is lost to the environment, probably lowering the air and water quality. Therefore, the nutrient flows reflect the transfer of N and P inputs and losses risks during the production life cycle.”.

• L83: please define what ‘virtual nutrient flows’ are in your study.

Answer: We have revised the definition in the Methods section. Please see details as follows: “While physical nutrients are considered as nutrients contained in traded products, virtual nutrients are the total inputs in the production of agricultural products”.

• L88-89: what do you mean by ‘interactions of the global flows of virtual N’?

Answer: Thanks for your valuable comments and sorry for an incomplete expression. We

have revised the sentence as follows: “Xu et al. assessed the evolution of global flows of virtual N and interactions between virtual water, energy, land, and other different flows through international trade networks during 1995-2008, and they distinguished the flows between adjacent versus distant countries.”

• L98: what does ‘differentiate’ mean here?

Answer: Sorry for the typo. We have revised the sentence as follows: “clarify the spatial-temporal dynamic characteristics of physical and virtual flows of N and P in global telecoupled agricultural trade networks”.

• L107 and throughout the manuscript: please use SI units, not 109 kg P or similar inappropriate units.

Answer: Thanks for your suggestion. We have checked the manuscript and have made changes accordingly. For example, we have changed “kg” to “Tg” in Line 107, Page 5.

• L108: the physical P trade flow is estimated to be 2.2 Tg P/yr in 2016 this study. This is quite lower than the estimated 3.0 Tg P/yr in 2011 by 3. Please provide some explanation for this lower value.

Answer: In the last version, only 42 types of products were considered in our manuscript, whereas 397 were considered in Ref.3, therefore our result was less than 3.0Tg/yr. We have now expanded the analysis to include with 320 types of products in our research. The result is now 3.5 Tg P/yr in 2016. We have revised the Methods and Results sections and provided the list of traded products in the Supplementary Information.

The revised details are as follows: “A total of 320 major agricultural products were considered in this study, including 17 major grain crops (e.g., barley, corn, rice, soybean, and wheat), processed products obtained from the aforementioned 17 crops (e.g., flour, bran, soybean oil), five types of livestock products (pork, beef, poultry, sheep and goat), and other processed products”. “The total physical nutrient flows of N and P increased from 10.3 Tg to 27.1 Tg and from 1.4 Tg to 3.5 Tg, respectively, between 1997 and 2016 (Fig. 2a, b)”.

• L111-112: I can’t understand that sentence.

Answer: We have revised the sentence to make it more readable. Please see details as follows: “There are 48 countries saw a more than 10-fold increase in both physical N and P exports over the period.”.

• L112-113: why focussing on those countries? Those have a rather second role in agricultural trade, making their import or export data sensitive to single changes in trade in some specific years. Those

countries also suffer from poor reporting of their trade data in international databases such as FAOStat. I recommend selecting countries with more solid statistical data and/or greater contribution to trade.

Answer: Thanks for your comments! We have now revised the manuscript and focused on countries that contribute greater.

• L117-118: do this refer to crop commodities (instead of 'agricultural commodities')? Moreover, I don't understand how the 97.9% has been calculated.

Answer: Thank you very much for your comment. Sorry for the typo. We meant the proportion of crops products to total agricultural products. We have revised this paragraph: "The physical nutrient flows mainly involved trade of crop products and their processed products, accounting for about 90% of total physical flows in 2016 and far exceeding the flows in the trade of animal products. "

• L122: does 'nutrient' refer to N or P??

Answer: Thanks for your comments! Sorry for the misleading expression. We have revised the sentences, please see details below: "The trade of crops and their processed products contributed to two-thirds of total virtual nutrient (both N and P) flows in 2016, while the trade of animal products accounted for the remaining one-third."

• L122-125 and Figure 2: I am really surprised by the massive contribution of cattle to virtual nutrient flows. Can you provide more explanation about this?

Answer: Thanks for your comments! We have added some discussion. Please see details below: "The virtual nitrogen of beef mainly comes from feed, excrement and other materials input during production period. Although the virtual nitrogen directly produced in the feeding process is less, the virtual nitrogen input in the feed planting stage is higher, resulting in the virtual nitrogen content of beef and its processed products being significantly higher than that of other plant products. Adjusting the sources of animal feed and improving dietary consumption by replacing beef with other food can effectively reduce virtual nitrogen."

• L139-142: those lines can be compressed.

Answer: We have revised the sentences as follows: "The results showed that the number of net sending systems decreased from 81 to 57 during 1997 to 2016. By contrast, the net receiving systems increased from 140 in 1997 to 164 in 2016."

• L154: does 'sending' mean 'exporting'? If yes, please select one term and use it consistently throughout the manuscript.

Answer: Thank you very much for your comment. We have checked the whole manuscript

and revised the relevant sentences.

• L156: what does ‘total nutrient flows’ mean? Is this the sum of imported N (or P) flows?

Answer: Sorry for the misleading expression. Here it means the percentage of N and P respectively. We have modified our expression. Please see details as follows: “China was the largest net receiving system of physical nutrients with 6.06 Tg of N and 0.62 Tg of P during 2016, accounting for 20.0% and 16.0% of the global total physical N and P flows, respectively.”

• L155-161: the date provided in the text do not match the number provided in Fig4. Please clarify!

Answer: Thank you very much for your valuable comment. We have redrawn Fig.4 after recalculating the result. The new figure is as follows:

Fig. 4. Top 5 net nutrient receiving systems and sending systems in 2016. a. Top 5 net receiving systems and sending systems of nitrogen (N); b. Top 5 net receiving systems and sending systems of phosphorus (P). Countries are ordered by the net volume of physical N or P they imported or exported. The green and red labels indicate the volume of net import and export of physical nutrients, which is the length of the green column minus the length of the red column. The yellow and blue labels represent volume of net import and export of virtual nutrients, which is the length of the yellow column minus the length of the blue column.

•Fig4: this is a nice and important figure but many aspects have to be improved. First, clarify if ‘inflow’ and ‘outflow’ at the extreme right of the figure are synonyms of ‘import’ and ‘export’.

Second, please clarify why green bars (representing imports) may exist below the X axis for some specific exporting countries (eg, France) and not above the X axis. Third, please clarify what the ‘ratio of domestic N fertiliser use’ means and how it has been calculated. Fourth, please explain and justify why the virtual flows have been plotted for export only (and not for import of virtual flows).

Answer: Thanks for your suggestions and comments! We have revised Figure 4 to make it clearer for readers. First, we use “import” and “export” instead of inflow and outflow to avoid the misleading expression. Second, we set the origin point of green bar at the end point of red bars. We think this would be easier for readers to compare the difference between each country’s total import and export. Third, this ratio is calculated by the export of virtual N and total application of N-fertilizer -and here we can discuss the risks of N loss through agricultural production export. Fourth, we have added a new bar to indicate the imports of virtual flows as you suggested.

Fig. 4. Top 5 net nutrient receiving systems and sending systems in 2016. a. Top 5 net receiving systems and sending systems of nitrogen (N); b. Top 5 net receiving systems and sending systems of phosphorus (P). Countries are ordered by the net volume of physical N or P they imported or exported. The green and red labels indicate the volume of net import and export of physical nutrients, which is the length of the green column minus the length of the red column. The yellow and blue labels represent volume of net import and export of virtual nutrients, which is the length of the yellow column minus the length of the blue column.

• L170: do you refer to physical or virtual flows?

Answer: Thanks for your question. Nutrient flows here include both physical and virtual nutrients flows. Here we analyzed the spatial distribution characteristics of the two. We have revised the section title.

• L171: please replace complicated by complex.

Answer: We have replaced it.

• L173-174: please replace 'more than' by 'greater than'.

Answer: Thanks! We have replaced it.

• L179: I can't understand that sentence 'a total of 45.5%...'

Answer: Thanks! We have revised the sentence and corrected the grammar. Please see details below: "As for soybean's physical N flows, 40.1% of China's imported physical N was from the United States, it also counted about 20.2% of the United States 's total export."

• Figure S5 is interesting but it is quite contradictory with Table 1. Fig S5 shows that flows from adjacent countries contribute more to global N or P physical flows than flows from non-adjacent countries. This is in contrast with Table 1 that shows that major physical flows occur overseas, between countries from different continents. Please clarify.

Answer: Thanks for your comment! Table 1 in the main text shows the biggest route. The supplementary material Figure S5.a shows the average volume of nutrient flow between non-adjacent systems and adjacent systems. The average volume from adjacent countries is greater but the total is not, because there aren't a large number of trade routes from adjacent countries. On the contrary, we can find that large trade flows tend to occur in trans-continental trade routes.

• L190-191: which nutrient do you refer?

Answer: Sorry for the misleading expression, we have revised the sentences. Here we display both N and P results. Please see details as follows: "Fig. 5 shows the spatial distribution of physical and virtual nutrient (N and P) flows associated with global agricultural trades among major country groups by geography in 2016 (see Fig.S6 for N, P flow patterns in 2000, 2005, 2010, and 2015)."

• L194 'in Europe': do you mean 'to' or 'from' Europe? And later in the same sentence: what does 'total global flows' mean?

Answer: Thanks for your comments! We have made them clearer. "In addition, the volume of physical nutrient export from Europe was estimated as 28% of the total global flows, but most of the flows occurred among European countries themselves."

• Figure 5: please provide numbers and units! Also, please provide differentiation between inflows and outflows to and from each world region.

Answer: Thanks for your comments! We have revised Figure 5 and the new version is shown as follows:

Fig. 5. The nitrogen (N) and phosphorus (P) flow patterns through the trade of agricultural products in 2016 (Gg/y). a. physical N flow; b. physical P flow; c. virtual N flow; and d. virtual P flow. The link colors correspond to the sending systems. The arrow points to the receiving systems. According to the FAO, all the countries were divided into 10 country groups by geography, i.e., East Asia (E.AS), South-East Asia (S-E.AS), Central and West Asia (C&W.AS), North America (NA), Central America (C.A), South America (SA), Oceania (OA), Africa (AF), East Europe (E.EU), West Europe (W.EU).

• L212: please replace ‘exported’ by ‘contributed’.

Answer: Thanks for your suggestion! We have revised the sentence as follows: “Nutrient flows in global agricultural trade networks exhibited a clear uneven structure. Overall, only 10%

of systems contributed about 90% of total exported nutrients during 2016.”.

- L212-213: this sentence is quite speculative. Please provide solid justification and references.

Answer: We agree with you. The sentence was unsubstantiated, so we have removed it.

- L218: I am not sure that the environmental damages that are detailed here are due to the nutrient-rich nature of croplands. Please reformulate more specifically.

Answer: We have revised the sentence to make it clearer. “However, to fulfill such intensive production, sending systems must constantly maintain and develop new croplands, causing damage to forests, grasslands, and other natural ecosystems.”

- L224: what does ‘demand in Brazil’s soybean exports’ mean?

Answer: Sorry for the unclear sentence. We have revised the sentence: “Another reason why Brazil's soybean exports increased is the change of U.S.-China soybean trade flow: China raised tariffs on soybeans imported from the U.S., which had a spillover effect on the China-Brazil soybean trade, and caused the land-use changes in Brazil.”

- L225: can you detail the years or period for the China-US trade war? In my understanding, this trade war is posterior to the study period ending in 2016.

Answer: Yes, we mean the China- U.S. trade disputes. We have deleted the “Trade war”.

- L231: what does ‘total volume of participation in the regional P recycling system’ mean?

Answer: The point we want to make is that sustained export over a long period of time changes the nutrient cycle (through food, recycling, etc.) of the original area. We have revised the relevant sentence. “The long-term continuous export of physical P changes the regional P recycling pattern, resulting in the constant loss of P from the original soil and the continuous decrease of domestic P resource availability”.

- L234: please provide the data source.

Answer: Thanks, we cited the data source and revised the sentence. “For instance, due to agricultural exports in 2010, the total volume of physical P flowing out of Argentina reached 0.21 Tg. As a net sending system, only 34% of the total nutrient uptake was recycled back to farmland, and the fertile soft land of the Pampas region has faced P depletion risks³³.”

- L258-259: I think I agree with this idea but it has to be developed. In fact, the nutrients imported through trade do not reach agricultural soils directly. They may ultimately reach agricultural soils if those traded products are used for feed or food use and if animal manure and human excreta are recycled back to agricultural soils.

Answer: Yes, thanks for your comments! We have revised this sentence to discuss the impact of nutrient flows. “For countries or regions with deficient soil nutrients, the inflow of physical nutrients through import trade can supplement nutrient reserves and might finally alleviate soil fertility deficiencies through nutrient cycling, such as the return of manure and the application of organic fertilizer.”

• L261-263: unclear sentences.

Answer: We have revised the sentence to make it clearer. “Africa imported 0.12 Tg physical P through agricultural trade in 2016, which significantly alleviated the P deficit in Africa”.

• L263-265: this is a repetition of lines 241-243.

Answer: Thanks for your comment, we have revised the section as follows. “The import of virtual nutrients means the conservation of domestic nutrient resources, especially P (because of the difficulty of P to regenerate).”

• L268-269: I can’t understand that sentence.

Answer: Thanks for your comments! We have revised the sentence to “Importing virtual nutrients also represents the transfer of the environmental burden from the receiving systems to the sending systems”.

• L283-295: these new results should not be within the Discussion but should instead be moved to the Result section. Also, I’m not sure using ‘waste’ of nutrient is appropriate. Please consider using ‘loss’ or something similar instead.

Answer: Thanks for your suggestion! We agree with your suggestion. We have divided this paragraph into two parts, to display and explain the results and methods respectively. Also, we have checked the whole manuscript and revised and replaced the words. For example, “Although the global trade of agricultural products presented an overall positive telecoupling effect on saving N and P resources, some inefficient flows resulted in unnecessary loss.”.

.

• Fig6: please explain how products were ranked.

Answer: We sorted the products by FAO item code from largest to smallest.

• L329-335: this is quite vague. Please improve.

Answer: Thanks for your comments! We have re-written the paragraph.

• L338-359: those lines could be compressed.

Answer: Thanks for your comments! We have revised these sentences.

• L381: although I'm wrong, I could not find this Table S3.

Answer: Yes, because the Table S3 is a very large table, we have uploaded the Excel file (Table S3) separately. Please check it in the attachment.

• L387: are you sure this is 'inflow' and not 'outflow'?

Answer: Sorry for the typo. The outflow is right. We have revised the word. "The final net import or export of each country or region was obtained by subtracting the total nutrient inflow associated with import trade from the total nutrient outflow of export trade."

• L398: I can't understand what you mean about 'the *l*th trade route of the product *i*' Moreover, I can't see any *l* index in the equation at the right of the sigma.

Answer: We have re-written the paragraph and revised the formula as follows:

Sending-receiving effects

Trade leads to the sending-receiving effects of redistributing global nutrient resource utilizations. As for physical nutrients, the sending-receiving effect is actual physical nutrients transferred from A to B, the specific calculation formula is:

$$NSRE_{p,i} = \sum_{l=1}^m N_{p,i} \times (T_{i,A,B} - T_{i,B,C_k}) \quad (2)$$

$$PSRE_{p,i} = \sum_{l=1}^m P_{p,i} \times (T_{i,A,B} - T_{i,B,C_k}) \quad (3)$$

where, A and B are the sending and receiving systems, respectively, in the *l*th trade route or nutrients flow of the product *i*; $T_{i,A,B}$ is the trade volume of product *i* exported from A to B, t; T_{i,B,C_k} is the trade volume of product *i* re-exported from B to C_k , t; $N_{p,i}$ and $P_{p,i}$ is the N and P content of product *i*; $NSRE_{p,i}$ and $PSRE_{p,i}$ are the physical sending-receiving effects of the product *i*; *m* is the total number of trade routes.

This study explored whether the sending-receiving effects on global N, P utilizations were positive (nutrient saving) or negative (nutrient wasting). The specific calculation method is:

$$NSRE_{v,i} = \sum_{l=1}^n (N_{v,i,B} - N_{v,i,A}) \times (T_{i,A,B} - T_{i,B,C_k}) \quad (4)$$

$$PSRE_{v,i} = \sum_{l=1}^n (P_{v,i,B} - P_{v,i,A}) \times (T_{i,A,B} - T_{i,B,C_k}) \quad (5)$$

where, $N_{v,i,A}$, $N_{v,i,B}$, $P_{v,i,A}$, $P_{v,i,B}$ are the virtual N, P contents of local product *i* in systems A and B, kg/t; $NSRE_{v,i}$ and $PSRE_{v,i}$ are the virtual sending-receiving effects of the product *i*. The sending-receiving effects of nutrient flows were obtained by the flows of all agricultural trade products considered in this study. If $NSRE_{v,i}$ or $PSRE_{v,i} > 0$, it indicates that the redistribution of agricultural product trades on the utilization of nutrient resources present a positive sending-receiving effect of saving N, P; if $NSRE_{v,i}$ or $PSRE_{v,i} < 0$, it indicates a negative sending-receiving effect on the utilization of nutrient resources of wasting N, P; *n* is the total number of nutrients flows.

- L403: please clarify ‘the redistribution of ag product trades on the utilisation....’.

Answer: Thanks for your comments! We have revised the sentence below: “it indicates that the nutrient redistribution through agricultural product trades presents a positive telecoupling effect of saving N and P during the utilization of nutrient resources globally”.

- L416: 60% of global caloric consumption is quite low. This means that the list of products you considered is in fact quite limited. I recommend that you add more products in your list. Also, please provide the corresponding estimate of crop production represented by this list of crop species you considered.

Answer: Thanks for your suggestions! We have re-done our calculation and analysis considering 320 kinds of agricultural productions (95% or more calories). We have provided the list of agricultural productions in Supplementary Materials.

- Supp Section S1.2: please express your data in kg/ha not kg/hm². Also, please provide precise data sources, in particular for fertiliser and manure application rate by crop and by country.

Answer: Thank you for your suggestion. We have made a replacement and explained the source of all the data.

- Supp L65: please double check this is ‘organic’ fertiliser.

Answer: Apologies for the typo. We have replaced with fertilizer.

- Supp L98: is this footprint or input?

Answer: This refers to input. We've made the revision.

- Suppl L71-75: this is confused. Please make this more explicit.

Answer: Thanks for your suggestion. We have rewritten our approach.

- Suppl L106-111: again, this is really confused. Please make this more explicit.

Answer: Thank you for your comments. We have improved our expression.

- Suppl L116: what does ‘or obtains a variety of products’ mean??

Answer: Here we refer to the processing of primary products into a variety of processed products. We have deleted this line.

- Suppl L121-126: I can’t understand why market values were used instead of more classical weighting by product biomass or caloric content. Also, I could not understand the difference between r and c. Finally, how do you account for raw products that lead to two different products that are considered in your study (e.g., soybean grains lead to soy oil and soy cakes)?

Answer: Thanks for your suggestion. We have replaced the conversion coefficient with the kcal ratio coefficient. See details in supplementary Section S1.3 as follows:

“S1.3 Calculation of virtual nutrient contents of processed products

We use factors based on caloric equivalents, according to the method by Kastner³ to transform the virtual nutrient contents of primary crop or animal products into that of processed products. The calculation formula is as follows:

$$N_{i,p}=K_i \times N_i \quad (5)$$

$$P_{i,p}=K_i \times P_i \quad (6)$$

where $N_{i,p}$, $P_{i,p}$ is virtual N or P content of the processed product, kg/t; N_i , P_i is virtual N or P content of the primary product, kg/t; K_i is kcal ratio between the processed product and the primary product.

The kcal contents of all products are provided in Table S1, according to FAO standard factors on nutritive values (<https://www.fao.org/economic/the-statistics-division-ess/publications-studies/publications/nutritive-factors/en/>).”

- Suppl L130: how did you account for supplementary P feed?

Answer: Thanks for your comment. We have added the calculation method of P feed in the Supplementary Information Section S1.2.2 as follows:

“S1.2.2 Virtual nutrient calculation of primary livestock products

The virtual nutrients of primary animal products mainly come from feed inputs. We used the feed conversion coefficient to convert virtual nutrients of animals into virtual nutrients of feed crops. The specific formula is as follows:

$$N_i=C_i \times F_i \times N_{feed} \quad (3)$$

$$P_i=C_i \times F_i \times P_{feed} \quad (4)$$

where N_i , P_i is virtual N or P content of the animal product, kg/t; C_i is feed conversion coefficient of the animal, kg/kg, retrieved from Mekonnen & Hoekstra¹; F_i is fraction of crop-based feed in total feed, retrieved from Mekonnen & Hoekstra¹; N_{feed} , P_{feed} is virtual N or P content of crop-based feed of the animal product, kg/t, crop-based feed were partitioned among seven main grain feed commodities - barley, maize, peas, rapeseed, sorghum, soybean, and wheat—according to data retrieved from Herrero et al.² All the data for feed conversion coefficients, fractions, and compositions of grains in crop-based feed by animal species and by region are provided in Table S2-S4.”

- Suppl L156-164: this is quite a vague discussion. Either improve or omit.

Thanks for your suggestion. We have deleted these lines.

References

1. MacDonald, G. K., Bennett, E. M. & Carpenter, S. R. Embodied phosphorus and the global connections of United States agriculture. *Environ. Res. Lett.* 7, 44024 (2012).
2. Nesme, T., Roques, S., Metson, G. S. & Bennett, E. M. The surprisingly small but increasing role of international agricultural trade on the European Union's dependence on mineral phosphorus fertiliser. *Environ. Res. Lett.* 11, 025003 (2016).
3. Nesme, T., Metson, G. S. & Bennett, E. M. Global phosphorus flows through agricultural trade. *Glob. Environ. Chang.* 50, (2018).

Answer: Thanks for your suggestions! We have studied and cited these relevant articles and enhanced our method in our manuscript. As some countries play roles as “transfer stations” in the trade networks, we use the data provided by Kastner to calculate re-export, which can be matched with the N and P input coefficient of producer country. The virtual N and P calculated by this method is the real nutrient transfer globally. At the same time, we redefine the spillover system and effect, and carry out quantitative analysis:

“Agricultural trade in the context of the telecoupling framework. Agricultural trade networks are subsets of global trade networks, which have typical network characteristics (Fig. 1a)⁴⁴. A total of 221 countries or regions and 27819 different export-import routes were involved in global agricultural trade networks in 2016⁴⁵. The international agricultural trade networks were viewed through the telecoupling framework⁸ that facilitates more comprehensive and systematic understanding than conventional trade research. In this study, the conceptual model of agricultural trade mainly focused on physical and virtual nutrient flows among different countries or regions. In telecoupled agricultural trade networks, exporting and importing countries (or regions) are defined as sending and receiving systems, respectively; and the countries (or regions) that are affected by the trade between sending and receiving systems are defined as "spillover systems" ^{8,13}. A global trade network is an aggregation of thousands of trade routes. For a specific trade route (telecoupled subsystem), importer or exporter countries not only represent sending and receiving systems but also play roles as “re-export transit stations”. For example, Singapore, which imports rice from India and exports it to Australia, is not directly involved in its own rice production and consumption. The diagram of spillover systems in the telecoupling framework is shown in Supplementary Materials Fig. S1. It is worth noting that the roles of sending, receiving and spillover systems in the trade network are not fixed. In other words, a country may have multiple roles in global trade network.

Along with the flow of agricultural products, physical and virtual nutrients move from sending to receiving systems. The socioeconomic and environmental effects are produced by the interactions between sending and receiving systems through trade and the accompanying nutrient flows. Each of the nutrient flows contributes to the changes in global allocation of

nutrient resources¹⁸, which in-turn has telecoupling effects on nutrient cycling and utilization on a global scale^{32,46}. The systems, flows, and effects are also interconnected with two other components of the telecoupling framework: causes (reasons behind the flows) and agents (decision-making entities such as traders that facilitate the flows), but causes and agents are beyond the scope of this paper.’

Reviewer #2 (Remarks to the Author):

In the present manuscript, the authors present a global quantification and analysis of nitrogen (N) and phosphorus (P) physical (embedded) and virtual flows through commercial exchanges between 188 countries for the year 1997 to 2016 and 42 agricultural commodities. Furthermore, the paper aims at analyses the natural resource use and environmental impacts of N and P flows on the sending (exporting) and receiving (importing) systems and on global nutrient cycles, as well as to present some policy suggestions for a more sustainable use of nutrient resources. Despite some studies about physical and virtual nutrients resources in trade have been recently published, no study have presented a simultaneous estimation of such different flows. Therefore, the authors present here an original analysis that well complements the existing literature on this topic. A priori, The topic and questions raised fit for a publication in Nature Communication. Despite the overall interest of the manuscript, I have a few major concerns that I would like to raise.

Answer: Thanks a lot for your comments and support of the innovation of this article! We agree with you and have revised the manuscript according to your valuable suggestions. The point-by-point responses and revised details are as follows.

My first major comment is referred to the methodology to calculate virtual N and P flows. From the detailed methods section in you Supplementary Information, I cannot understand how the virtual flows for each of the 42 agricultural commodities was calculated, in particular for the virtual inputs referred to chemical fertilizers. According to lines 63-64, the authors calculated the “total nitrogen input for crops”. To do so, they considered, for each country, the total amount of different N-based synthetic fertilizers divided by the total yield of all grain crops – all data derived mainly from FAOSTAT – . This is understandable, since FAOSTAT does not report any specific fertilization rate for each crop species. Nevertheless, most results are then presented for a set of crop commodities (e.g. Figure 2). It is not clear how the authors calculated virtual N and P resources for each crop commodity based on the methodological description. This is a major problems that requires a detailed explanation, since it could impact most of the manuscript results. Note also that in most papers dealing with virtual nutrient resources, only synthetic fertilizers are accounted for. Here the authors considers all nutrients inputs to soils; I think that this choice should be justified. Furthermore, I am not totally convinced by the accuracy of the calculation process used to transform processed commodities to primary commodities (S.1.2.2 and equation 8 in the SI).

Answer: We adopted a new method for the calculation of virtual nutrients. The virtual nutrients of different crops can be effectively distinguished. The details of the method have been added in supplementary documents. Also please see the detailed method below:

In this study, gridded crop-specific fertilizer, and manure application data at 0.083 degrees downloaded from EarthStat were used to extract nitrogen (N) and phosphorus (P) application

rate for 17 major crops in 2000. These 17 major crops were wheat, maize, rice, barley, millet, sorghum, soybean, sunflower, potato, cassava, sugarcane, sugar beet, oil palm fruit, rapeseed, groundnut, seed cotton and rye. For the calculation of N and P application rate for the other 85 minor crops, as direct data on N and P application rate are not available, crop-specific yield, N and P content, crop-specific harvest area fractional and N and P balance which responded to excess or deficiency of nitrogen and phosphorus compared to the current level were used for the calculation. The content data are from FAO and <https://www.gov.uk/government/publications/composition-of-foods-integrated-dataset-cofid>. Crop-specific yields and other related data are sourced from <http://www.earthstat.org/>.

The N and P application rate of 102 different crops in different countries in 2000 was finally obtained. The results obtained by this calculation method are in agreement with the actual N and P application rate data. Then, the actual N and P application rate and the calculated N and P application rate were linearly scaled from 2000 to 2016 over all countries to ensure that the total N and P application rate of 102 crops was consistent with the actual N and P application rate.

It should be noted that N and P contents used in this study were based on the weight of the fruiting part rather than the whole crop-specific yield, which could be uncertain for global-scale calculation. In addition, the N balance deficit of some countries was too large, resulting in negative N and P application rates. In this study, this paper treated it as an outlier and normalized to 0.

The authors use a price ration (i.e. the ratio of the market value of the product to the aggregate market value of all products) to transform processed to primary products. Whether I think to understand the reasoning behind this choice, I am skeptical whether the choice of this coefficient is actually compelling. I guess that this has been done to avoid any double counting. That is that, for example, to produce 1 kg of soybean oils has required a certain amount of soybeans. But such soybeans has also produced some oilseed-cake. Thus, the transformation of these two sub-products in the original primary product should avoid any double counting. This is totally correct, but most studies uses a coefficient based on the kcal ration between the sub-product and the primary product to transform processed commodities into primary commodities (for example, McDonald et al. (2015) Rethinking agricultural trade relationships in an era of globalization). I am thus not sure that using a price index leads to similar results. In addition, the source of such ratios – reported in Table S2 – (and of commodities' prices) is not mentioned.

Answer: Many thanks for your helpful comments. We have made improvements in response to your concerns. In terms of processing products converting into primary products, we

substituted the kacl ratio coef proposed by Kastner et al. 2014. which can effectively avoid double counting. We have listed all the coefficients for all products in the Supplementary Information Data1.

My second major comment concerns a general confusion between the results reporting over physical and virtual N/P flows. I would invite the authors to clearly distinguish whether the results and discussions are referred either to physical or virtual resources, since this is often not so clear. Especially, all results about net flows seems to reader to physical nutrients flows; if it is the case this should be clearly defined. A great plus of the present work, as mentioned in the introduction, is to simultaneously report over both physical and virtual flows. To better stress this innovative point, I think that the authors could expand their discussion bringing a few information about the relation between virtual and physical flows. For example, does countries catheterized by high physical exports shows above or below average virtual nutrients flows?

Answer: Thank you very much for the helpful advice! We have revised the manuscript to explicitly mention whether each result or discussion section refers to physical or virtual nutrients and added analysis of the comparison between physical or virtual nutrients to better demonstrate the innovation of our research.

The revised section is as follows:

“Spatial distribution and variation of net receiving and sending systems. The results showed that the largest net receiving systems of nutrients were mainly within Asia, while the net sending systems were mainly in North America and South America. The results showed that the number of net sending systems decreased from 99 to 62 during 1997 to 2016. By contrast, the net receiving systems increased from 122 in 1997 to 159 in 2016 (Fig. 3). The top 10 net sending systems contributed to about 70% of the total exported physical nutrients, and the top 20 net sending systems contributed to more than 90% of the total exported physical nutrients.

Fig. 3. Nitrogen (N) and phosphorus (P) receiving and sending systems in 1997(a-c, g-f) and 2016(d-f, j-l). N flows through gross export (a, d), gross import (b, e) and net budget (c, f); P flows through gross export (g, j), gross import (h, k) and net budget (i, l). Net nutrient flows were calculated as the difference between gross physical N (or P) imported and gross physical N (or P) exported. The red and blue colors indicate exporting and importing, respectively; gray color means no available data.

Fig. 4 shows the top 5 net receiving and sending systems of nutrients during 2016. China was the largest net receiving system of physical nutrients with 6.06 Tg of N and 0.62 Tg of P during 2016, accounting for 20.0% and 16.0% of the global total physical N and P flows, respectively. Thereafter, Japan (0.95 Tg of N and 0.13 Tg of P) and Mexico (0.78 Tg of N and 0.23 Tg of P) were the second and third ranked receiving systems. The top 5 net receiving systems imported 33% of the total physical nutrient flows. On the other hand, the United States (5.20 Tg of N and 0.55 Tg of P) and Brazil (4.21 Tg of N and 0.39 Tg of P) generated the largest net exports during 2016. The total exported physical N and P by the United States and Brazil were 30.0% and 23.1% of the total global physical N and P flows, respectively.

Fig. 4. Top 5 net nutrient receiving systems and sending systems in 2016. a. Top 5 net receiving systems and sending systems of nitrogen (N); b. Top 5 net receiving systems and sending systems of phosphorus (P). Countries are ordered by the net volume of physical N or P they imported or exported. The green and red labels indicate the volume of net import and export of physical nutrients, which is the length of the green column minus the length of the red column. The yellow and blue labels represent volume of net import and export of virtual nutrients, which is the length of the yellow column minus the length of the blue column.”

Finally, concerning the three objectives mentioned at line 101-103, I hardly found any discussion regarding policy suggestions. If this objective is maintained, I think that the authors has a bit of room for improving their discussion to these terms.

Thanks for your advice. We have added policy suggestions based on the objective we proposed in the introduction.

In addition to such major comments, here a series of minor comments that may help the author's in revising the paper.

Answer: Many thanks! Your comments and suggestions helped us improve our manuscript a lot.

L37: I think that the correct English term should be “N/P flows embedded” and not embodied.

Furthermore, at the same line, to what the “other components” refers?

Answer: Thank you very much for your comments. We have double checked and made corrections in our whole manuscript. In addition, “other components” here refers to other elements of the framework of telecoupling, including agents, causes, and effects.

Figure 1: panel a: flows are difficult to distinguish. Maybe filter on the most important flows? Panel b: I find this representation not totally clear: what are the causes and effects shown here? Does it refers to environmental effects? How is a spillover system defined?

Answer: Thanks for your kind comments! We made a revision as follows:

Figure 1 panel a: We have re-drawn the figure and made it clearer.

Fig. 1. Global nitrogen (N) and phosphorus (P) flows under the telecoupling framework. a. global agricultural trade networks display the top 100 bilateral agricultural trade routes (the thickness of the trade vector represents the volume of the trade flow); b. global agricultural trade in the telecoupling framework (systems and effects in the telecoupling framework are illustrated in Supplementary Material Fig. S1); and c. sources of physical and virtual N and P.

Panel b: The international agricultural trade networks were viewed through the telecoupling framework⁸ that facilitates more comprehensive and systematic understanding than conventional trade research. In this study, the conceptual model of agricultural trade mainly focused on physical and virtual nutrient flows among different countries or regions. In telecoupled agricultural trade networks, exporting and importing countries (or regions) are defined as sending and receiving systems, respectively; and the countries (or regions) that are affected by trade but not directly involved in trade are defined as "spillover systems" (Fig. 1b)^{8,13}.

As for telecoupling effects in telecoupling framework, we have clarified the quantification method of sending-receiving effect, spillover effects and telecoupling effects. The revised section is as follows:

“Agricultural trade in the context of the telecoupling framework. Agricultural trade networks are subsets of global trade networks, which have typical network characteristics (Fig. 1a)⁴⁴. A total of 221 countries or regions and 27819 different export-import routes were involved in global agricultural trade networks in 2016⁴⁵. The international agricultural trade networks were viewed through the telecoupling framework⁸ that facilitates more comprehensive and systematic understanding than conventional trade research. In this study, the conceptual model of agricultural trade mainly focused on physical and virtual nutrient flows among different countries or regions. In telecoupled agricultural trade networks, exporting and importing countries (or regions) are defined as sending and receiving systems, respectively; and the countries (or regions) that are affected by trade but not directly involved in trade are defined as "spillover systems"^{8,13}. Global trade network is an aggregation of thousands of trade routes. For a specific trade route (telecoupling subsystem), importer or exporter countries not only represent sending and receiving systems but also play roles as “re-export transit stations”, for example, Singapore, which imports rice from India and exports it to Australia, is not directly involved in its own rice production and consumption. Therefore, we define the countries in the telecoupling subsystem excluding the sending system and receiving system as spillover systems. The diagram of spillover systems in telecoupling framework is shown in Supplementary Materials Fig. S1. It is worth noting that the roles of sending, receiving and spillover systems in the trade network are not fixed, these three are only clear in each specific trade route; in other words, a country may have multiple roles in global trade network.”

Fig. S1 Systems and effects within an agricultural trade network under the telecoupling framework. a. Schematic diagram of global agricultural trade network; b. A specific trade route shows sending system, receiving system, and spillover system as well as sending-receiving effect, spillover effect, and telecoupling effect.

Sending-receiving effects

Trade leads to the sending-receiving effects of redistributing global nutrient resource utilizations. As for physical nutrients, the sending-receiving effect is actual physical nutrients transferred from A to B, the specific calculation formula is:

$$NSRE_{p,i} = \sum_{l=1}^m N_{p,i} \times (T_{i,A,B} - T_{i,B,C_k}) \quad (2)$$

$$PSRE_{p,i} = \sum_{l=1}^m P_{p,i} \times (T_{i,A,B} - T_{i,B,C_k}) \quad (3)$$

where, A and B are the sending and receiving systems, respectively, in the l th trade route or nutrients flow of the product i ; $T_{i,A,B}$ is the trade volume of product i exported from A to B, t ; T_{i,B,C_k} is the trade volume of product i re-exported from B to C_k , t ; $N_{p,i}$ and $P_{p,i}$ is the N, P content of product i ; $NSRE_{p,i}$ and $PSRE_{p,i}$ are the physical sending-receiving effects of the product i ; m is the total number of trade routes.

This study explored whether the sending-receiving effects on global N, P utilizations were positive (nutrient saving) or negative (nutrient wasting). The specific calculation method is:

$$NSRE_{v,i} = \sum_{l=1}^n (N_{v,i,B} - N_{v,i,A}) \times (T_{i,A,B} - T_{i,B,C_k}) \quad (4)$$

$$PSRE_{v,i} = \sum_{l=1}^n (P_{v,i,B} - P_{v,i,A}) \times (T_{i,A,B} - T_{i,B,C_k}) \quad (5)$$

where, $N_{v,i,A}$, $N_{v,i,B}$, $P_{v,i,A}$, $P_{v,i,B}$ are the virtual N, P contents of local product i in systems A and B, kg/t; $NSRE_{v,i}$ and $PSRE_{v,i}$ are the virtual sending-receiving effects of the product i . The sending-receiving effects of nutrient flows were obtained by the flows of all agricultural trade products considered in this study. If $NSRE_{v,i}$ or $PSRE_{v,i} > 0$, it indicates that the redistribution of agricultural product trades on the utilization of nutrient resources present a positive sending-receiving effect of saving N or P; if $NSRE_{v,i}$ or $PSRE_{v,i} < 0$, it indicates a negative sending-receiving effect on the utilization of nutrient resources of wasting N or P; n is the total number of nutrients flows.

Spillover effect and telecoupling effect.

The spillover effect of physical trade flow and virtual nutrients flow were calculated by ratio of re-export to total export for receiving system. The calculation method of spillover effect for physical and virtual flow is:

$$NSE_{p,l} = \sum_{i=1}^m N_{p,i} \times T_{i,B,C_k} \quad (6)$$

$$PSE_{p,l} = \sum_{i=1}^m P_{p,i} \times T_{i,B,C_k} \quad (7)$$

$$NSE_{v,l} = \sum_{i=1}^n (N_{v,i,C_k} - N_{v,i,A}) \times T_{i,B,C_k} \quad (8)$$

$$PSE_{v,l} = \sum_{i=1}^n (P_{v,i,C_k} - P_{v,i,A}) \times T_{i,B,C_k} \quad (9)$$

where $NSE_{p,l}$, $PSE_{p,l}$ are physical spillover effect of the l th trade route, kg; $NSE_{v,l}$, $PSE_{v,l}$ are virtual spillover effect of the l th trade route, kg; N_{v,i,C_k} and P_{v,i,C_k} are the virtual N, P contents of local product i in systems C_k , kg/t, it is calculated according to the re-export volume from B to C_k and the virtual N content of C_k ; the telecoupling effect is the summation of sending-receiving effect and spillover effect of a specific trade route. The calculation method is:

Physical telecoupling effect is the total amount of physical sending-receiving effect and physical spillover effect; Similarly, the virtual telecoupling effect is the summation of the virtual sending-receiving effect and virtual spillover effect:

$$NTE_p = \sum_{l=1}^m NSRE_{p,l} + \sum_{l=1}^m NSE_{p,l} \quad (10)$$

$$PTE_p = \sum_{l=1}^m PSRE_{p,l} + \sum_{l=1}^m PSE_{p,l} \quad (11)$$

$$NTE_v = \sum_{l=1}^m NSRE_{v,l} + \sum_{l=1}^m NSE_{v,l} \quad (12)$$

$$PTE_v = \sum_{l=1}^m PSRE_{v,l} + \sum_{l=1}^m PSE_{v,l} \quad (13)$$

where NTE_p , PTE_p is the telecoupling effects for N, P flows, kg; NTE_v , PTE_v is the virtual telecoupling effects for N, P flows, kg; NTE , PTE is the telecoupling effects for N, P flows, kg.

And we added the result for sending-receiving effect, spillover effect, and telecoupling effect, see details as follows:

“Sending-receiving effects, spillover effects, and telecoupling effects in agricultural trade network. From a global perspective, the international nutrient flows generated in telecoupled systems of agricultural trade have become an increasingly important part of the global nutrient cycle, leading to the redistribution of nutrient resources. In 2016, the total physical N + P flows reached nearly 27% of the total volume of N + P in global agricultural products consumption and the total virtual N + P flows accounted for about 33.7% of the total N + P inputs into the global agricultural system^{23,27}. The results demonstrate that the embedded nutrient flows presented significant positive sending-receiving effects on saving N, P resources. The positive sending-receiving effects generally increased from 1997-2016 (Fig.6a, 6b). The trade flow is defined as an efficient flow when saving nutrients, while the opposite consequence is defined as inefficient. For example, the trade flow of soybeans from Brazil to China is efficient. The global physical sending-receiving effect of N and P was up to 23.3 Tg and 3.02 Tg, respectively, and the global virtual sending-receiving effect of N and P was up to 62.3 Tg and 73.9 Tg, respectively, showing a high positive saving effect. The virtual N and P sending-receiving effects of lentils, hazelnuts, coconuts, sunflower seed, and seed cotton is 100 times higher than the physical N and P sending-receiving effect.

The dispersion degree of spillover effect of different trade routes is higher, which is related to the total import and export volume and re-export rate of trading countries (Fig. 6c, 6d). The

largest physical N and virtual N spillover effects were 1.37 Tg for the German-Netherlands route and 7.56 Tg for the China-USA route, respectively. In general, the spillover effect of physical flow is significantly greater than that of virtual flow, and this difference holds for the entire study period. However, as time goes by, N and P spillovers of both physical and virtual flows show an increasing trend. In 2016, the total spillover effect of N flows among global trade entities reached 3.81 Tg, while the total spillover effect of P flow entities reached 1.23 Tg. Wheat, corn, rapeseed, and soybean were the four products with the largest spillover effects, which reached more than 100 Gg. However, the spillover effect of wheat and corn decreased with time, soybean increased first and then decreased, and rapeseed increased by years. The telecoupling effect of global agricultural trade fluctuates and increases over time, and the sending-receiving effect accounts for a large proportion, which indicates that the spillover effects of re-export target countries are less than the internal effect of the sending and receiving systems.

Fig. 6. Sending-receiving effects, spillover effects, and telecoupling effects of agricultural trade networks. a. Physical and virtual sending-receiving effects of nitrogen (N) and phosphorus (P) flow for each kind of product (red represents positive values and blue represents negative values); b, c. spillover effects of nitrogen (N) and phosphorus (P) flow in 1997, 2003, 2009, and 2016; d, e. global agricultural trade physical telecoupling effect of nitrogen (N) and phosphorus (P) from 1997 to 2016; f, g. global agricultural trade virtual telecoupling effect of nitrogen (N) and phosphorus (P) from 1997 to 2016.

L98: please briefly defined if the 42 products includes crops and/or livestock products, and their share in comparison to total trade flows.

Answer: Thanks for your comments. We have adjusted the product types. Now 320 types of products are considered in our research, which accounts for over 95% of total trade flow. The detailed list of trade productions is provided in the Supplementary Information.

L101: natural resources depletion?

Answer: Yes, thanks for your reminder. We have revised the sentence. Please see the details below: “Specific objectives were to 1) clarify the spatial-temporal dynamic characters of physical and virtual N, P flows in global telecoupled agricultural trade networks; 2) analyze the natural resource depletion and environmental impacts of N, P flows on different sending and receiving systems and on global nutrient cycling; and 3) discuss the policy suggestions conducive to sustainable utilization of nutrient resources and environmental protection.”

L119-120: Are you referring here to crop based nutrients flows, or animal products are also included here?

Answer: Thanks for your comments! Here it should be “crop based nutrients flows”, sorry for the unclear expression. We have corrected it.

L125-127: this results is important, since it shows that the trade of livestock products has a hige impact on virtual resources (and so potential environmental impact). Maybe this result could be better developed in the discussion?

Answer: Thanks for your suggestion. We have added discussion on this point. “The virtual nitrogen of beef mainly comes from feed, excrement, and other materials input during production period. Although the virtual nitrogen directly produced in the feeding process is less, the virtual nitrogen input for cultivating animal feed is higher, resulting in the virtual nitrogen content of beef and its processed products being significantly higher than that of other plant products. Adjusting the sources of animal feed and improving dietary consumption by replacing beef with other food can effectively reduce virtual nitrogen.”

Figure 2: As referred to my previous comment, how the different virtual flows for each traded category have been estimated?

Answer: Thanks for your comments! We have calculated the virtual nutrient embedded in each agricultural product at country and regional levels. The differences among countries or regions are caused by production pattern and fertilization situations. As some countries play role as “transfer station” in the trade process, the products imported and exported through these countries are not produced in their local cropland. Therefore, we use the data provided by Kastner to exclude “transfer station” and get the trade data from producer country to consumer country, which is matched with the N and P input coefficient of the producer country. The virtual N and P calculated by this method is the real nutrient transfer globally. We combine the trade matrix (excluding re-export) data and the virtual nutrients embedded in each agricultural product to calculate each virtual flow along with the trade flow.

We have revised the sentences in the Method section to make it clearer: “S1.2.1 Calculation of virtual nutrient contents of crops

In this study, gridded crop-specific fertilizer, and manure application data at 0.083 degrees, downloaded from EarthStat (<http://www.earthstat.org/nutrient-application-major-crops/>), were used to extract nitrogen (N) and phosphorus (P) application rates for 17 major crops in more than 221 countries and regions in 2000. These 17 major crops were wheat, maize, rice, barley, millet, sorghum, soybean, sunflower, potato, cassava, sugarcane, sugar beet, oil palm fruit, rapeseed, groundnut, seed cotton and rye.

For the calculation of N and P application rates for the other 85 minor crops, as direct data on N and P application rates were not available, crop-specific yield, N and P content, crop-specific harvest area fraction and N and P balance which responded to excess or deficiency of nitrogen and phosphorus compared to the current level were used for the calculation, as shown in the following equation:

$$N_{app} = crop_{yield} \times N_{content} + crop_{HAF} \times crop_{balance} \quad (1)$$

where N_{app} is N application rate, kg; $crop_{yield}$ is crop-specific yield, kg; $N_{content}$ is N content; $crop_{HAF}$ is harvested area fraction of the specific crop; $crop_{balance}$ is the N balance of crop, kg.

The calculation process for phosphorus was similar:

$$P_{app} = crop_{yield} \times P_{content} + crop_{HAF} \times crop_{balance} \quad (2)$$

where P_{app} is P application rate, kg; $crop_{yield}$ is crop-specific yield, kg; $P_{content}$ is P content; $crop_{HAF}$ is harvested area fraction of the specific crop; $crop_{balance}$ is the P balance of crop, kg.

N and P content data were sourced from <https://fao.org/economic/the-statistics-division-ess/publications-studies/publications/nutritive-factors/en/> and

<https://www.gov.uk/government/publications/composition-of-foods-integrated-dataset-cofid>.

Crop-specific without N or P content were replaced by taking the average of the same item group. Crop-specific yields and other related data were sourced from

<http://www.earthstat.org/>.

The N and P application rate of 102 different crops in different countries in 2000 were finally obtained. The results obtained by this calculation method matched with the actual N and P application rate. Then, the actual N and P application rate and the calculated N and P application rate were linearly scaled from 2000 to 2016 over all countries to ensure that the total N and P application rate of 102 crops was consistent with the actual N and P application rate.

It should be noted that N and P contents used in this study were based on the weight of the fruiting part rather than the whole crop-specific yield, which could be uncertain for global-scale calculation. In addition, the N balance deficit of some countries was too large, resulting in negative N and P application rates. In this study, this paper treated it as an outlier and normalized to 0.”

L138: net receiving/exporting systems are defined only in Figure 4. I think this definition should appear here. Furthermore, it is unclear whether the authors refers to physical or virtual resources.

Answer: Thanks for your comments! We determine whether a country or region is a net receiving or net sending system by comparing the relationship between nutrient inflow and outflow. In addition, we compare and analyze both physical and virtual nutrients. We have revised the relevant sentences.

L146: total physical or virtual nutrients?

Answer: Thanks for your comments! We have revised the sentence: “The top 10 net sending systems contributed to about 70% of the total exported physical nutrients, and the top 20 net sending systems contributed to more than 90% of the total exported physical nutrients.”.

Figure 4: It may be interesting to add the total virtual imports (if the figure does not become too charged).

Answer: Thank you for your advice! We have added a new bar to indicate the imports virtual flows as you suggested.

Fig. 4. Top 5 net nutrient receiving systems and sending systems in 2016. a. Top 5 net receiving systems and sending systems of nitrogen (N); b. Top 5 net receiving systems and sending systems of phosphorus (P). Countries are ordered by the net volume of physical N or P they imported or exported. The green and red labels indicate the volume of net import and export of physical nutrients, which is the length of the green column minus the length of the red column. The yellow and blue labels represent volume of net import and export of virtual nutrients, which is the length of the yellow column minus the length of the blue column.

L176: There is no Table S3 in the supplementary information.

Answer: Yes, because the Table S3 is a very large table and we have uploaded the Excel file (Table S3) separately. Please check it in the attachment.

L177 (and throughout the whole paper): please, be consistent with the units used (sometimes kg, sometimes Tg, etc). Can you report all the results to the main unit – i.e. Tg ?

Answer: Thanks for your comments! We have checked the whole manuscript and revised the unit.

L173-188: I think that this paragraph could be shortened to gain some space for expanding some of the results/discussions following my previous comments.

Answer: Thanks for your comments! We've shortened this paragraph. We have deleted information that can be obtained directly from Table 1.

L188: you mention here a significance analysis, but I couldn't find any information on such analysis in the methods. This should be clarified.

Answer: We have presented a detailed assessment of nutrient flow and distance between countries in supplementary materials Figure S5b.

Table 1: if I am not wrong, some of the numbers reported in column "Virtual P" do not correspond to flows showed in Figure 5. For example, the flow from BRA+ARG (SA) to CHN (E.AS) of 0.67Tg does not match with the flow reported in figure 5b (0,28Tg)

Answer: Thanks! We have checked the whole manuscript and revised.

Figure 5. This figure is interesting but I would revise it to (i) delete all minors flows, (ii) indicated the values in Tg on the round axis. As mentioned, also check also the match with data reported in table 1.

Answer: Thanks for your suggestion! We have redrawn the figure as follows:

Fig. 5. The nitrogen (N) and phosphorus (P) flow patterns through the trade of agricultural products in 2016 (Gg/y). a. physical N flow; b. physical P flow; c. virtual N flow; and d. virtual P flow. The link colors correspond to the sending systems. The arrow points to the receiving systems. According to the FAO, all the countries were divided into 10 country groups by geography, i.e., East Asia (E.AS), South-East Asia (S-E.AS), Central and West Asia (C&W.AS), North America (NA), Central America (C.A), South America (SA), Oceania (OA), Africa (AF), East Europe (E.EU), West Europe (W.EU).

L216: why agricultural production efficiency was the main economic cause for high export? Either (i) a result or (ii) a reference should be brought in support.

Answer: Thanks for your comments! We have referred to the opinions of reviewer #1. Since this sentence is quite speculative, we have deleted it.

L223: 72% of physical or virtual nutrients?

Answer: Thanks! We have revised the sentence as follows:

“A total of 69% of the physical N exported by Brazil came from trading soybeans in 2016, and soybean exports brought considerable economic benefits to Brazil.”

L232-233: this is true for P, but much less the case for N (which do not accumulate in soils).

L233-234: this sentence is unclear. Please clarify. Why would it possible to recycle N and not P.

Answer: Most of the world's production of phosphate fertilizer relies on mineral extraction because it is relatively fixed and difficult to transfer in soil. It is also difficult for natural phosphorus to be recycled into the soil system as a nutrient for plants to use.

L234-236: This would not be the case if exports are compensated by fertilizers inputs.

Answer: The application of fertilizer can indeed alleviate the problem of soil fertility loss, but as a non-renewable mineral resource, the continuous export of phosphate rock has the risk of resource shortage for a country or region. We have revised the sentence and made it clearer.

L239: P depletion → how was the P budget calculated? Is this a calculation done by the authors?

Answer: The data of P budget is from Lun et al., 2018 (Reference Number 24: Global and regional phosphorus budgets in agricultural systems and their implications for phosphorus-use efficiency. Earth Syst. Sci. Data, 10, 1–18, 2018. <https://doi.org/10.5194/essd-10-1-2018>). We have added the references and revised the Methods section.

S1.2.1 Calculation of virtual nutrient contents of crops

In this study, gridded crop-specific fertilizer, and manure application data at 0.083 degrees downloaded from EarthStat (<http://www.earthstat.org/nutrient-application-major-crops/>)

were used to extract nitrogen (N) and phosphorus (P) application rate for 17 major crops in more than 221 countries and regions in 2000. These 17 major crops were wheat, maize, rice, barley, millet, sorghum, soybean, sunflower, potato, cassava, sugarcane, sugar beet, oil palm fruit, rapeseed, groundnut, seed cotton and rye.

For the calculation of N and P application rate for the other 85 minor crops, as direct data on N and P application rate are not available, crop-specific yield, N and P content, crop-specific harvest area fractional and N and P balance which responded to excess or deficiency of nitrogen and phosphorus compared to the current level were used for the calculation, as shown in the following equation.

$$N_{app} = crop_{yield} \times N_{content} + crop_{HAF} \times crop_{balance} \quad (1)$$

where N_{app} is N application rate, kg; $crop_{yield}$ is crop-specific yield, kg; $N_{content}$ is N content; $crop_{HAF}$ is harvested area fraction of crop-specific; $crop_{balance}$ is N balance of crop, kg.

The calculation process for phosphorus is similar.

$$P_{app} = crop_{yield} \times P_{content} + crop_{HAF} \times crop_{balance} \quad (2)$$

where P_{app} is P application rate, kg; $crop_{yield}$ is crop-specific yield, kg; $P_{content}$ is P content; $crop_{HAF}$ is harvested area fraction of crop-specific; $crop_{balance}$ is P balance of crop, kg.

N and P content data sourced from <https://fao.org/economic/the-statistics-division-ess/publications-studies/publications/nutritive-factors/en/> and <https://www.gov.uk/government/publications/composition-of-foods-integrated-dataset-cofid>. Crop-specific which has no data of N or P content were replaced by taking the average of the same item group. Crop-specific yields and other related data sourced from <http://www.earthstat.org/>.

The N and P application rate of 102 different crops in different countries in 2000 were finally obtained. The results obtained by this calculation method matched with the actual N and P application rate. Then, the actual N and P application rate and the calculated N and P application rate were linearly scaled from 2000 to 2016 over all countries to ensure that the total N and P application rate of 102 crops was consistent with the actual N and P application rate.

It should be noted that N and P contents used in this study were based on the weight of the fruiting part rather than the whole crop-specific yield, which could be uncertain for global-scale calculation. In addition, the N balance deficit of some countries was too large, resulting in negative N and P application rates. In this study, this paper treated it as an outlier and normalized to 0.

L239-241: I agree that P depletion would happen in the long term. But negative P budgets can be

partially buffered by high soil P stocks. I think that the authors should discuss this point here. Furthermore, looking at figure S7b, it seems that Net exporter countries are not necessarily linked with a negative P budget. Maybe there is a cleared link with total P virtual exports?

Answer: Thanks for your comments! We added the comparative relationship between nutrient gains and losses caused by trade and total resources. If the net outflow of nutrients occurs in areas rich in nutrient resources, agricultural production and soil nutrients will not be affected, but there is still a large risk of resource loss. The Fig. S6 is as follows: the red areas bear more of the risk of resource shortages, such as U.S. and Brazil.

Fig. S6. Physical nitrogen (N) and phosphorus (P) flows through the trade of agricultural products as percentages of domestic mineral N and P fertilizer uses in 2016. a. N flow as a percentage of domestic mineral N; and b. P flow as a percentage of domestic mineral P. Negative values represent outflow of N, P, and positive values represent inflow of N, P.

L248-249: I think this example is out of place here.

Answer: Thanks for your suggestion! We have deleted this example.

L259 “imports became the main nutrient source”: this is just theoretical as a number of organic

waste resources after the consumption of agricultural commodities are not exploited in many countries (especially waste-water resources).

Answer: Yes, we agree with your opinion. We only discussed the possible benefits of nutrient imports. The nutrient recycling and using can hardly be achieved in a few years, whether from technology or management.

L266-263: “can supplement nutrients reserves”: this is true for P, but not for N

L264: maybe you could report an indicator, such as a ration between the P imported over P deficit?

Answer: Yes, we agree with you. Our data only refers to this risk, which is more applicable to P.

L266: how does P regenerates? This formulation is unclear to me.

Answer: We have revised the sentences. “The import of virtual nutrients means the conservation of domestic nutrient resources, especially P (due to its difficulty in regenerating through natural cycling).”

L268: I would start a new paragraph here.

Answer: Thanks for your suggestions. We have revised as you suggested.

L272-273: Why this promotes “the increase of nutrient consumption”? Unclear to me.

Answer: We have changed to “the increase of food consumption”.

L276: please revise this sentence (English formulation)

Answer: Thanks for your suggestions. We have revised as you suggested. “Many net receiving systems such as South Korea and Japan have increasing environmental risks due to large imports of physical nutrients (Fig.S7)”

Figure 6: this is a nice figure. I would just invert the colors if possible (red for wasting and blue for savings).

Answer: Thanks for your suggestions! We have revised Figure 6 to show the spillover effects in agricultural trade networks. The new version is as follows:

Fig. 6. Sending-receiving effects, spillover effects, and telecoupling effects of agricultural trade networks. **a.** Physical and virtual sending-receiving effects of nitrogen (N) and phosphorus (P) flow for each kind of product (red represents positive values and blue represents negative values); **b, c.** spillover effects of nitrogen (N) and phosphorus (P) flow in 1997, 2003, 2009, and 2016; **d, e.** global agricultural trade physical telecoupling effect of nitrogen (N) and phosphorus (P) from 1997 to 2016; **f, g.** global agricultural trade virtual telecoupling effect of nitrogen (N) and phosphorus (P) from 1997 to 2016.

L313-315: this is especially true for P, which is a resource that is highly concentrated in a few countries (morocco, US, china).

Answer: Yes, we agree with your opinion.

L345: here is “well defined network characteristics?”

Answer: We have revised the sentence: “Agricultural trade networks are subsets of global trade networks, which has typical network characteristics (Fig. 1a).”

L356: How where “spillover systems” considered in this analysis?

Answer: In telecoupled agricultural trade networks, the countries (or regions) that are affected by directly trade but not directly involved in trade route are defined as "spillover systems" (Fig. 1b). The spillover effect of trade network is a aggregation of spillover effects of many individual trade routes. The revised section is as follows:

“Agricultural trade in the context of the telecoupling framework.

Agricultural trade networks are subsets of global trade networks, which have typical network characteristics (Fig. 1a)⁴⁴. A total of 221 countries or regions and 27819 different export-import routes were involved in global agricultural trade networks in 2016⁴⁵. The international agricultural trade networks were viewed through the telecoupling framework⁸ that facilitates more comprehensive and systematic understanding than conventional trade research. In this study, the conceptual model of agricultural trade mainly focused on physical and virtual nutrient flows among different countries or regions. In telecoupled agricultural trade networks, exporting and importing countries (or regions) are defined as sending and receiving systems, respectively; and the countries (or regions) that are affected by trade but not directly involved in trade are defined as "spillover systems" ^{8,13}. Global trade network is an aggregation of thousands of trade routes. For a specific trade route (telecoupling subsystem), importer or exporter countries not only represent sending and receiving systems but also play roles as “re-export transit stations”, for example, Singapore, which imports rice from India and exports it to Australia, is not directly involved in its own rice production and consumption. Therefore, we define the countries in the telecoupling subsystem excluding the sending system and receiving system as spillover systems. The diagram of spillover systems in telecoupling framework is shown in Supplementary Materials Fig. S1. It is worth noting that the roles of sending, receiving and spillover systems in the trade network are not fixed, these three are only clear in each specific trade route; in other words, a country may have multiple roles in global trade network.

Fig. S1 Systems and effects within an agricultural trade network under the telecoupling framework. a. Schematic diagram of global agricultural trade network; b. A specific trade route shows sending system, receiving system, and spillover system as well as sending-receiving effect, spillover effect, and telecoupling effect.”

Sending-receiving effects

Trade leads to the sending-receiving effects of redistributing global nutrient resource utilizations. As for physical nutrients, the sending-receiving effect is actual physical nutrients transferred from A to B, the specific calculation formula is:

$$NSRE_{p,i} = \sum_{l=1}^m N_{p,i} \times (T_{i,A,B} - T_{i,B,C_k}) \quad (2)$$

$$PSRE_{p,i} = \sum_{l=1}^m P_{p,i} \times (T_{i,A,B} - T_{i,B,C_k}) \quad (3)$$

where, A and B are the sending and receiving systems, respectively, in the l th trade route or nutrients flow of the product i ; $T_{i,A,B}$ is the trade volume of product i exported from A to B, t; T_{i,B,C_k} is the trade volume of product i re-exported from B to C_k , t; $N_{p,i}$ and $P_{p,i}$ is the N, P content of product i ; $NSRE_{p,i}$ and $PSRE_{p,i}$ are the physical sending-receiving effects of the product i ; m is the total number of trade routes.

This study explored whether the sending-receiving effects on global N, P utilizations were positive (nutrient saving) or negative (nutrient wasting). The specific calculation method is:

$$NSRE_{v,i} = \sum_{l=1}^n (N_{v,i,B} - N_{v,i,A}) \times (T_{i,A,B} - T_{i,B,C_k}) \quad (4)$$

$$PSRE_{v,i} = \sum_{l=1}^n (P_{v,i,B} - P_{v,i,A}) \times (T_{i,A,B} - T_{i,B,C_k}) \quad (5)$$

where, $N_{v,i,A}$, $N_{v,i,B}$, $P_{v,i,A}$, $P_{v,i,B}$ are the virtual N, P contents of local product i in systems A and B, kg/t; $NSRE_{v,i}$ and $PSRE_{v,i}$ are the virtual sending-receiving effects of the product i . The sending-receiving effects of nutrient flows were obtained by the flows of all agricultural trade products considered in this study. If $NSRE_{v,i}$ or $PSRE_{v,i} > 0$, it indicates that the redistribution of agricultural product trades on the utilization of nutrient resources present a positive sending-receiving effect of saving N or P; if $NSRE_{v,i}$ or $PSRE_{v,i} < 0$, it indicates a

negative sending-receiving effect on the utilization of nutrient resources of wasting N or P; n is the total number of nutrients flows.

Spillover effect and telecoupling effect.

The spillover effect of physical trade flow and virtual nutrients flow were calculated by ratio of re-export to total export for receiving system. The calculation method of spillover effect for physical and virtual flow is:

$$NSE_{p,l} = \sum_{i=1}^m N_{p,i} \times T_{i,B,C_k} \quad (6)$$

$$PSE_{p,l} = \sum_{i=1}^m P_{p,i} \times T_{i,B,C_k} \quad (7)$$

$$NSE_{v,l} = \sum_{i=1}^n (N_{v,i,C_k} - N_{v,i,A}) \times T_{i,B,C_k} \quad (8)$$

$$PSE_{v,l} = \sum_{i=1}^n (P_{v,i,C_k} - P_{v,i,A}) \times T_{i,B,C_k} \quad (9)$$

where $NSE_{p,l}$, $PSE_{p,l}$ are physical spillover effect of the l th trade route, kg; $NSE_{v,l}$, $PSE_{v,l}$ are virtual spillover effect of the l th trade route, kg; N_{v,i,C_k} and P_{v,i,C_k} are the virtual N, P contents of local product i in systems C_k , kg/t, it is calculated according to the re-export volume from B to C_k and the virtual N content of C_k ; the telecoupling effect is the summation of sending-receiving effect and spillover effect of a specific trade route. The calculation method is:

Physical telecoupling effect is the total amount of physical sending-receiving effect and physical spillover effect; same as physical telecoupling effect, the virtual telecoupling effect is the virtual sending-receiving effect and virtual spillover effect:

$$NTE_p = \sum_{l=1}^m NSRE_{p,l} + \sum_{l=1}^m NSE_{p,l} \quad (10)$$

$$PTE_p = \sum_{l=1}^m PSRE_{p,l} + \sum_{l=1}^m PSE_{p,l} \quad (11)$$

$$NTE_v = \sum_{l=1}^n NSRE_{v,l} + \sum_{l=1}^n NSE_{v,l} \quad (12)$$

$$PTE_v = \sum_{l=1}^n PSRE_{v,l} + \sum_{l=1}^n PSE_{v,l} \quad (13)$$

where NTE_p , PTE_p is the telecoupling effects for N, P flows, kg; NTE_v , PTE_v is the virtual telecoupling effects for N, P flows, kg; NTE , PTE is the telecoupling effects for N, P flows, kg.

L386: referred to my main comment #1, are virtual flows calculated for each commodity or aggregated? Please clarify.

Answer: Thanks! We adopted a new method to calculate virtual nutrients. The virtual nutrients of different crops can be effectively distinguished, and the details of the methods were added in supplementary documents.

L422-425: does the authors accounted for re-exports – i.e. commodities produced by country A, could transit though country B, to be finally consumed in country C → the FAOSTAT trade matrix does not accounts for re-exports. Some previous works on flows has developed procedures for this (Kastner et al. 2011 Tracing distant environmental impacts of agricultural products from a consumer

perspective). I think that the authors should be clarified if re-exports were accounted for or not.

Answer: Many thanks for the reminder. As some countries play role as “transfer station” in the trade process, the products imported and exported through these countries are not produced in their local cropland. Therefore, we use the data provided by Kastner et al to exclude “transfer station” and get the trade data from producer country to consumer country, which is matched with the N and P input coefficients of producer country. The virtual N and P calculated by this method is the real nutrient transfer globally. We have adjusted virtual nutrient flow calculations method and revised the manuscript.

Supplementary information

L 112-113: for manure, FAOSTAT does not provided P inputs. How was this estimated?

Answer: We have used a new method to calculate the virtual N and P input rates. Firstly, we used the crop yield data provided by FAO and the content of solid nutrient elements to calculate the solid nutrition, and combined with the balance of farmland N and P to get the surplus level of farmland N and P. Finally, we calculated all the N and P inputs, including fertilizer and organic fertilizer, etc. In this way, we can obtain the virtual nutritional parameters of different crops in different countries.

L140: why primary livestock products are also transformed here?

Answer: Sorry for the typo. Here it should be “Processed livestock products”, we have revised the sentence. In addition, we re-describe the calculation of primary animal products and processed livestock products separately in the supplementary materials.

Figure S4, L208-212: I suggest to report these lines also to the caption of Figure 3.

Answer: Thanks for your kind suggestions. We have redrawn and revised the Figure3. See details as follows:

Fig. 3. Nitrogen (N) and phosphorus (P) receiving and sending systems in 1997(a-c, g-f) and 2016(d-f, j-l). N flows through gross export (a, d), gross import (b, e) and net budget (c, f); P flows through gross export (g, j), gross import (h, k) and net budget (i, l). Net nutrient flows were calculated as the difference between gross physical N (or P) imported and gross physical N (or P) exported. The red and blue colors indicate exporting and importing, respectively; gray color means no available data.

Figure S7: As mentioned before, please clarify how P budgets were calculated

Answer: The data of P budget is from Lun et al., 2018 (Reference Number 24: Global and regional phosphorus budgets in agricultural systems and their implications for phosphorus-use efficiency. *Earth Syst. Sci. Data*, 10, 1–18, 2018. <https://doi.org/10.5194/essd-10-1-2018>). We have added the references and revised the Method section.

figure S8: Please, verify the quality of the figure. What does black and white colored countries stand for? Maybe you could also scale down your legend (no dark blue/red countries in the maps).

Answer: Sorry that this figure presented the wrong appearance on the PDF file. We have modified and uploaded the source file to ensure the figure quality in this version. Please see details as follows:

Fig. S6. Physical nitrogen (N) and phosphorus (P) flows through the trade of agricultural products as percentages of domestic mineral N and P fertilizer uses in 2016. a. N flow as a percentage of domestic mineral N; and b. P flow as a percentage of domestic mineral P. Negative values represent outflow of N, P, and positive values represent inflow of N, P.

Reviewer #1 (Remarks to the Author):

Comments on manuscript Nat Comms XXX

Overall, this manuscript underwent significant improvements compared to the previous version. This makes it easier to read and understand. The manuscript is about to reach the appropriate level of solidity and originality that is required for any Nature Communication article. However, some significant improvements are needed, in particular to achieve the appropriate level of novelty for publication.

Besides, I consider that several major comments I made on the initial version of the manuscript have not been considered, which is always a bit disappointing. I strongly recommend to the authors to take those comments as seriously as possible.

The line numbers I refer throughout my review are based on the track-change mode of the revised manuscript.

My first major comment is about the originality of the manuscript. As said during the first round of reviews, I am not sure the manuscript brings the necessary amount of novelty. More precisely, several papers about have already been published on global physical and virtual N and P flows during the last decade. The added-value of this manuscript compared to those published studies is not very clear. My recommendation is to focus the paper on the comparison of physical vs. virtual nutrient flows. Indeed, none of previous papers brought together and compared the physical vs. virtual N and P global flows whereas the comparison is promising to assess to what extent trade is affected by differences in nutrient-use efficiency. This is something the authors should explore significantly more in their manuscript. In particular, much more could be said about Figure 5, which is currently an under-commented figure. Without digging into that direction, I am not sure the manuscript will bring the appropriate level of novelty for publication in Nature Communications.

My second major comment is about the clarity, precision and rigor of the text. Several sections, many sentences and some figures were significantly improved by the authors during their revisions. However, too many sections and sentences remain vague and hardly understandable. This is especially true about what the authors call spillover effects. The Methods at lines 464-465 explains that spillover effect is based on re-export of traded products whereas the only example given about spillover (at line 274) is about indirect effects of change in trade flow, that has nothing to do with re-export. The level of confusion is again increased by the Supplementary Methods S1.3 that explains that re-export trade has been neutralized, following Kastner et al (2011) method. This is an example of the so confusing text; see also my many minor comments that tag unclear sentences and sections.

My third major comment is about the saving or wasting of nutrient that is brought by trade. This topic was addressed thanks to a 'self-reliant scenario' in the initial version of the manuscript. The authors have decided to remove this self-reliant scenario in this revised version, which is fair enough. The topic is now addressed through the equations 4 and 5 that look fine. However, although the text provides some conclusions about this (e.g., lines 41, 225-226, 330-336, 359-361), I could not see where the related results are shown. Showing the results and data about those saving or wasting of nutrient through trade is strongly needed before any conclusion can be drawn on this! It would also help to dig a bit more about those wasting or saving effects.

My last major comment is about the lack of comparison of this study with previously published studies on the same topics. This is especially needed to highlight to what extent this study is original and novel. The comparison with previous studies (e.g., from Lun et al 2018 or Barbieri et al 2022) should include both the methods – most virtual nutrient flows are based on mineral fertilizer use whereas this one accounts for all sources of nutrients added to soils – and the results. This would help this study to be

more solid.

Addressing those major comments is really needed and I recommend the authors to take them seriously in order to bring the manuscript at the appropriate level for publication.

In addition, please find a series of minor comments:

- line 30-33: this is a vague sentence. Please be more precise and sharp
- line 35 (and elaborated other components...): this is vague
- line 58-60: too vague and general sentence. Please be more precise or sharp
- Figure 2: please clarify if soybean is included in the 'beans and nuts' category. Because soybean is commented in many different places of the text, and because of the key role of soybeans in physical N and P flows through trade, I suggest to put soybeans in a separate trade category.
- Line 136-138: I can't understand the sentence 'the virtual N produced during the feeding process is less than the virtual N input for cultivating animal feed'. Nitrogen is not produced: it is fixed, lost, applied, transferred but not produced!
- Line 151: consider replacing 'g-f' by 'g-i'.
- Figure 4 was really difficult to understand for me. Reading the figure, I understand that China is a major gross physical N exporter (positive light green bar)... whereas the text explains that China is a major physical N importer. Similarly, based on the figure, it looks like USA are a major gross N importer (negative dark red bar), whereas USA are known for being a net N exporter. In addition, I could not understand what means the gray segment at the bottom of each physical N or P bars. The same is true about the blue segment at the bottom of each virtual N or P bars. Moreover, I could not understand what means 'ratio of domestic N or P fertilizer use'. The ratio between what and what? Finally, removing the numbers at the bottom of each bar would make the figure easier to read. This whole set of comment suggests that this figure needs profound revisions, eventually by drawing a new figure from scratch.
- Line 222-223 ('the total physical N+P flows reached nearly 27% of the total volume of N+P in consumption of agricultural products globally'): unclear sentence. What does 'N+P flows' mean? Did you some N and P flows? I am not quite sure this would make sense. In addition, what does 'volume of N+P in consumption of agricultural products globally' mean??
- Line 224-225: what does mean 'total N+P inputs into the global agricultural system'? Is this soil nutrient inputs? Or does it include feed inputs? In other words, what are the boundaries of the 'global agricultural system'?
- Line 227: what does 'increased from 1997-2016' mean? Increase from 1997 to 2016?
- Line 229: this is not shown in this manuscript
- Lines 229-234: this is unclear. Probably because the 'sending-receiving effect' is explained in Methods only, and not in the core of the manuscript. The same is true about the saving or wasting of nutrients.
- Figure 6 is again very difficult to read. This is especially true because the sending-receiving effects and the spillover-effects are not explained in the text but only in the Methods. From the figure caption, I could not understand what positive and negative values mean. In addition, panels b and c were very difficult to understand. Why physical and virtual nutrient flows have roughly opposite directions? Finally, how are telecoupling effect calculated? Please also correct the units by using international system units, as recommended in the first round of comments.
- Line 244: I can't understand that sentence
- Line 261: what is 'system': do you mean country?
- Line 278: P is not that difficult to recycle after flowing out of the agricultural production system. It can be captured recovered through wastewater or bio-waste treatment and sludge recycling.
- Line 284-285: I can't understand the sentence that remains too general.
- Line 286: as already mentioned in the first round of review, please use system international units, not hm.
- Lines 290-293: I am not really convinced. Trade-related barriers for recycling N or P back to cropland soils are mostly due to trade of feed products for distant animal

feeding. This has little to do with kitchen residues. In addition, this sentence does not add much to the discussion. Providing avenues for addressing that question would be more effective.

- Lines 298 and 313: please consider if embedded should be replaced by embodied here.
- Line 312: 'regenerating P through natural cycling' does not mean anything clear. Please be more specific.
- Line 313: what does 'substituted from the receiving systems' mean?
- Line 382-383: this is a too general sentence that I can't understand.
- Line 389-396: this is a very general paragraph that does not bring a lot of clarity. Please try to be more specific
- Line 417-418: excluding re-export is fine. But I can't understand why this would apply to virtual but not to physical nutrient flows?
- Lines 441-445: I understand that 'sending-receiving effect' is in fact the physical nutrient flow from country A to country B, corrected by re-export. If I'm right, why using an obscure 'sending-receiving effect' expression instead of the simple 'physical nutrient transfer'?
- Equations 2 and 3: I could not understand what p and k indices mean? Also, what is the 'trade route' and how does it intervene in the equation?
- Line 455: what does 'systems' mean? Countries?
- Line 476-485: I could not understand those equations and text, essentially because I could not understand what spillover means in your framework (see my major comments). This is also probably the reason why I could not understand most of Figure 6.
- Supp lines 36-45: how N and P cropland balances were determined? Where does the related data come from?
- Supp lines 54-56: I understand that fertilization rate data were collected for year 2000. However, the authors mention some data linearization between 2000 and 2016. I could not understand where the data for year 2016 come from.
- Figure S5: why different years for N and P (2000 vs. 2010)? Please select the same year for comparing both nutrients.

Reviewer #2 (Remarks to the Author):

I found the manuscript greatly improved in comparison to the previous version, and most of my comments were fully addressed. The methodology and results are clearer and better explained for most of the points I raised in the previews round of reviews. In particular, the explanation of calculation of the sending-receiving effects on both physical and virtual flows is now much clearer.

Concerning the methods, I still found a couple of issues that could be better clarified, as follows:

1) when describing the new methods applied to calculate virtual nutrients, and, more specifically, when describing the data used to assess the crop-specific fertilizer and manure data, the authors indicate to have used for the 85 minor crops the N and P balance, corresponding to the excess or deficiency of N and P compared to the current levels (and in equation 1 and 2 in the supplementary information). Al though I agree with the procedure, I find unclear how this balances are calculated (data from which source? Are application via "seeds, irrigation water, deposition and BNF (line 425) accounted here and how?), and what the "compared to the current levels" indicates. This clarification is important because the calculation of the N and P application rates strongly influences the results presented at lines 219-234 and Figure 6. The efficiency and the sending-receiving effect due to trade really depends on the sending and receiving countries application rate. Thus, it is really important to well understand how these rates were estimated.

2) concerting the application of a correction for re-exports trade data (following the procedure proposed by Kastner et al.: the authors declare at section S.1.3

(supplementary information) to have corrected re-export flows. This seems also to be the case, since they manage to separate in between the flows from sending, receiving and spillover countries. Nevertheless, in the main text at line 415, the authors declares that the re-exports flows are omitted.

Since the nutrient consumption in agricultural production systems among different countries or regions are not the same, the flow of virtual nutrients excludes re-exports based on real trade data and realizes the traceability of trade products. How the two points are compatible with each other?

Minor comments:

L 499 – code availability. This is just a suggestion: I'm a big fan of free code access, I would this invite the authors to deposit their code on an online repository (GitHub for example)

L114 (and elsewhere): those are flows per year. I would make this explicit in the units : Tg P or N / yr

L116 : the estimation of virtual P flown in 2016 (24.5 Tg/yr) is largely higher than the amount estimated by other studies (for example, see Barbieri et al. 2021: Food system resilience to phosphorus shortages on a telecoupled planet → 5.4 Tg/yr). This is partially due to the fact that previous works have considered only P fertiliser application and not manure application, but the difference is still considerable. What do you think is the main source of difference between these estimations? It might be interesting to underline such discrepancy in the manuscript.

Figure 3: a lot of information here: maybe write explicitly the information shown for each set of maps on the side of each line and columns (i.e. N and P / gross export – import – budgets) etc. to make the figure easy reading without having to jump from the caption to the figure?

Figure 4: this figure is interesting. I am just a bit confused with the colors and the representation of the Net P and N import both for physical and virtual flows. Are these represented by the small bars at the end of the orange bars for instance? I think this could be better represented.

L290: I totally agree, this is an important point for the research agenda.

Figure 6a: most of the inefficient trade effect for both physical and virtual flows are identified with the trade of meat products. I think this important result may merit to be highlighted in the text (for example, at lines 330 in the discussion?)

L365; why export products should have a different nutritional content than non-export products? Or do you mean that we should account for the type of products that are exported vs non-exported and, so, analyse in detail the trade effects based on their nutritional content?

Points by point responses (bold) to reviewer's comments (regular)

Reviewer #1 (Remarks to the Author):

Comments on manuscript Nat Comms XXX

Overall, this manuscript underwent significant improvements compared to the previous version. This makes it easier to read and understand. The manuscript is about to reach the appropriate level of solidity and originality that is required for any Nature Communication article. However, some significant improvements are needed, in particular to achieve the appropriate level of novelty for publication.

Answer: Thanks for your comments! We appreciated your kind suggestions and comments. We have rewritten the Abstract and Introduction sections to explain the novelty and significance of this study. Also, we added some results to Results, Discussion and Supplementary Materials to support the statements about the novelty and significance.

Abstract (Line 4-9): “The flows of physical and virtual nutrients along with agricultural products have discrepant effects on natural resources in different countries. However, existing literature has not quantified or analyzed such effects yet. Comparing the differences between physical and virtual nutrients would identify trade risks and optimize nutrient efficiency. Our study quantified the physical and virtual N and P flows embedded in the global telecoupled agricultural trade network among 221 countries or regions from 1997 to 2016 and elaborated their components of the telecoupling framework.”

Introduction: “However, no studies have comprehensively measured both physical and virtual flows of N and P through global agricultural trade simultaneously, therefore making it difficult to compare the different effects of physical and virtual flows or to evaluate the telecoupling effects of N and P flows on a global scale. In addition, due to the different natural resources and environmental conditions of different countries or regions, the risks brought by agricultural trade flows are not the same, making it difficult for macro-control and resource optimization.”

Besides, I consider that several major comments I made on the initial version of the manuscript have not been considered, which is always a bit disappointing. I strongly recommend to the authors to take those comments as seriously as possible.

Answer: Thanks for your comments! We are very sorry that you did not think several of your major comments on the original version were not considered. We have checked all your previous major comments and have made further revisions in this version.

For your 1st major comment “The manuscript suffers from poor redaction in far too many places. This makes the text lacking of the needed rigorous expression. See my many detailed comments below. A serious and thorough check of the manuscript is needed. To be honest, I am on the edge of considering that such a lack of rigour of the text and figures is actually a reason for rejecting the manuscript...”

Answer: Thanks for your comment! We have checked again and revised our writing with more clear and concise words to correct the expression.

For your 2nd major comment “The authors have calculated virtual nutrient flows by considering any N or P source that is provided to soils. This includes chemical fertilisers but also animal manure, biological N fixation, nutrient in irrigation water, etc. This makes quite a difference with other studies that estimated virtual nutrient flows by focussing on chemical fertilisers only; see for instance 1,2. Such a difference in methods has to be detailed and seriously discussed by the authors. I recommend to provide (i) the reasons why these authors have decided to consider all the sources of nutrient added to soils and (ii) some comparison with previous estimates provided by the literature, e.g., about P for the USA or Europe.”

Answer: Thanks for your comment! According to the definition of virtual nutrients, we use source -sink analysis to quantify the amount of nutrient for agricultural products life cycle. To make it clearer, we have revised the Methods section: “In this paper, source-sink analysis was used to calculate virtual nutrients. In addition to fertilizer input, natural input is considered to make the results more complete⁵⁰. The sources of virtual nutrients include the application of inorganic and organic fertilizers, seeds, irrigation water, atmospheric deposition, and biological N fixation (Fig. 1c)³⁸. The nutrients, which are not absorbed by the products, e.g., discharged into the water or atmosphere and accumulated in the soil, will lead to negative impacts on the environment⁵⁰. While physical nutrients are considered as nutrients contained in traded products, virtual nutrients are the total inputs in the production of agricultural products, whether being absorbed or not. Therefore, the flows of virtual nutrients involve the importer transferring inputs and losses of nutrients in the production process to the exporter^{51,52}. The virtual nutrient contents of various agricultural products from each country or region and their specific calculation methods are explained in Table S3 and Supplementary Materials Sections 1.2 and 1.3.” As for the comparison between the results of this paper and the published literatures, we adopted the same method as Barbieri et al, which was consistent with the research conclusion of the same year. On this basis, we also supplemented the analysis on the time scale (extended the crop species and the study year).

For your 3rd major comment “The authors proposed a method to estimate to what extent trade helps to save (or loose) nutrient resources. Although I understand the method that was used, I think

it lacks two aspects. First, the self-reliant scenario (that consists of relocating any imported product within the imported country) is by itself unrealistic in many aspects: importing countries may lack of croplands, water resources, labour or even nutrient resources. This unrealistic aspect of this self-reliant scenario has to be discussed. Second, by concluding that traded products that flow from high nutrient-use efficient countries to low nutrient-use efficient countries helps to save nutrient resources, the authors neglect that trade is also a barrier to recycle nutrients back to the croplands where traded products have been produced. This should also be discussed and eventually quantified.”

Answer: Thanks for your comment! First, in the last version, we have re-considered the telecoupling effects under the telecoupling framework. We have clarified the quantification method of sending-receiving effects, spillover effects and telecoupling effects and did quantitative analysis. “Agricultural trade in the context of the telecoupling framework. Agricultural trade networks are subsets of global trade networks, which have typical network characteristics (Fig. 1a). A total of 221 countries or regions and 27819 different export-import routes were involved in global agricultural trade networks in 2016⁴⁵. The international agricultural trade networks were viewed through the telecoupling framework⁸ that facilitates more comprehensive and systematic understanding than conventional trade research. In this study, the conceptual model of agricultural trade mainly focused on physical and virtual nutrient flows among different countries or regions. In telecoupled agricultural trade networks, exporting and importing countries (or regions) are defined as sending and receiving systems, respectively; and the countries (or regions) that are affected by trade but not directly involved in trade are defined as "spillover systems" ^{8,13}. Global trade network is an aggregation of thousands of trade routes. For a specific trade route (telecoupled subsystem), importing or exporting countries (or regions) not only represent sending and receiving systems but also play roles as “re-export transit stations”. For example, Singapore, which imports rice from India and re-exports it to Australia, is not directly involved in the production and consumption of the rice imported from India. In this case, Australia is a spillover system. The diagram of spillover systems in the telecoupling framework is shown in Supplementary Materials Fig. S1. It is worth noting that the roles of sending, receiving and spillover systems in the trade network are not fixed, these three are only clear in each specific trade route; in other words, a country or region may have multiple roles in the global trade network.”

Sending-receiving effects. Trade leads to the sending-receiving effects of redistributing global nutrient resource utilizations. As for physical nutrients, the sending-receiving effect is actual physical nutrients transferred from A to B, and the specific calculation formula is:

$$NSRE_{p,i} = \sum_{l=1}^m N_{p,i} \times (T_{i,A,B} - T_{i,B,C_k}) \quad (2)$$

$$PSRE_{p,i} = \sum_{l=1}^m P_{p,i} \times (T_{i,A,B} - T_{i,B,C_k}) \quad (3)$$

where, A and B are the sending and receiving systems, respectively, in the l trade route or nutrient flow of the product i , l is the trade route number (from 1 to m); $T_{i,A,B}$ is the trade

volume of product i exported from A to B, t ; T_{i,B,C_k} is the trade volume of product i re-exported from B to C_k (the end points of re-export trade), t ; $N_{p,i}$ and $P_{p,i}$ is the physical N and P content of product i ; $NSRE_{p,i}$ and $PSRE_{p,i}$ are the physical sending-receiving effects of the product i .

This study explored whether the sending-receiving effects on global N and P utilizations were positive (nutrient saving) or negative (nutrient wasting). The specific calculation method is:

$$NSRE_{v,i} = \sum_{l=1}^n (N_{v,i,B} - N_{v,i,A}) \times (T_{i,A,B} - T_{i,B,C_k}) \quad (4)$$

$$PSRE_{v,i} = \sum_{l=1}^n (P_{v,i,B} - P_{v,i,A}) \times (T_{i,A,B} - T_{i,B,C_k}) \quad (5)$$

where, $N_{v,i,A}$, $N_{v,i,B}$, $P_{v,i,A}$, $P_{v,i,B}$ are the virtual N and P contents of local product i in country A and B, kg t^{-1} ; $NSRE_{v,i}$ and $PSRE_{v,i}$ are the virtual sending-receiving effects of the product i . The sending-receiving effects of nutrient flows were obtained by the flows of all agricultural trade products considered in this study. If $NSRE_{v,i}$ or $PSRE_{v,i} > 0$, it indicates that the redistribution of agricultural product trades on the utilization of nutrient resources present a positive sending-receiving effect of saving N or P; if $NSRE_{v,i}$ or $PSRE_{v,i} < 0$, it indicates a negative sending-receiving effect on the utilization of nutrient resources of wasting N or P; n is the total number of nutrients flows.

Spillover and telecoupling effects. Due to the existence of re-export trade, there is no complete overlap between trade importers, exporting countries, agricultural producers, and consumers. Moreover, due to the differences in production efficiency among countries, spillover effects are considered in this paper to quantify the nutrient impacts brought by the re-export trade process. Trade routes assumed as above. The calculation method of spillover effect for physical and virtual flow is:

$$NSE_{p,l} = \sum_{i=1}^m N_{p,i} \times T_{i,B,C_k} \quad (6)$$

$$PSE_{p,l} = \sum_{i=1}^m P_{p,i} \times T_{i,B,C_k} \quad (7)$$

$$NSE_{v,l} = \sum_{i=1}^n (N_{v,i,C_k} - N_{v,i,A}) \times T_{i,B,C_k} \quad (8)$$

$$PSE_{v,l} = \sum_{i=1}^n (P_{v,i,C_k} - P_{v,i,A}) \times T_{i,B,C_k} \quad (9)$$

where $NSE_{p,l}$, $PSE_{p,l}$ are physical spillover effect of the l th trade route, kg ; $NSE_{v,l}$, $PSE_{v,l}$ are virtual spillover effect of the l th trade route, kg ; N_{v,i,C_k} and P_{v,i,C_k} are the virtual N, P contents of local product i in systems C_k , kg t^{-1} ; it is calculated according to the re-export volume from B to C_k and the virtual N content of C_k ; the telecoupling effect is the summation of sending-receiving effect and spillover effect of a specific trade route.

The physical telecoupling effect is the total amount of physical sending-receiving effects and physical spillover effects. Similarly, the virtual telecoupling effect is the summation of the virtual sending-receiving effects and virtual spillover effects:

$$NTE_p = \sum_{l=1}^m NSRE_{p,l} + \sum_{l=1}^m NSE_{p,l} \quad (10)$$

$$PTE_p = \sum_{l=1}^m PSRE_{p,l} + \sum_{l=1}^m PSE_{p,l} \quad (11)$$

$$NTE_v = \sum_{l=1}^n NSRE_{v,l} + \sum_{l=1}^n NSE_{v,l} \quad (12)$$

$$PTE_v = \sum_{l=1}^n PSRE_{v,l} + \sum_{l=1}^n PSE_{v,l} \quad (13)$$

where NTE_p, PTE_p is the telecoupling effects of N and P flows, kg; NTE_v, PTE_v is the virtual telecoupling effects of N and P flows, kg; and NTE, PTE is the telecoupling effects of N and P flows, kg.”

Secondly, P recycling is one of the key strategies to improve the resilience of the trade network, but very few mechanisms currently exist to return P in different waste streams from agricultural importing countries to agricultural exporting countries. Our current understanding of methods and data sources does not support the quantitative study in this part, and we believe that this will be a key direction of future research.

For your 4th major comment: “One great asset of this manuscript is the quantification of both physical and virtual nutrient flow in the same single paper. However, the authors have not really compared those two kinds of estimates. It may be interesting, though, to explore to what extent virtual nutrient flows are driven by physical flows. I am curious to see if traded products are associated with higher or lower virtual nutrient requirements than non-traded products. Additional analysis may be expected here.”

Answer: Thanks for your comment! This study defines and quantifies the flow of physical and virtual nutrients driven by agricultural trade. According to the above principles, this paper focused on analyzing the evolution of nutrient transfer pattern, and analyzed the resource and environment risks caused by nutrient use efficiency with the telecoupling framework. On this basis, we rewrote the results and highlighted the differences between physical and virtual nutrients flows.

“Fig. 5 shows the spatial distribution of physical and virtual N and P flows associated with global agricultural trade among major country groups by geography in 2016 (see Fig. S4 for N and P flow patterns in 2000, 2005, 2010, and 2015). North America and South America were the top two nutrient sending systems. The physical nutrient export volumes of these two groups during 2016 accounted for 76% and 59% of the total global flows. The export routes from North America to East Asia and from South America to East Asia had the largest nutrient flow, while Southeast Asia was also a major nutrient importer from North and South America. In addition, the volume of physical nutrient exports from Europe was estimated as 21% of the total global flows, but most of the flows occurred among European countries themselves.”

Fig. 5. The nitrogen (N) and phosphorus (P) flow patterns through the trade of agricultural products in 2016 ($Gg\ y^{-1}$). a. virtual N flows; and b. virtual P flows. The link colors correspond to the sending systems. The arrow points to the receiving systems. According to the global ranking, the agricultural production efficiency of each product in each country (or region) is divided into three grades: high, medium, and low. Bright red represents the outflow from the country (or region) with low efficiency, light red represents the outflow from the country (or region) with medium efficiency, and green represents the outflow from the country (or region) with high efficiency; c-d.

Top 20 inefficient routes flow of N(c) and P(d) in 2016. Red color represents the flow of inefficient country (or region) to efficient country (or region), and yellow color represents the flow of inefficient country (or region) to inefficient country (or region). The bubble size of a country (or region) represents the overall nutrient conversion efficiency of the country (or region). The country (or region) names are abbreviated according to the ISO nomenclature (Supplementary Data 4).

For your 5th major comments: “Methods for estimating virtual flows are vague and poorly presented (suppl section S1). Clear and detailed methods (with complete presentation, appropriate units, and actual data sources) are lacking for explaining how nutrient inputs to soils have been estimated by crop and by country. I also recommend that fertiliser and manure application rates to soils by crop and by country are given in a supplementary table.”

Answer: Thanks for your comment! We have revised the section according to your suggestions:

Section S1. Methods for calculating the unit nutrient contents of agricultural products

“S1.1 Calculation of virtual nutrient contents of agricultural products

S1.1.1 Calculation of virtual nutrient contents of crops

In this study, gridded crop-specific fertilizer, and manure application data at 0.083 degrees, downloaded from EarthStat (<http://www.earthstat.org/nutrient-application-major-crops/>), were used to extract nitrogen (N) and phosphorus (P) application rates for 17 major crops in more than 221 countries and regions in 2000. These 17 major crops were wheat, maize, rice, barley, millet, sorghum, soybean, sunflower, potato, cassava, sugarcane, sugar beet, oil palm fruit, rapeseed, groundnut, seed cotton and rye.

For the calculation of N and P application rates for the other 85 minor crops, as direct data on N and P application rates were not available, crop-specific yield, N and P content, crop-specific harvest area fraction and N and P balance which responded to excess or deficiency of nitrogen and phosphorus compared to the current level were used for the calculation, as shown in the following equation:

$$N_{app} = crop_{yield} \times N_{content} + crop_{HAF} \times crop_{balance} \quad (1)$$

where N_{app} is N application rate, kg; $crop_{yield}$ is crop-specific yield, kg; $N_{content}$ is N content; $crop_{HAF}$ is harvested area fraction of the specific crop; $crop_{balance}$ is the N balance of crop, kg.

The calculation process for phosphorus was similar:

$$P_{app} = crop_{yield} \times P_{content} + crop_{HAF} \times crop_{balance} \quad (2)$$

where P_{app} is P application rate, kg; $crop_{yield}$ is crop-specific yield, kg; $P_{content}$ is P content; $crop_{HAF}$ is harvested area fraction of the specific crop; $crop_{balance}$ is the P balance of crop, kg.

N and P content data were sourced from <https://fao.org/economic/the-statistics-division-ess/publications-studies/publications/nutritive-factors/en/> and

<https://www.gov.uk/government/publications/composition-of-foods-integrated-dataset-cofid>.

Crop-specific without N or P content were replaced by taking the average of the same item group. Crop-specific yields and other related data were sourced from

<http://www.earthstat.org/>.

The N and P application rates of 102 different crops in different countries in 2000 were finally obtained. The results obtained by this calculation method matched with the actual N and P application rate. Then, the actual N and P application rate and the calculated N and P application rate were linearly scaled from 2000 to 2016 over all countries to ensure that the total N and P application rate of 102 crops was consistent with the actual N and P application rate.

It should be noted that N and P contents used in this study were based on the weight of the fruiting part rather than the whole crop-specific yield, which could be uncertain for global-scale calculation. In addition, the N balance deficit of some countries was too large, resulting in negative N and P application rates. In this study, this paper treated it as an outlier and normalized to 0.

S1.1.2 Virtual nutrient calculation of primary livestock products

The virtual nutrients of primary animal products mainly come from feed inputs. We used the feed conversion coefficient to convert virtual nutrients of animals into virtual nutrients of feed crops. The specific formula is as follows:

$$N_i = C_i \times F_i \times N_{feed} \quad (3)$$

$$P_i = C_i \times F_i \times P_{feed} \quad (4)$$

where N_i, P_i is virtual N or P content of the animal product, kg t^{-1} ; C_i is feed conversion coefficient of the animal, kg kg^{-1} , retrieved from Mekonnen & Hoekstra¹; F_i is fraction of crop-based feed in total feed, retrieved from Mekonnen & Hoekstra¹; N_{feed}, P_{feed} is virtual N or P content of crop-based feed of the animal product, kg t^{-1} , crop-based feed were partitioned among seven main grain feed commodities - barley, maize, peas, rapeseed, sorghum, soybean, and wheat—according to data retrieved from Herrero et al.² All the data for feed conversion coefficients, fractions, and compositions of grains in crop-based feed by animal species and by region are provided in Table S2-S4.

S1.2 Calculation of virtual nutrient contents of processed products

We use factors based on caloric equivalents, according to the method by Kastner³ to transform the virtual nutrient contents of primary crop or animal products into that of processed products. The calculation formulas are as follows:

$$N_{i,p} = K_i \times N_i \quad (5)$$

$$P_{i,p} = K_i \times P_i \quad (6)$$

where $N_{i,p}, P_{i,p}$ is virtual N or P content of the processed product, kg t^{-1} ; N_i, P_i is virtual N or P content of the primary product, kg t^{-1} ; K_i is kcal ration between the processed product and the primary product.

The kcal contents of all products are provided in Table S1, according to FAO standard factors on nutritive values (<https://www.fao.org/economic/the-statistics-division-ess/publications->

studies/publications/nutritive-factors/en/).”

The line numbers I refer throughout my review are based on the track-change mode of the revised manuscript.

Answer: Thanks for your kind marks and reminder, it really helps us a lot!

My first major comment is about the originality of the manuscript. As said during the first round of reviews, I am not sure the manuscript brings the necessary amount of novelty. More precisely, several papers about have already been published on global physical and virtual N and P flows during the last decade. The added-value of this manuscript compared to those published studies is not very clear. My recommendation is to focus the paper on the comparison of physical vs. virtual nutrient flows. Indeed, none of previous papers brought together and compared the physical vs. virtual N and P global flows whereas the comparison is promising to assess to what extent trade is affected by differences in nutrient-use efficiency. This is something the authors should explore significantly more in their manuscript. In particular, much more could be said about **Figure 5**, which is currently an under-commented figure. Without digging into that direction, I am not sure the manuscript will bring the appropriate level of novelty for publication in Nature Communications.

Answer: Thanks for your comments! We have followed your very helpful suggestion and have re-written the literature review section and discussed the novelty and urgency of why we analyzed both the N and P, virtual and physical nutrients. Virtual nutrient flow patterns were analyzed in this version (Figure 5 was re-drawn). In order to distinguish the characteristics of each route, we classified the routes according to the production efficiency of the sending system and the receiving system. Also, we have added results on the patterns of physical flows of both N and P (Fig. S4). We added some sentences in Results section and Supplementary Materials regarding the analysis and results of resources and environmental risks.

Fig. 5. The nitrogen (N) and phosphorus (P) flow patterns through the trade of agricultural products in 2016 ($Gg\ y^{-1}$). | a. virtual N flows; and b. virtual P flows. The link colors correspond to the sending systems. The arrow points to the receiving systems. According to the global ranking, the agricultural production efficiency of each product in each country (or region) is divided into three grades: high, medium, and low. Bright red represents the outflow from the country (or region) with low efficiency, light red represents the outflow from the country (or region) with medium efficiency, and green represents the outflow from the country (or region) with high efficiency; c-d. Top 20 inefficient routes flow of N(c) and P(d) in 2016. Red color represents the flow of inefficient

country (or region) to efficient country (or region), and yellow color represents the flow of inefficient country (or region) to inefficient country (or region). The bubble size of a country (or region) represents the overall nutrient conversion efficiency of the country (or region). The country (or region) names are abbreviated according to the ISO nomenclature (Supplementary Data 4).

Fig. S4 Global physical nitrogen (N) and phosphorus (P) flow patterns in 2000, 2005, 2010, and 2015 (Gg y⁻¹). a. Physical N flow pattern in 2000; b. Physical N flow pattern in 2005; c. Physical N flow pattern in 2010; d. Physical N flow pattern in 2015; e. Physical P flow pattern in 2000; f. Physical P flow pattern in 2005; g. Physical P flow pattern in 2010; h. Physical P flow pattern in 2015. The global nutrient flow patterns of physical N and P were almost the same. North America exports to East Asia and South America exports to East Asia were the two largest nutrient flow routes. Southeast Asia was also an important receiving system of nutrient exports from North and South America. In addition, the imported nutrients from Central America mainly came from North America, while Europe also imported nutrients from South America. Furthermore, the nutrient export volume of the entire European region reached about 20% of the global total in 2015, but it was mainly due to many nutrient flows within the European regions. Oceania almost did not import nutrients, and Africa almost did not export nutrients; and Oceania played a smaller role in the global nutrient flows of agricultural products.

In addition, to identify the dominant factors of country's N pollution and P resource problems, we compared the resource and environmental risks caused by import N and export P respectively.

Fig. S7. Comparison of environmental and resource risks by country in 2016. | To identify the dominant characteristics of country's N pollution and P resource problems, we compared the

resource and environmental risks caused by import N and export P respectively. Panel a showed the comparison of pollution risk from large import entities N and resource shortage risk from export P; panel b showed the comparison of the risk of pollution borne by massive export of virtual N and the risk of resource shortage caused by export of P.

My second major comment is about the clarity, precision, and rigor of the text. Several sections, many sentences and some figures were significantly improved by the authors during their revisions. However, too many sections and sentences remain vague and hardly understandable. This is especially true about what the authors call spillover effects. The Methods at lines 464-465 explains that spillover effect is based on re-export of traded products whereas the only example given about spillover (at line 274) is about indirect effects of change in trade flow, that has nothing to do with re-export. The level of confusion is again increased by the Supplementary Methods S1.3 that explains that re-export trade has been neutralized, following Kastner et al (2011) method. This is an example of the so confusing text; see also my many minor comments that tag unclear sentences and sections.

Answer: Thanks for your comments! We have rewritten the method section and illustrated the calculation method and purpose more clearly with Fig. 1 in the supplementary material.

“Spillover and telecoupling effects. Due to the existence of re-export trade, there is no one-to-one correspondence between “exporting country” and “producer”, “importing country” and “consumer”, e.g., Singapore imports sugarcane (importing country), its re-exports sugarcane and plays the role of exporting country while it is not a producer. Moreover, due to the differences in production efficiency among countries, spillover effects are considered in this paper to quantify the nutrient impacts brought by the re-export trade process. The calculation method of spillover effect for physical and virtual flow is:

$$NSE_{p,l} = \sum_{i=1}^m N_{p,i} \times T_{i,B,C_k} \quad (6)$$

$$PSE_{p,l} = \sum_{i=1}^m P_{p,i} \times T_{i,B,C_k} \quad (7)$$

$$NSE_{v,l} = \sum_{i=1}^n (N_{v,i,C_k} - N_{v,i,A}) \times T_{i,B,C_k} \quad (8)$$

$$PSE_{v,l} = \sum_{i=1}^n (P_{v,i,C_k} - P_{v,i,A}) \times T_{i,B,C_k} \quad (9)$$

where $NSE_{p,l}$, $PSE_{p,l}$ are physical spillover effect of the l th trade route, kg; $NSE_{v,l}$, $PSE_{v,l}$ are virtual spillover effect of the l th trade route, kg; N_{v,i,C_k} and P_{v,i,C_k} are the virtual N, P contents of product i in system C_k , kg t^{-1} ; it is calculated according to the re-export volume from B to C_k and the virtual N content of C_k ; the telecoupling effect is the summation of sending-receiving effect and spillover effect of a specific trade route.”

My third major comment is about the saving or wasting of nutrient that is brought by trade. This topic was addressed thanks to a ‘self-reliant scenario’ in the initial version of the manuscript. The

authors have decided to remove this self-reliant scenario in this revised version, which is fair enough. The topic is now addressed through the equations 4 and 5 that look fine. However, although the text provides some conclusions about this (e.g., lines 41, 225-226, 330-336, 359-361), I could not see where the related results are shown. Showing the results and data about those saving or wasting of nutrient through trade is strongly needed before any conclusion can be drawn on this! It would also help to dig a bit more about those wasting or saving effects.

Answer: Thanks for your comments! Thank you for your recognition of our last round of modification. The results of the saving and waste effects are shown in Fig.6a. In addition, the detailed results about the sending-receiving effect are in Supplementary Data1.

Fig. 6. Sending-receiving effects, spillover effects, and telecoupling effects of agricultural trade networks. a. Physical and virtual sending-receiving effects of nitrogen (N) and phosphorus (P) flow for each kind of product (red represents positive values and blue represents negative values); b, c. spillover effects of nitrogen (N) and phosphorus (P) flow in 1997, 2003, 2009, and 2016; d, e. global agricultural trade physical telecoupling effect of nitrogen (N) and phosphorus (P) from 1997 to 2016; f, g. global agricultural trade virtual telecoupling effect of nitrogen (N) and phosphorus (P) from 1997 to 2016. (All data was shown in Supplementary Data 1)

My last major comment is about the lack of comparison of this study with previously published studies on the same topics. This is especially needed to highlight to what extent this study is original and novel. The comparison with previous studies (e.g., from Lun et al 2018 or Barbieri et al 2022) should include both the methods – most virtual nutrient flows are based on mineral fertilizer use whereas this one accounts for all sources of nutrients added to soils – and the results. This would help this study to be more solid.

Answer: Thanks for your comments! We appreciated your kind suggestions and comments. We have rewritten the Abstract and Introduction sections to explain the novelty and significance of this study. Also, we added some results to Results, Discussion and Supplementary Materials to support the statements about the novelty and significance.

Abstract (Line 4-9): “The flows of physical and virtual nutrients along with agricultural products have discrepant effects on natural resources in different countries. However, existing literature has not quantified or analyzed such effects yet. Comparing the differences between physical and virtual nutrients would identify trade risks and optimize nutrient efficiency. Our study quantified the physical and virtual N and P flows embedded in the global telecoupled agricultural trade network among 221 countries or regions from 1997 to 2016 and elaborated their components of the telecoupling framework.”

Introduction: “However, no studies have comprehensively measured both physical and virtual flows of N and P through global agricultural trade simultaneously, therefore making it difficult to compare the different effects of physical and virtual flows or to evaluate the telecoupling effects of N and P flows on a global scale. In addition, due to the different natural resources and environmental conditions of different countries or regions, the risks brought by agricultural trade flows are not the same, making it difficult for governance and resource optimization.”

Methods section: “Compared with Lun et al ²³ and Barbieri et al ⁵⁰, this study expanded to 320 crops, including almost all conventionally traded products, the detailed description of these selected agricultural products and their physical N and P contents are listed in Supplementary Data 1.”

Addressing those major comments is really needed and I recommend the authors to take them seriously in order to bring the manuscript at the appropriate level for publication.

In addition, please find a series of minor comments:

- line 30-33: this is a vague sentence. Please be more precise and sharp

Answer: Thanks for your comments! We have revised the relevant section. “The flows of physical nutrients and virtual nutrients along with agricultural products trade have different effects on resources and the environment in specific countries or regions, but the effects have

not been simultaneously quantified and comprehensively analyzed in the literature.”

• line 35 (and elaborated other components...): this is vague

Answer: Thanks for your comments! We have revised this sentence: “Thus, this study quantified the physical and virtual N and P flows embedded in global telecoupled agricultural trade networks among 221 countries or regions from 1997 to 2016 and elaborated other components of the telecoupling framework.”

• line 58-60: too vague and general sentence. Please be more precise or sharp

Answer: Thanks for your comments! We have revised the sentences to make it clearer: “There are many such individual subsets (pairs of trading countries) in the trade networks although their roles are not fixed (a particular country may be both an exporter and an importer), the simplified concept of a spillover system and its effects within the telecoupling framework is shown in Supplementary Material Fig. S1.”

• Figure 2: please clarify if soybean is included in the ‘beans and nuts’ category. Because soybean is commented in many different places of the text, and because of the key role of soybeans in physical N and P flows through trade, I suggest to put soybeans in a separate trade category.

Answer: Thanks for your comments! Soybean is included in the beans and nuts category according to WTO category principle. For classification consistency, we maintain this regular classification in Figure 2, despite the prominent position of soybean trade. In order to make up for the shortcomings of data presentation, we provided the calculation results of all specific categories (including soybeans) and uploaded them in the form of Excel attached (Extended Data 1).

Fig. 2. Total physical and virtual nitrogen (N) and phosphorus (P) flows during 1997-2016. a. physical N flows; **b.** physical P flows; **c.** virtual N flows; and **d.** virtual P flows. All products were divided into 8 categories. The specific products contained in each category are provided in Supplementary Data 1.”

• Line 136-138: I can’t understand the sentence ‘the virtual N produced during the feeding process is less than the virtual N input for cultivating animal feed’. Nitrogen is not produced: it is fixed, lost, applied, transferred but not produced!

Answer: Thanks for your comments! We have revised the relevant sentence: “The virtual nitrogen transferred during the feeding process is less than the virtual nitrogen input for cultivating animal feed.”

• Line 151: consider replacing ‘g-f’ by ‘g-i’.

Answer: Thanks for your comments! We have revised the relevant section.

Fig. 3. Nitrogen (N) and phosphorus (P) receiving and sending systems in 1997(a-c, g-i) and 2016(d-f, j-l). N flows through gross export (a, d), gross import (b, e) and net budget (c, f); P flows through gross export (g, j), gross import (h, k) and net budget (i, l). Net nutrient flows were calculated as the difference between gross physical N (or P) imported and gross physical N (or P) exported. The red and blue colors indicate exporting and importing, respectively; gray color indicates that no data was available.

• Figure 4 was really difficult to understand for me. Reading the figure, I understand that China is a major gross physical N exporter (positive light green bar)... whereas the text explains that China is a major physical N importer. Similarly, based on the figure, it looks like USA are a major gross N importer (negative dark red bar), whereas USA are known for being a net N exporter. In addition, I could not understand what means the gray segment at the bottom of each physical N or P bars. The same is true about the blue segment at the bottom of each virtual N or P bars. Moreover, I could not understand what means 'ratio of domestic N or P fertilizer use'. The ratio between what and what? Finally, removing the numbers at the bottom of each bar would make the figure easier to read. This whole set of comment suggests that this figure needs profound revisions, eventually by drawing a new figure from scratch.

Answer: Thanks for your comments! In order to make this chart more legible, we have revised the details of this chart.

Fig. 4. Top 5 net nutrients receiving systems and sending systems in 2016. a. Top 5 net receiving systems and sending systems of nitrogen (N); b. Top 5 net receiving systems and sending systems of phosphorus (P). Countries are ordered by the net volume of physical N or P they imported or exported. The green and red labels indicate the volume of net import and export of physical nutrients, which is the length of the green column minus the length of the red column. The yellow and blue labels represent volume of net import and export of virtual nutrients, which is the length of the yellow column minus the length of the blue column. Results for all countries are shown in Supplementary Data 3. Note that the red and yellow columns start at 0, while the green or blue columns start at the bottom of the red or yellow columns. Therefore, the part where the two colors do not coincide is the net import or export value.

• Line 222-223 ('the total physical N+P flows reached nearly 27% of the total volume of N+P in consumption of agricultural products globally'): unclear sentence. What does 'N+P flows' mean? Did you some N and P flows? I am not quite sure this would make sense. In addition, what does 'volume of N+P in consumption of agricultural products globally' mean??

Answer: Thanks for your comments! According to reviewer #2's comments, we revised the "N and P" to "N+P" when we discuss the total amount of virtual nutrients. Thus, N+P flows mean the total amount of N and P flows. To make it clearer, we have revised the sentences as follows: "From a global perspective, the international nutrient flows generated in telecoupled systems

of agricultural trade have become an increasingly important part of the global nutrient cycle, leading to the redistribution of nutrient resources. In 2016, the total physical N+P nutrient flows reached nearly 27% of the total volume of N+P resources in consumption of agricultural products globally, and the total virtual N+P nutrient flows accounted for about 33.7% of the total N+P soil nutrient inputs of the global agricultural system^{23,27}. The results demonstrate that the embedded nutrient flows presented significant positive sending-receiving effects on saving N+P resources.”

• Line 224-225: what does mean ‘total N+P inputs into the global agricultural system’? Is this soil nutrient inputs? Or does it include feed inputs? In other words, what are the boundaries of the ‘global agricultural system’?

Answer: Thanks for your comments! According to Lassaletta et al.²⁰ and Lun et al.²³, “the total N+P nutrients inputs into the global agricultural system” is total soil input, considering the traceability and avoidance of duplication, it excludes feed inputs. We have revised the relevant sentence to make it clearer: “the total virtual N+P nutrient flows accounted for about 33.7% of the total N+P soil nutrient inputs of the global agricultural system.”

• Line 227: what does ‘increased from 1997-2016’ mean? Increase from 1997 to 2016?

Answer: Thanks for your comments! We have revised the relevant section. “The positive sending-receiving effects generally increased from 1997 to 2016 (Fig.6a, 6b).”

• Line 229: this is not shown in this manuscript

Answer: Thanks for your comments! Sorry for the mistake, we have deleted this sentence.

• Lines 229-234: this is unclear. Probably because the ‘sending-receiving effect’ is explained in Methods only, and not in the core of the manuscript. The same is true about the saving or wasting of nutrients.

Answer: Thanks for your comments!

Also, we have revised the relevant section. “Nutrients flow along agricultural products trade from the country or region with high efficiency of nutrient transformation to the country or region with low efficiency, which will save nutrient input compared with producing products in the country or region with low efficiency locally, which is manifested as nutrient saving effect; on the contrary, it is the waste of nutrients. The global virtual sending-receiving effect of N and P was up to 62.3 Tg and 73.9 Tg, respectively. There are a lot of products that transfer from high-efficient to low-efficient regions, and while the current global agricultural trade routes are not optimal, it still shows a high positive saving effect.”

• Figure 6 is again very difficult to read. This is especially true because the sending-receiving effects and the spillover-effects are not explained in the text but only in the Methods. From the figure caption, I could not understand what positive and negative values mean. In addition, panels b and c were very difficult to understand. Why physical and virtual nutrient flows have roughly opposite directions? Finally, how are telecoupling effect calculated? Please also correct the units by using international system units, as recommended in the first round of comments.

Answer: Thanks for your comments!

First, we have added the descriptions of various effects in the telecoupling framework in the main text. Second, different colors correspond to savings or waste effects, and are represented as positive or negative value, we have added the sentences in the revised manuscript. “If $NSRE_{v,i}$ or $PSRE_{v,i} > 0$, it indicates that the redistribution of agricultural product trades on the utilization of nutrient resources present a positive sending-receiving effect of saving N or P; if $NSRE_{v,i}$ or $PSRE_{v,i} < 0$, it indicates a negative sending-receiving effect on the utilization of nutrient resources of wasting N or P.” Third, the spillover effects of physical nutrient flows are calculated from the pure nutrient content in re-exports, so they are all positive values. The spillover effects of virtual nutrient flows are calculated from the producer, consumer, and the virtual nutrient content. However, the current re-export often ends up in countries and regions with low nutrients efficiency, so most of them show negative values. Therefore, the trends of physical nutrient flow and virtual nutrient flow are generally opposite. Fourth, the telecoupling effect is the summation of sending-receiving effect and spillover effect, this study uses this value to measure the combined impact of the sending-receiving effect and spillover effect. The units of value in this paper have been checked and modified as recommended.

• Line 244: I can't understand that sentence

Answer: Thanks for your comments! We have revised the relevant section. “The spillover effects for different trade routes are obviously different, which is related to the ratio of re-export volume to the total trade volume (Fig. 6c, 6d). The largest physical N and virtual N spillover effects were 1.37 Tg for the German-Netherlands route and 7.56 Tg for the China-USA route, respectively.”

• Line 261: what is 'system': do you mean country?

Answer: Thanks for your comments! Sorry for the mistake, we have revised the relevant section. “Overall, only 10% of the countries contributed about 90% of total exported nutrients in 2016.”

• Line 278: P is not that difficult to recycle after flowing out of the agricultural production system.

It can be captured recovered through wastewater or bio-waste treatment and sludge recycling.

Answer: Thanks for your comments! Yes, it is. However, from the economic input-output point of view, recycling P consumes additional energy and economic costs. We revised these sentences. “The reason was that recycling P from water, soil, and other environment system after flowing out of the agricultural production system may consume extra energy and economic costs.”

• Line 284-285: I can't understand the sentence that remains too general.

Answer: Thanks for your comments! According to the comments of reviewer #3 in the previous round, we added this sentence to discuss the risks of land. To make it more readable, we have revised the sentences: “Considering the risk of impact on agricultural land, the net outflows of N or P nutrients also affect the N or P cycle of regional croplands. The three most affected countries are Argentina, Ukraine, and the Russian Federation, with the area of 8.71×10^4 , 7.79×10^4 , and 6.15×10^4 km², respectively (the affected area was calculated by total net nutrient outflow and nutrient consumption per unit area).”

• Line 286: as already mentioned in the first round of review, please use system international units, not hm.

Answer: Thanks for your comments! We have revised the sentences: “Considering the risk of impact on agricultural land, the net outflows of N or P nutrients also affect the N or P cycle of regional croplands. The three most affected countries are Argentina, Ukraine, and the Russian Federation, with the area of 8.71×10^4 , 7.79×10^4 , and 6.15×10^4 km², respectively (the affected area was calculated by total net nutrient outflow and nutrient consumption per unit area).”

• Lines 290-293: I am not really convinced. Trade-related barriers for recycling N or P back to cropland soils are mostly due to trade of feed products for distant animal feeding. This has little to do with kitchen residues. In addition, this sentence does not add much to the discussion. Providing avenues for addressing that question would be more effective.

Answer: Thanks for your comments! We have deleted the sentence and added new discussion. But based on reviewer #2's comments, we also kept some sentences: “In future research, it is necessary to identify the main sources of nutrient recovery barriers and explore the optimization policies to achieve the sustainability of regional nutrient resources as much as possible. Also, it is necessary to quantify the impact of trade impeding recycling of nutrients back to the croplands to better assess the risk of nutrient losses in net sending systems.”

• Lines 298 and 313: please consider if embedded should be replaced by embodied here.

Answer: Thanks for your comments! Yes, we have revised the relevant sentences by replacing embedded with embodied.

• Line 312: 'regenerating P through natural cycling' does not mean anything clear. Please be more specific.

Answer: Thanks for your comments! We have revised the relevant section. "The import of virtual nutrients means the conservation of domestic nutrient resources, especially P (due to its difficulty to recycling through natural cycling considering economic and technological perspectives)."

• Line 313: what does 'substituted from the receiving systems' mean?

Answer: Thanks for your comments! We have revised the relevant section. "For instance, the imports to Africa in 2016 embodied 0.37 Tg of virtual P, which means the sending system instead of Africa consumed these resources."

• Line 382-383: this is a too general sentence that I can't understand.

Answer: Thanks for your comments! We have revised the relevant sentences: "In this study, we use the telecoupling framework⁸ to analyze the international agricultural trade networks because it can facilitate more comprehensive and systematic understanding than conventional trade research."

• Line 389-396: this is a very general paragraph that does not bring a lot of clarity. Please try to be more specific

Answer: Thanks for your comments! We have revised the paragraph: "A global trade network is an aggregation of thousands of trade routes. For a specific trade route (an individual telecoupled system), importing countries (or regions) not only represent receiving systems but also play roles as "re-export transit stations" (e.g., the Singapore imports rice from India and then exports rice to Australia, it is not directly consuming the rice imported from India, it serves only as a transit point for rice). The simplified diagram of telecoupling framework is shown in Supplementary Materials Fig. S1. It is worth noting that the roles of sending, receiving and spillover systems in the trade network are not fixed, and these three are only clear in each specific trade route; in other words, a country or region may have multiple roles in the global trade network."

• Line 417-418: excluding re-export is fine. But I can't understand why this would apply to virtual but not to physical nutrient flows?

Answer: Thanks for your comments! This is an important question. Since this study assumes

that the same product has the same content of physical nutrients even if it is produced from different countries, the flow analysis of physical nutrients can be achieved without considering re-export trade. Virtual nutrient is more concerned about the impacts or risks of the production process on the local resources and environment in the producer countries or regions, so we identified and eliminated the re-export trade to distinguish the real producers and consumers of the agricultural products. To make it clearer, we revised the sentences: “Since the efficiency of nutrient use (virtual nutrient content) in agricultural production systems varies among different countries or regions, we used the data that excluded re-exports to match the nutrients flow to real producer/consumer, this method can realize the traceability of trade products and virtual nutrient flows ^{48,49}.”

- Lines 441-445: I understand that ‘sending-receiving effect’ is in fact the physical nutrient flow from country A to country B, corrected by re-export. If I’m right, why using an obscure ‘sending-receiving effect’ expression instead of the simple ‘physical nutrient transfer’?

Answer: Thanks for your comments! Since our analysis of global trade in this study is based on the telecoupling framework, this paper chooses to use the term (sending-receiving effect) according to the definition and interpretation of the telecoupling framework. Also, there are nutrient flows from and to spillover systems in some situations, while sending-receiving effects focus on the effects between sending and receiving systems. Furthermore, here we focus on the impacts (the amount of physical nutrient transfer), not the process per se such as physical nutrient transfer. We have revised the Materials and Methods section for easier understanding.

- Equations 2 and 3: I could not understand what p and k indices mean? Also, what is the ‘trade route’ and how does it intervene in the equation?

Answer: Thanks for your comments! Sorry for the unclear, p in formula means physical, k in formula means the order of importer of re-export. Trade route means pathways in agricultural trade, we do the calculation for each route firstly and sum it up to get the sending-receiving effect. To make it clearer, we have revised the sentences:

Sending-receiving effects. Trade leads to the sending-receiving effects of redistributing global nutrient resource utilizations. As for physical nutrients, the sending-receiving effect is actual physical nutrients transferred from A to B, and the specific calculation formula is:

$$NSRE_{p,i} = \sum_{l=1}^m N_{p,i} \times (T_{i,A,B} - T_{i,B,C_k}) \quad (2)$$

$$PSRE_{p,i} = \sum_{l=1}^m P_{p,i} \times (T_{i,A,B} - T_{i,B,C_k}) \quad (3)$$

where, A and B are the sending and receiving systems, respectively, in the *l*th trade route or nutrient flow of the product *i*, *l*th is the trade route number (from 1 to *m*); $T_{i,A,B}$ is the trade volume of product *i* exported from A to B, T_{i,B,C_k} is the trade volume of product *i* re-exported from B to C_k (the end points of re-export trade), $N_{p,i}$ and $P_{p,i}$ is the physical N and P content of product *i*;

$NSRE_{p,i}$ and $PSRE_{p,i}$ are the physical sending-receiving effects of the product i .

• Line 455: what does ‘systems’ mean? Countries?

Answer: Thanks for your comments! Here, “system” means “countries or regions”, we have revised the sentence: “where $N_{v,i,A}, N_{v,i,B}, P_{v,i,A}, P_{v,i,B}$ are the virtual N and P contents of local product i in country or region A and B, kg t^{-1} ”

• Line 476-485: I could not understand those equations and text, essentially because I could not understand what spillover means in your framework (see my major comments). This is also probably the reason why I could not understand most of Figure 6.

Answer: Thanks for your comments!

We have rewritten the Methods section and illustrated the calculation method and purpose more clearly with Fig. 1 in the supplementary material. In telecoupled agricultural trade networks, exporting and importing countries (or regions) are defined as sending and receiving systems, respectively; and the countries (or regions) that affect or are affected by trade but are not sending/receiving systems are defined as "spillover systems"^{8,13}. A global trade network is an aggregation of thousands of trade routes. For a specific trade route, some countries (or regions) play the roles as “re-export transit stations” (e.g., Singapore imports rice from India and then exports rice to Australia, it is not directly consuming the rice imported from India, it serves only as a transit point for rice). These countries or regions are treated as spillover systems. The simplified diagram of telecoupling framework is shown in *Supplementary Materials Fig. S1*. It is worth noting that the roles of sending, receiving and spillover systems in the trade network are not fixed, and these three are only clear in each specific trade route; in other words, a country or region may have multiple roles in the global trade network.

Due to the differences in production efficiency among countries, spillover effects are considered to quantify the nutrient impacts brought by the re-export trade process.

“Spillover and telecoupling effects. Due to the existence of re-export trade, there is no one-to-one correspondence between “exporting country” and “producer”, “importing country” and “consumer”, e.g., Singapore imports sugarcane (importing country), its re-exports sugarcane and plays the role of exporting country while it is not a producer. Moreover, due to the differences in production efficiency among countries, spillover effects are considered to quantify the nutrient impacts brought by the re-export trade process. The calculation method of spillover effect for physical and virtual flow is:

$$NSE_{p,l} = \sum_{i=1}^m N_{p,i} \times T_{i,B,C_k} \quad (6)$$

$$PSE_{p,l} = \sum_{i=1}^m P_{p,i} \times T_{i,B,C_k} \quad (7)$$

$$NSE_{v,l} = \sum_{i=1}^n (N_{v,i,C_k} - N_{v,i,A}) \times T_{i,B,C_k} \quad (8)$$

$$PSE_{v,l} = \sum_{i=1}^n (P_{v,i,C_k} - P_{v,i,A}) \times T_{i,B,C_k} \quad (9)$$

where $NSE_{p,l}$, $PSE_{p,l}$ are physical spillover effect of the l th trade route, kg; $NSE_{v,l}$, $PSE_{v,l}$ are virtual spillover effect of the l th trade route, kg; N_{v,i,C_k} and P_{v,i,C_k} are the virtual N, P contents of local product i in systems C_k , kg t⁻¹; it is calculated according to the re-export volume from B to C_k and the virtual N content of C_k ; the telecoupling effect is the summation of sending-receiving effect and spillover effect of a specific trade route.”

- Supp lines 36-45: how N and P cropland balances were determined? Where does the related data come from?

Answer: Thanks for your comments! The data come from MacDonald et al (Agronomic phosphorus imbalances across the world's croplands) and Liu et al (A high-resolution assessment on global nitrogen flows in cropland). We also have added the references in manuscript.

- Supp lines 54-56: I understand that fertilization rate data were collected for year 2000. However, the authors mention some data linearization between 2000 and 2016. I could not understand where the data for year 2016 come from.

Answer: Thanks for your comments! According to literature review and data search, only the year 2000 data is available in the public data at present. Due to the big gap between the existed data and the time range of this study, we used the data of 2000 and combined other available data to estimate the data in 2016 we needed (covering study period from 2000 to 2016). The description of the method and steps as follows: “According to principle of total input nutrients consistency, the N and P application rate of 102 different crops in different countries in 2000 were obtained to the analysis. Then, we compared and verified the total nutrient input calculated by the above equilibrium method with the existing statistical data, and found that the difference between the two results was small, which could meet the calculation accuracy (compared the estimated results with nutrients application from FAOSTAT, <https://www.fao.org/faostat/en/#data/RFB>, the relative error <5%). Finally, we assumed that the changes of nutrients application of different crops were consistent with the changes of the total nutrient application. Based on this rule, the data of 2000 were linearly extended to cover the study period (2000 to 2016).”

- Figure S5: why different years for N and P (2000 vs. 2010)? Please select the same year for comparing both nutrients.

Answer: Thanks for your comments! Sorry for the mistake, they are the same year to compare the values, we have revised the legend. “Fig. S5 Cropland budget and net flows through agricultural trade of nitrogen (N) (a) in 2016 and phosphorus (P) (b) in 2016.”

Reviewer #2 (Remarks to the Author):

I found the manuscript greatly improved in comparison to the previous version, and most of my comments were fully addressed. The methodology and results are clearer and better explained for most of the points I raised in the previews round of reviews. In particular, the explanation of calculation of the sending-receiving effects on both physical and virtual flows is now much clearer.

Answer: Thanks for your comments! We have revised them one by one according to your suggestions, and the details are as follows.

Concerning the methods, I still found a couple of issues that could be better clarified, as follows:

1) when describing the new methods applied to calculate virtual nutrients, and, more specifically, when describing the data used to assess the crop-specific fertilizer and manure data, the authors indicate to have used for the 85 minor crops the N and P balance, corresponding to the excess or deficiency of N and P compared to the current levels (and in equation 1 and 2 in the supplementary information). Al thought I agree with the procedure, I find unclear how this balance are calculated (data from which source? Are application via “seeds, irrigation water, deposition and BNF (line 425) accounted here and how?), and what the “compared to the current levels” indicates. This clarification is important because the calculation of the N and P application rates strongly influences the results presented at lines 219-234 and Figure 6. The efficiency and the sending-receiving effect due to trade really depends on the sending and receiving countries **application rate**. Thus, it is important to well understand how these rates were estimated.

Answer: Thanks for your comments! The method of cropland N and P balance was referred from MacDonald et al (Agronomic phosphorus imbalances across the world's croplands) and Liu et al (A high-resolution assessment on global nitrogen flows in cropland). The data source of crop nutrients balances was OECD Data (<https://data.oecd.org/agrland/nutrient-balance.htm>). For the Nutrient balance, it has already considered the various sources of nutrients to the soil or crops (e.g., seeds, irrigation water...). A nutrient deficit (negative value) indicates declining soil fertility. A nutrient surplus (positive data) indicates a risk of polluting soil, water, and air. The nutrient balance is defined as the difference between the nutrient inputs entering a farming system (mainly livestock manure and fertilizers) and the nutrient outputs leaving the system (the uptake of nutrients for crop and pasture production). The sentences (Line 425) in main text and Fig 1c are aiming to help readers understand the source

of nutrients more intuitively, but the calculation is still using the nutrient balance method.

To make it clearer, we revised the sentence: “In this study, source-sink method was used to consider virtual nutrients⁵⁰, the sources of virtual nutrients include the application of inorganic and organic fertilizers, seeds, irrigation water, atmospheric deposition, and biological N fixation (Fig. 1c)³⁸. We used the nutrient balance method to calculate the virtual nutrient coefficients for each crop, as detailed in the Supplementary Materials Section 1.1.”. The “compared to the current levels” is wrong sentence and we have deleted.

We have clarified in the manuscript and we also have added the references. We made estimates based on available data to obtain long time series of multi-crop data, and we have revised the sentences: “According to principle of total input nutrients consistency, the N and P application rate of 102 different crops in different countries in 2000 were obtained to the analysis. Then, we compared and verified the total nutrient input calculated by the above equilibrium method with the existing statistical data, and found that the difference between the two results was small, which could meet the calculation accuracy (compared the estimated results with nutrients application from FAOSTAT, <https://www.fao.org/faostat/en/#data/RFB>, the relative error <5%). Finally, we assumed that the changes of nutrients application of different crops were consistent with the changes of the total nutrient application. Based on this rule, the data of 2000 were linearly extended to cover the study period (2000 to 2016).”

2) concerting the application of a correction for re-exports trade data (following the procedure proposed by Kastner et al.: the authors declare at section S.1.3 (supplementary information) to have corrected re-export flows. This seems also to be the case, since they manage to separate in between the flows from sending, receiving and spillover countries. Nevertheless, in the main text at line 415, the authors declares that the re-exports flows are omitted.

Answer: Thanks for your comments! This is an important question. We think that physical nutrients are more important to reflect the physical N/P flows, so we calculated the customs import and export data and explored the distributions and changes of physical nutrient flows brought by trade. Virtual nutrient is more concerned about the impacts or risks of the production process on the local resources and environment in the producer countries or regions, so we identified and eliminated the re-export trade to distinguish the real producers and consumers of the agricultural products. To make it clearer, we revised the sentences: “Since the efficiency of nutrient use (virtual nutrient content) in agricultural production systems varies among different countries or regions, we used the data that excludes re-exports to match the nutrients flow to real producer/consumer, this method can realize the traceability of trade products and virtual nutrients flows ^{48,49}.”

Since the nutrient consumption in agricultural production systems among different countries or regions are not the same, the flow of virtual nutrients excludes re-exports based on real trade data and realizes the traceability of trade products. How the two points are compatible with each other?

Answer: Thanks for your comments! The purpose of tracing virtual nutrient flow (trade excluding re-export) pathway using this method is to provide basic data for evaluating sending-receiving effect and spillover effect. Both are essential data in the calculation step of this manuscript. We used the trade data after the elimination of re-exports to calculate the virtual nutrient flows in order to better reflect and look for the savings or waste effects of production efficiency differences in the trade process, because this helps us to find the real consumers and producers. At the same time, our analysis of the physical trade is to understand the pattern of agricultural trade and the difference between the physical and virtual nutrients more conveniently. We modified the method section to clarify the expression, and added some results to compare the physical and virtual nutrients flows. “The flow of physical nutrients considers the direct trade data from the FAO trade matrix. Since the nutrient consumption in agricultural production systems among different countries or regions are not the same, we used the data excludes re-exports, this method can reflect the real producer/consumer and realize the traceability of trade products^{48,49}. Physical nutrients, which are the physical N or P elements contained in the products harvested by the agricultural production system, are transferred from the exporter to the importer along the international trade route. Compared with Lun et al ²³ and Barbieri et al ⁵⁰, this study expanded to 320 crops, including almost all conventionally traded products, the detailed description of these selected agricultural products and their physical N and P contents are listed in Supplementary Data 1.”

Minor comments:

L 499 – code availability. This is just a suggestion: I’m a big fan of free code access, I would like to invite the authors to deposit their code on an online repository (GitHub for example)

Answer: Thanks for your suggestions! We will make the code freely available after the paper is published, which will help more scholars understand our work and conduct subsequent research.

L114 (and elsewhere): those are flows per year. I would make this explicit in the units : Tg P or N / yr

Answer: Thanks for your comments! We have revised the expression.

“The total physical nutrient flows increased from 10.3 Tg N to 27.1 Tg N and from 1.4 Tg P to 3.5 Tg P, respectively, between 1997 and 2016 (Fig. 2a, b); the global virtual nutrient flows increased from 13.4 Tg N to 36.6 Tg N and from 8.9 Tg P to 24.5 Tg P, respectively (Fig. 2c, d).”

“Fig. 4 shows the top 5 net receiving and sending systems of nutrients during 2016. China was the largest net receiving system of physical nutrients with 6.06 Tg N and 0.62 Tg P during 2016, accounting for 20.0% and 16.0% of the total global physical N and P flows, respectively. Thereafter, Japan (0.95 Tg N and 0.13 Tg P) and Mexico (0.78 Tg N and 0.23 Tg P) were the second and third ranked receiving systems. The top 5 net receiving systems imported 33% of the total physical nutrient flows. On the other hand, the United States (5.20 Tg N and 0.55 Tg P) and Brazil (4.21 Tg N and 0.39 Tg P) generated the largest net exports during 2016.”

“The global physical flows were up to 23.3 Tg N and 3.02 Tg P, respectively. Trade in agricultural products involves a large volume of nutrients and has important implications for global nutrient redistribution. The global virtual sending-receiving effect was up to 62.3 Tg N and 73.9 Tg P, respectively.”

L116 : the estimation of virtual P flown in 2016 (24.5 Tg/yr) is largely higher than the amount estimated by other studies (for example, see Barbieri et al. 2021: Food system resilience to phosphorus shortages on a telecoupled planet → 5.4 Tg/yr). This is partially due to the fact that previous works have considered only P fertiliser application and not manure application, but the difference is still considerable. What do you think is the main source of difference between these estimations? It might be interesting to underline such discrepancy in the manuscript.

Answer: Thanks for your comments! In this paper, the calculation boundary and method are explained, we have revised the sentences in Method section: “Compared with Lun et al²³ and Barbieri et al⁵⁰, this study expanded to 320 crops, including almost all conventionally traded products, the detailed description of these selected agricultural products and their physical N and P contents are listed in Supplementary Data 1.”

“In this paper, source-sink analysis was used to calculate virtual nutrients. In addition to fertilizer input, natural input is considered to make the results more complete. The sources of virtual nutrients include the application of inorganic and organic fertilizers, seeds, irrigation water, atmospheric deposition, and biological N fixation (Fig. 1c)³⁸. The nutrients, which are not absorbed by the products, e.g., discharged into the water or atmosphere and accumulated in the soil, will lead to negative impacts on the environment⁵⁰. While physical nutrients are considered as nutrients contained in traded products, virtual nutrients are the total inputs in the production of agricultural products, whether being absorbed or not. Therefore, the flows of virtual nutrients involve the importer transferring inputs and losses of nutrients in the production process to the exporter^{51,52}. The virtual nutrient contents of various agricultural products from each country or region and their specific calculation methods are explained in Table S3 and Supplementary Materials Sections 1.2 and 1.3.”

Figure 3: a lot of information here: maybe write explicitly the information shown for each set of

maps on the side of each line and columns (i.e. N and P / gross export – import – budgets) etc. to make the figure easy reading without having to jump from the caption to the figure?

Answer: Thanks for your comments! We have revised Figure 3.

Fig. 3. Nitrogen (N) and phosphorus (P) receiving and sending systems in 1997(a-c, g-i) and 2016(d-f, j-l). N flows through gross export (a, d), gross import (b, e) and net budget (c, f); P flows through gross export (g, j), gross import (h, k) and net budget (i, l). Net nutrient flows were calculated as the difference between gross physical N (or P) imported and gross physical N (or P) exported. The red and blue colors indicate exporting and importing, respectively; gray color indicates that no data was available.

Figure 4: this figure is interesting. I am just a bit confused with the colors and the representation of the Net P and N import both for physical and virtual flows. Are this represented by the small bars at the end of the orange bars for instance? I think this could be better represented.

Answer: Thanks for your comments! Based on reviewer #1's suggestion, we have revised the Figure to make it more readable.

Fig. 4. Top 5 net nutrient receiving systems and sending systems in 2016. a. Top 5 net receiving systems and sending systems of nitrogen (N); b. Top 5 net receiving systems and sending systems of phosphorus (P). Countries are ordered by the net volume of physical N or P they imported or exported. The green and red labels indicate the volume of net import and export of physical nutrients, which is the length of the green column minus the length of the red column. The yellow and blue labels represent volume of net import and export of virtual nutrients, which is the length of the yellow column minus the length of the blue column. Results for all countries are shown in Supplementary Data 3. Note that the red and yellow columns start at 0, while the green or blue columns start at the bottom of the red or yellow columns. Therefore, the part where the two colors do not coincide is the net import or export value.

L290: I totally agree, this is an important point for the research agenda.

Answer: Thanks!

Figure 6a: most of inefficient trade effect for both physical and virtual flows are identified with the trade of meat products. I think this important results may merit to be highlighted in the text (for example, at lines 330 in the discussion?)

Answer: Thanks for your comments! We have revised the paragraph: “Reducing meat consumption may help reduce inefficient production at the consumer end. For example, the

United Nations has recommended that EU member states reduce their consumption of meat and dairy products. At the same time, it is also necessary to upgrade technology in countries or regions with low production efficiency.”

L365; why export products should have a different nutritional content than non-export products? Or do you mean that we should account for the type of products that are exported vs non-exported and, so, analyses in details the trade effects based on their nutritional content?

Answer: Thanks for your comments! This suggestion was put forward by reviewer #3 (round 1 revision), but it is difficult to realize due to the lack of relevant data at present. Our review of relevant reports or studies indicate that some export goods may differ from local domestic consumption goods (e.g., processing technology, quality, etc.), so we have briefly discussed them here. “Restricted by the relevant data, this study did not consider the difference of nutritional content between export products and non-export products. Future studies could further consider the difference between these two kinds of products to clarify the trade effect more precisely (see the uncertainty analysis in Supplementary materials Section S2).”

Reviewer #2 (Remarks to the Author):

The authors have addressed most of my previously raised comments. I thus think that the manuscript is almost ready for publication.

I just have a couple of additional remarks that have not been fully addressed:

1. You mention that data on N and P balances comes from OECD (Supplementary Materials and rebuttal letter). Looking at the source you cite, balance data are only available for a few set of the 221 countries considered in the study (OECD countries). I still find unclear which data has been used for all missing countries. I found that the clarifications about such calculation have not been fully addressed.

2. In your rebuttal letter you mention to have changed all units from TG N / P to Tg N or P yr⁻¹, but I cannot see any change in the units used throughout the manuscript. Please address this issue.

3. Figure 4: I still find this figure quite complex and difficult to interpret/understand. I do not see any particular change as in comparison to the previous version (I couldn't notice any change in the colors / bars etc.). This point has been raised by the 1st Reviewer too, so please address it.

Minor comments:

Figure 5: you report here the most important inefficient trade routes, but the definition of an efficient and inefficient route is given only at lines 250-251. The definition should be given earlier.

Lines 67-70: the sentence is uncompleted or missing something

Pietro Barbieri

Points-by-points responses(bold) to reviewer's comments(regular)

Reviewer #2 (Remarks to the Author):

The authors have addressed most of my previously raised comments. I thus think that the manuscript is almost ready for publication.

I just have a couple of additional remarks that have not been fully addressed:

Answer: Thanks for your thorough review and valuable feedback on our manuscript. Your positive comments mean a lot to us. I want to assure you that we will carefully address all the suggestions and comments you have provided and do our best to improve the manuscript to the best of our ability. Thank you once again for your time and efforts.

1. You mention that data on N and P balances comes from OECD (Supplementary Materials and rebuttal letter). Looking at the source you cite, balance data are only available for a few set of the 221 countries considered in the study (OECD countries). I still find unclear which data has been used for all missing countries. I found that the clarifications about such calculation have not been fully addressed.

Answer: Thanks for your comments! Sorry for the unclear writing. In the first sentence of this paragraph, we explained that one of our data sources was the EarthStat. We compared and supplemented this data from the OECD, these two databases are used at the same time. To ensure clarity in our explanation, we have made modifications to these sentences, which now write as follows (Supplementary Information Section S1.1.1):

“For the calculation of N and P application rates for the other 85 minor crops, as direct data on N and P application rates were not available, crop-specific yield, N and P content, crop-specific harvest area fraction and N and P balance which responded to excess or deficiency of nitrogen and phosphorus compared to the current level were used for the calculation, as shown in the following equation:

$$N_{app} = crop_{yield} \times N_{content} + crop_{HAF} \times crop_{balance} \quad (1)$$

where N_{app} is N application rate, kg; $crop_{yield}$ is crop-specific yield, kg; $N_{content}$ is N content; $crop_{HAF}$ is harvested area fraction of the specific crop; $crop_{balance}$ is the N balance of crop, kg, data from EarthStat and (<http://www.earthstat.org/total-nutrient-balance-140-crops/>) and OECD (<https://data.oecd.org/agrland/nutrient-balance.htm>).

The calculation process for phosphorus was similar:

$$P_{app} = crop_{yield} \times P_{content} + crop_{HAF} \times crop_{balance} \quad (2)$$

where P_{app} is P application rate, kg; $crop_{yield}$ is crop-specific yield, kg; $P_{content}$ is P content; $crop_{HAF}$ is harvested area fraction of the specific crop; $crop_{balance}$ is the P balance of crop, kg, data from EarthStat and (<http://www.earthstat.org/total-nutrient-balance-140-crops/>) and OECD (<https://data.oecd.org/agrland/nutrient-balance.htm>).”

2. In your rebuttal letter you mention to have changed all units from TG N / P to Tg N or P yr⁻¹, but I cannot see any change in the units used throughout the manuscript. Please address this issue.

Answer: Thanks for your comments! We have checked the whole manuscript and corrected. There have been no instances of the expressions "/" or "or" in the manuscript so far.

3. Figure 4: I still find this figure quite complex and difficult to interpret/understand. I do not see any particular change as in comparison to the previous version (I couldn't notice any change in the colors / bars etc.). This point has been raised by the 1st Reviewer too, so please address it.

Answer: Thanks for your comments! To make it easier to read, we have adjusted the style of the Figure. We adjusted and added to the coordinate axes, re-exported the images, and increased the contrast to make it clearer, see detail as follows:

Fig. 4. Top 5 net nutrient receiving systems and sending systems in 2016. a. Top 5 net receiving systems and sending systems of nitrogen (N); b. Top 5 net receiving systems and sending systems of phosphorus (P). Countries are ordered by the net volume of physical N or P they imported or exported. The green and red labels indicate the volume of net import and export of physical nutrients, which is the length of the green column minus the length of the red column. The yellow and blue labels represent volume of net import and export of virtual nutrients, which is the length of the yellow column minus the length of the blue column. Results for all countries are shown in Supplementary Data 3. Note that the red and yellow columns start at 0, while the green or blue columns start at the bottom of the red or yellow columns. Therefore, the part where the two colors do not coincide is the net import or export value.

Minor comments:

Figure 5: you report here the most important inefficient trade routes, but the definition of an efficient and inefficient route is given only at lines 250-251. The definition should be given earlier.

Answer: Thanks for your comments! We have added the definition in Methods section. “If $NSRE_{v,i}$ or $PSRE_{v,i} > 0$, it indicates that the redistribution of agricultural product trades on the utilization of nutrient resources present a positive sending-receiving effect of saving N or P, which are defined as efficient flows; if $NSRE_{v,i}$ or $PSRE_{v,i} < 0$, it indicates a negative sending-receiving effect on the utilization of nutrient resources of wasting N or P, which are defined as inefficient flows; n is the total number of nutrients flows.”

Lines 67-70: the sentence is uncompleted or missing something

Answer: Thanks for your careful comments! We have revised this sentence: “Sending, receiving and spillover systems are fundamental subsets of a telecoupled system, as for global trade, there are many such individual subsets (pairs of trading countries) in the trade networks although their roles are not unchangeable (a particular country can be both an exporter/sender and an importer/receiver), the simplified concept of a spillover system and its effects within the telecoupling framework is shown in Supplementary Information Fig. S1.”